# FROM RISK TO UNCERTAINTY: GENERATING PREDICTIVE UNCERTAINTY MEASURES VIA BAYESIAN ESTIMATION

**Nikita Kotelevskii**[1,2]*    **Vladimir Kondratyev**[3]    **Martin Takáč**[1]    **Éric Moulines**[3,1]

**Maxim Panov**[1]

[1]Department of Machine Learning, MBZUAI, UAE
[2]CAIT, Skoltech, Russia
[3]CMAP, École polytechnique, France

## ABSTRACT

There are various measures of predictive uncertainty in the literature, but their relationships to each other remain unclear. This paper uses a decomposition of statistical pointwise risk into components, associated with different sources of predictive uncertainty, namely aleatoric uncertainty (inherent data variability) and epistemic uncertainty (model-related uncertainty). Together with Bayesian methods, applied as an approximation, we build a framework that allows one to generate different predictive uncertainty measures. We validate our method on image datasets by evaluating its performance in detecting out-of-distribution and misclassified instances using the AUROC metric. The experimental results confirm that the measures derived from our framework are useful for the considered downstream tasks.

## 1 INTRODUCTION

Nowadays, predictive models are applied in a variety of fields requiring high-risk decisions such as medical diagnosis (Shen et al., 2017; Litjens et al., 2017), finance (Ozbayoglu et al., 2020; Heaton et al., 2017), autonomous driving (Grigorescu et al., 2020; Mozaffari et al., 2020) and others. A careful analysis of model predictions is required to mitigate the risks. Hence, it is of high importance to evaluate the predictive uncertainty of the models. In recent years, a variety of approaches to quantify predictive uncertainty have been proposed (Kotelevskii et al., 2022a; Lahlou et al., 2023; Kendall & Gal, 2017; Van Amersfoort et al., 2020; Liu et al., 2020a; Lakshminarayanan et al., 2017; Malinin & Gales, 2021; Schweighofer et al., 2023a;b) and others. Specific attention has been paid to the distinction between different sources of uncertainty. It is commonly agreed to consider two sources of uncertainty (Hüllermeier & Waegeman, 2021): **aleatoric** uncertainty, that effectively reduces to the inherent stochastic relationship between the inputs (objects) and the outputs (labels), and **epistemic** uncertainty, which is referred to as the uncertainty due to the "lack of knowledge" about the true data distribution. Distinguishing between aleatoric and epistemic uncertainties is crucial in practice because it helps identifying whether uncertainty can be reduced by gathering more data (epistemic) or if it is inherent to the problem (aleatoric), thus guiding better decision-making and model improvement.

Despite the practical importance and widespread usage of uncertainty quantification, *there is still no common strict formal definition of both types of uncertainty*. This leads to a number of different measures to quantify either type of uncertainty (see for example (Lakshminarayanan et al., 2017; Gal et al., 2017; Malinin & Gales, 2021; Hüllermeier & Waegeman, 2021; Kotelevskii et al., 2022a; Schweighofer et al., 2023a)). Recently, for information-theoretical measures, the step towards unified definition was made in (Schweighofer et al., 2023a). However, it is not clear how all these measures

---

*Corresponding author. `nikita.kotelevskii@mbzuai.ac.ae`

of uncertainty are related to each other in general. Do they complement or contradict each other? Are they special cases of some general class of measures? In this paper, we introduce a statistical approach for predictive uncertainty quantification, reasoning in terms of pointwise risk estimation. This allows us to reconstruct a lot of known and popular measures of predictive uncertainty, as well as show how to build new ones. Our contributions are as follows:

1. Following (Kotelevskii et al., 2022a; Lahlou et al., 2023; Liu et al., 2019), we consider pointwise risk decomposition into distinct parts, that are responsible for capturing different sources of uncertainty. We show, that this decomposition, applied specifically to strictly proper scoring rules (Gneiting & Raftery, 2007; Gruber & Buettner, 2023) leads to a general framework, that is amenable for generation of uncertainty measures.

2. We show how one can make this framework practical with the help of Bayesian estimation (see Section 4). We show that commonly used measures of epistemic uncertainty, such as Mutual Information (Houlsby et al., 2011; Gal et al., 2017), Expected Pairwise Kullback-Leibler divergence (EPKL) (Schweighofer et al., 2023a; Malinin & Gales, 2018), predictive variance and maximum probability score are special cases within our general approach. We also highlight the limitations of our framework, elaborating on discussions from (Schweighofer et al., 2023a; Wimmer et al., 2023).

3. We experimentally evaluate different predictive uncertainty quantification measures from the proposed framework in various tasks. Specifically, we consider out-of-distribution detection and misclassification detection; see Section 6. Our results highlight the conditions under which each measure is most effective, providing practical insights for selecting appropriate uncertainty measures.

## 2 PREDICTIVE UNCERTAINTY QUANTIFICATION VIA RISKS

Assume we have a dataset $D_{tr} = \{(X_i, Y_i)\}_{i=1}^N$, where pairs $X_i \in \mathbb{R}^d, Y_i \in \mathcal{Y}$ are i.i.d. random variables sampled from a joint training distribution $P_{tr}(X, Y)$. We consider a classification task over $K$ classes, i.e. $\mathcal{Y} = \{1, \ldots, K\}$. We can express this joint distribution as a product: $P_{tr}(X, Y) = P_{tr}(Y \mid X)P_{tr}(X)$.

In practice, we typically consider a parametric model $P(Y \mid X, \theta)$ with parameters $\theta$ to approximate $P_{tr}(Y \mid X)$. We denote the true class probabilities for an input $x$ as $\eta(x) = P_{tr}(Y \mid X = x)$, and the predicted probabilities as $\eta_\theta(x) = P(Y \mid X = x, \theta)$. We will often omit the index $\theta$ and denote the predicted probability vector by $\hat{\eta}$.

### 2.1 POINTWISE RISK AS A MEASURE OF UNCERTAINTY

The goal of uncertainty quantification is to measure the degree of confidence of predictive models, distinguishing between aleatoric and epistemic sources of uncertainty. In the paper, following (Kotelevskii et al., 2022a; Lahlou et al., 2023), we introduce uncertainty via the statistical concept of risk.

In machine learning, the main concern is the model's "error" at a particular input point $x$. One way to express this error is through the expected risk. Let $\ell \colon \mathbb{R}^K \times \mathcal{Y} \to \mathbb{R}$ be a loss function that measures how well $\hat{\eta}(x)$ matches the true label $y$. The pointwise risk $\mathrm{R}(\eta, \hat{\eta} \mid x)$ for a model $\hat{\eta}$ is defined as:

$$\mathrm{R}(\eta, \hat{\eta} \mid x) = \int \ell(\hat{\eta}(x), y) \, dP(y \mid X = x). \tag{1}$$

Thus, pointwise risk is an expected loss received by a specific predictor $\hat{\eta}$ at a particular input point $x$. Importantly, pointwise risk, while being a natural measure of expected model error, can not be used directly as a measure of uncertainty as it is not possible to compute it due to unknown true data distribution. We will discuss possible ways to transform pointwise risk into the practical uncertainty measure in Section 4.

Note, that we use distribution $P$, which might differ from $P_{tr}$. If $P(X) \neq P_{tr}(X)$ but $P(Y \mid X = x) = P_{tr}(Y \mid X = x)$ for any $x$, this situation is called "covariate shift". We consider this setup and assume $\eta(x) = P(Y \mid X = x)$ is *valid* for any $x$, meaning it is a vector of length $K$ regardless of the input. Limitations of this assumption are discussed in Appendix B.

## 2.2 Aleatoric and Epistemic Uncertainties via Risks

Predictive uncertainty can be divided into two parts. **Aleatoric** uncertainty expresses the degree of ambiguity in data and does not depend on the model, being an inherent property of the data given a particular choice of feature representation. **Epistemic** uncertainty is vaguely defined but typically is associated with the "lack of knowledge" of choosing the right model parameters $\theta$ and model misspecification. When the source of uncertainty is not important, practitioners consider **total** uncertainty that aggregates all the sources.

Pointwise risk allows for the following decomposition:

$$\underbrace{\mathrm{R}(\eta, \hat{\eta} \mid x)}_{\text{Total risk}} = \underbrace{\mathrm{R}_{\text{Bayes}}(\eta \mid x)}_{\text{Bayes risk}} + \underbrace{\mathrm{R}(\eta, \hat{\eta} \mid x) - \mathrm{R}_{\text{Bayes}}(\eta \mid x)}_{\text{Excess risk}}, \tag{2}$$

where $\mathrm{R}_{\text{Bayes}}$ is the pointwise Bayes risk, defined as:

$$\mathrm{R}_{\text{Bayes}}(\eta \mid x) = \int \ell(\eta(x), y) \, dP(y \mid X = x).$$

The Bayes risk represents the expected error from the true data-generative process $\eta(x)$. It does not depend on the parameters of the model nor the choice of model architecture, and hence can be seen as a measure of *aleatoric* uncertainty. Note that in our analysis, we treat $x$ as a noiseless observed variable. Alternatively, one could assume that only noisy observations $\tilde{x}$ are available and include the uncertainty in observations into the definition of aleatoric uncertainty. However, we do not explore this scenario in this work. The second term in equation (2) is "Excess risk" and represents the difference between the risks computed for the approximation and for the true model at a given input point $x$. Thus, it naturally represents a lack of knowledge about the true data distribution, i.e. *epistemic* uncertainty. We note that decomposition (2) was previously considered in the context of uncertainty quantification in (Kotelevskii et al., 2022a; Lahlou et al., 2023).

Although the decomposition (2) is useful, it doesn't provide much information about the properties of these risk functions in general cases. Therefore, we consider a specific class of loss functions, strictly proper scoring rules, which allows us to do a theoretical analysis.

## 3 Risks for Strictly Proper Scoring Rules

**Strictly proper scoring rules** (Gneiting & Raftery, 2007) represent a class of loss functions that ensure that the minimizing predictive distributions coincide with the data-generating distribution $P(Y \mid X)$. Let's say a forecaster can produce a vector of predicted probabilities $P \in \mathcal{P}_K$, where $\mathcal{P}_K$ is a space of discrete probability distributions over $K$ classes. Then, $\ell(P, y) \colon \mathcal{P}_K \times \mathcal{Y} \to \mathbb{R}$ is the penalty the forecaster would have, given that event $y$ is materialized. Its expected value with respect to some distribution $Q$ we will denote as $\ell(P, Q) = \int \ell(P, y) dQ(y)$.

A scoring rule is called *strictly proper* if, for any $P, Q \in \mathcal{P}_K$, it satisfies the condition $\ell(P, Q) \geq \ell(Q, Q)$, with equality holding only when $P = Q$. Under mild assumptions (see Theorem 3.2 in (Gneiting & Raftery, 2007)), any strictly proper scoring rule can be represented as:

$$\ell(\eta, y) = \langle G'(\eta), \, \eta \rangle - G'_y(\eta) - G(\eta),$$

where $\langle . , . \rangle$ is a scalar product, $G \colon \mathcal{P}_K \to \mathbb{R}$ is a strictly convex function, and $G'(\eta) = \{G'_1(\eta), \dots G'_K(\eta)\}$ is a vector of element-wise subgradients.

**Risk decompositions for strictly proper scoring rules.** Here we derive the explicit expressions for different types of risks for strictly proper scoring rules (detailed derivations are given in Appendix C).

- **Total Risk (Total Uncertainty):**

$$\mathrm{R}_{\text{Tot}}(\eta, \hat{\eta} \mid x) = \langle G'(\hat{\eta}), \, \hat{\eta} \rangle - G(\hat{\eta}) - \langle G'(\hat{\eta}), \, \eta \rangle. \tag{3}$$

Note, that Total risk depends *linearly* on the true data generative distribution $\eta$.

- **Bayes Risk (Aleatoric Uncertainty):**

$$\mathrm{R}_{\text{Bayes}}(\eta \mid x) = -G(\eta). \tag{4}$$

Note, that Bayes risk is a concave function of $\eta$, since function $G$ is convex.

| | **Log score** | **Brier score** | **Zero-one score** | **Spherical score** |
|---|---|---|---|---|
| $G(\eta)$ | $\sum_{k=1}^{K} \eta_k \log \eta_k$ | $-\sum_{k=1}^{K} \eta_k(1-\eta_k)$ | $\max_k \eta_k - 1$ | $\|\eta\|_2 - 1$ |
| $\bar{\eta}$ | $\frac{\exp \mathbb{E}\log\eta}{\sum_{k'}\left(\exp\mathbb{E}\log\eta\right)_{k'}}$ | $\mathbb{E}\eta$ | Not defined | $\|x_0\|_2\left[n + \frac{m}{\sqrt{1-\|m\|_2^2}}\right]$ |
| **Aleatoric** (Bayes risk) | $\mathbb{H}\eta$ | $1-\|\eta\|_2^2$ | $1-\max_k \eta_k$ | $1-\|\eta\|_2$ |
| **Epistemic** (Excess risk) | $D_{\mathrm{KL}}[\eta\|\hat{\eta}]$ | $\|\eta-\hat{\eta}\|_2^2$ | $\max_k \eta_k - \eta_{\arg\max_k \hat{\eta}_k}$ | $\|\eta\|_2(1-\langle\frac{\hat{\eta}}{\|\hat{\eta}\|_2}, \frac{\eta}{\|\eta\|_2}\rangle)$ |
| **Total** (Risk) | $\mathrm{CE}[\eta\|\hat{\eta}]$ | $\|\eta-\hat{\eta}\|_2^2 - \|\eta\|_2^2 + 1$ | $1-\eta_{\arg\max_k \hat{\eta}_k}$ | $1-\|\eta\|_2\langle\frac{\hat{\eta}}{\|\hat{\eta}\|_2}, \frac{\eta}{\|\eta\|_2}\rangle$ |

Table 1: Expressions for risks and central predictions, computed for different strictly proper scoring rules. We omitted $x$ for clarity. $D_{\mathrm{KL}}$ stands for Kullback-Leibler divergence and CE for Cross-Entropy. See Appendix D for full derivations of risk instantiations and Appendix L for central predictions.

- **Excess risk (Epistemic Uncertainty)**:

$$\mathrm{R}_{\mathrm{Exc}}(\eta, \hat{\eta} \mid x) = D_G\big(\eta \,\|\, \hat{\eta}\big), \tag{5}$$

  where $D_G\big(\eta \,\|\, \hat{\eta}\big) = G(\eta) - G(\hat{\eta}) - \langle G'(\hat{\eta})\,,\, \eta - \hat{\eta}\rangle$ is Bregman divergence (Bregman, 1967).

In what follows, we will assume the dependency of risks on $\eta, \hat{\eta}$ and for clarity will omit it, writing it as a function of $x$ instead.

**Specific Instances of Proper Scoring Rules.** Different choices of the convex function $G$ lead to different proper scoring rules. Table 1 shows the results for some popular cases often used in machine learning algorithms (see detailed derivations in Appendix D).

From Table 1, we see, that some of the risks correspond to well-known aleatoric uncertainty measures. For example, the Bayes risk for the Log score is given by the entropy of the predictive distribution. For the Zero-one score, this component is given by the so-called MaxProb, also widely applied (Geifman & El-Yaniv, 2017; Kotelevskii et al., 2022a; Lakshminarayanan et al., 2017). For the Excess risk, we obtain different examples of Bregman divergence which lead to some well-known uncertainty measures when coupled with the Bayesian approach to risk estimation (see Section 4).

**Estimating risks.** The derived equations are useful but require access to the true data-generative distribution $\eta$, which is typically unknown. One approach to deal with this problem was introduced in (Kotelevskii et al., 2022a), where authors considered a specific model $\eta_\theta$, namely Nadaraya-Watson kernel regression, as it has useful asymptotic properties to approximate Excess risk. Another approach, the DEUP (Lahlou et al., 2023) proposed a method for estimating Excess risk by directly training a model to predict errors. However, in general cases, it is hardly possible to derive these results. In this paper, we consider a Bayesian approach to approximate $\eta$ that allows us to derive both well-known from the literature and new uncertainty measures based on one unified framework.

## 4 BAYESIAN RISK ESTIMATION

The derived equations for risks depend on the true data generative distribution $\eta$ and on some approximation of it $\hat{\eta}$. In particular, $\eta$ appears in all the risks and is unknown. One needs to deal with that to obtain a computable uncertainty measure. In the Bayesian paradigm, one considers a posterior distribution over model parameters $p(\theta \mid D_{tr})$ that immediately leads to a distribution over predictive distributions $\eta_{\theta|D_{tr}}$. The goal of this section is to give a complete recipe for risk estimation under the Bayesian approach.

One can think of the risks as functions of $\eta$ and $\hat{\eta}$, namely $g(\eta, \hat{\eta})$, where $g$ is a shortcut for any risk function (Bayes risk is a function of only one argument). We can approximate the risks with the help of posterior distribution using one of three ideas:

1. **Bayesian averaging of risk.** One can consider $\mathbb{E}_\theta g(\eta_\theta, \hat{\eta})$ to approximate an impact of the true model $\eta$. In a fully Bayesian paradigm, the same can be done with $\hat{\eta}$ leading to the fully Bayesian risk estimate $\mathbb{E}_\theta\mathbb{E}_{\tilde{\theta}}\, g(\eta_{\tilde{\theta}}, \eta_\theta)$.

2. **Central label** (Posterior predictive distribution). One may use posterior predictive distribution $\hat{\eta}_{D_{tr}} = \arg\min_z \mathbb{E}_\theta D_G\big(\eta_\theta \,\|\, z\big) = \mathbb{E}_\theta\eta_\theta$ and plug it in risk equations instead of $\eta$

and/or $\hat{\eta}$. This estimate naturally appears in decompositions of Bregman divergences (Pfau, 2013; Adlam et al., 2022), and in this context it is called "central label".

3. **Central prediction.** Interestingly, there is another natural estimate, that appears in the literature on Bregman divergences (Pfau, 2013; Adlam et al., 2022; Gruber & Buettner, 2023) and is referred to as "central prediction". It is defined as $\bar{\eta} = \arg\min_z \mathbb{E}_\theta D_G(z \parallel \eta_\theta)$.

In general, one can consider nine approximations of Excess and Total risks (three approximation options for each of the two arguments) and three of Bayes risk. We denote the specific type of approximation listed above by positional indices 1, 2 and 3. For example, $\tilde{g}^{(1,1)} = \mathbb{E}_{\tilde{\theta}} \mathbb{E}_\theta \, g(\eta_{\tilde{\theta}}, \eta_\theta)$, $\tilde{g}^{(3,1)} = \mathbb{E}_\theta g(\bar{\eta}, \eta_\theta)$, and so on.

In this section, we present some of the formulas for the resulting aleatoric, and epistemic uncertainty measures with brief remarks on different approximations. For detailed derivations, other approximation options (not listed in the main part) and discussions, refer to Appendix E.

For **Bayes risk** we obtain three cases as it doesn't depend on $\hat{\eta}$:

$$\tilde{R}_{\text{Bayes}}^{(1)}(x) = -\mathbb{E}_\theta G(\eta_\theta), \quad \tilde{R}_{\text{Bayes}}^{(2)}(x) = -G(\hat{\eta}_{D_{tr}}) \quad \text{and} \quad \tilde{R}_{\text{Bayes}}^{(3)}(x) = -G(\bar{\eta}).$$

For **Excess risk** we obtain the whole family of approximations. We list some of them here (see others in Appendix E):

- **Expected Pairwise Bregman Divergence (EPBD):**
$$\tilde{R}_{\text{Exc}}^{(1,1)}(x) = \mathbb{E}_{\tilde{\theta}} \mathbb{E}_\theta D_G(\eta_{\tilde{\theta}} \parallel \eta_\theta).$$
In a special case of Log score, it is an Expected Pairwise KL (EPKL; Malinin & Gales (2021); Schweighofer et al. (2023a)).

- **Bregman Information (BI):**
$$\tilde{R}_{\text{Exc}}^{(1,2)}(x) = \mathbb{E}_\theta D_G(\eta_\theta \parallel \hat{\eta}_{D_{tr}}).$$
In a special case of Log score, it is Mutual Information (Gal et al., 2017; Houlsby et al., 2011; Malinin & Gales, 2018). In case of Brier score, it is sum of predictive variances (class-wise).

- **Reverse Bregman Information (RBI):**
$$\tilde{R}_{\text{Exc}}^{(2,1)}(x) = \mathbb{E}_\theta D_G(\hat{\eta}_{D_{tr}} \parallel \eta_\theta).$$
Its special case for Log score is known as Reverse Mutual Information (RMI; Malinin & Gales (2021)).

- **Modified Bregman Information (MBI):**
$$\tilde{R}_{\text{Exc}}^{(1,3)}(x) = \mathbb{E}_\theta D_G(\eta_\theta \parallel \bar{\eta}).$$
It is similar to Bregman Information, but the deviation is computed from the "central prediction", not the central label (BMA).

- **Modified Reverse Bregman Information (MRBI):**
$$\tilde{R}_{\text{Exc}}^{(3,1)}(x) = \mathbb{E}_\theta D_G(\bar{\eta} \parallel \eta_\theta).$$
This has similar structure to Reverse Bregman Information (RBI). Again, the deviation here is computed from another "central prediction".

One advantage of Bayesian models over point predictors is that they can give different predictions for the same input. This means we don't have to average out uncertainty; instead, we can look at the disagreements between different versions of the model. From this perspective, EPDB is especially promising because it doesn't rely on averaging but considers all possible pairwise disagreements between models. Therefore it does not average out uncertainty.

We observe that the general approach presented in this work allows us to obtain many existing uncertainty measures in the case of the Log score loss function while leading to the whole family of new measures (see Table 1). We refer to Appendix E for additional discussion and to Appendix F for the discussion of connections of these approximations between each other. The limitations of the approach are discussed in detail in Appendix B.

### 4.1 BEST RISK APPROXIMATION CHOICE

Given the plethora of measures, one may fairly ask if there is a best risk approximation of each type. For Excess and Total risks, we elaborate on it in Appendix E. Here, we address the choice for Bayes risk.

Recall, that for the estimation of Bayes risk we have three options. However, it is not clear, which of these approximations is better. To investigate it, we assume that there exists a vector of true parameters $\theta^*$, that for any input point $x$ the following holds: $\eta(x) = p(y \mid x) = p(y \mid x, \theta^*)$. Note, that according to equation (4), Bayes risk is concave. Using $r(\cdot) = -\mathbb{E}_x G(\cdot)$, the following follows from Jensen's inequality:

$$\mathbb{E}_\theta r(\eta_\theta) \leq r(\mathbb{E}_\theta \eta_\theta).$$

At the same time, we know that for Bayes risk the following must hold for any $\theta$:

$$r(\eta) = r\big(p(y \mid x, \theta^*)\big) \leq r(\eta_\theta).$$

Hence, the following holds: $r(\eta) \leq \mathbb{E}_\theta r(\eta_\theta)$. Therefore, we have the following relation:

$$r(\eta) \leq \mathbb{E}_\theta r(\eta_\theta) \leq r(\mathbb{E}_\theta \eta_\theta), \tag{6}$$

or in other words that $\mathrm{R}_{\mathrm{Bayes}}^{(1)}(x)$ is tighter approximation (on average), than $\mathrm{R}_{\mathrm{Bayes}}^{(2)}(x)$. However, there is seemingly no general answer for $r(\bar{\eta})$.

### 4.2 CONNECTION TO ENERGY-BASED MODELS

Interestingly, we can establish a connection of our framework and energy-based models (Liu et al., 2020b). Let us consider a specific instantiation of MRBI for Logscore. Also, let us recap the free energy function $E(x; f_\theta)$ from (Liu et al., 2020b; Grathwohl et al., 2020):

$$E(x; f_\theta) = -T \log \sum_{j=1}^{K} \exp\Big[\frac{f_\theta(x)}{T}\Big]_j.$$

It can be shown (see Appendix G) that

$$\tilde{\mathrm{R}}_{\mathrm{Exc}}^{(3,1)}(x) = \frac{1}{T}\big(E(x; \mathbb{E}_\theta f_\theta) - \mathbb{E}_\theta E(x; f_\theta)\big). \tag{7}$$

From this decomposition we see, that $\tilde{\mathrm{R}}_{\mathrm{Exc}}^{(3,1)}$ is the difference between two different Bayesian estimates of the energy, where the first term is the energy, induced by expected logit, while second is expected energy, induced by each individual set of parameters. Therefore, it is interesting to check, whether $\tilde{\mathrm{R}}_{\mathrm{Exc}}^{(3,1)}$, predicted by our framework, will be better, than either of its components.

## 5 RELATED WORK

The field of uncertainty quantification for predictive models, especially neural networks, has seen rapid advancements in recent years. Among these, methods allowing explicit uncertainty disentanglement are particularly interesting due to the ability to use estimates of different sources of uncertainty in various downstream tasks. For instance, epistemic uncertainty is effective in out-of-distribution data detection (Hüllermeier & Waegeman, 2021; Mukhoti et al., 2023; Kotelevskii et al., 2022b; Vazhentsev et al., 2023; Kotelevskii et al., 2024) and active learning (Gal et al., 2017; Beluch et al., 2018).

Another direction in uncertainty quantification involves credal set-based approaches, which represent uncertainty using sets of probability distributions. Caprio et al. (2023) introduced Credal Bayesian Deep Learning, leveraging credal sets for more conservative uncertainty estimates, while Caprio et al. (2024) proposed a novel Bayes' theorem for upper probabilities. These methods offer robust uncertainty quantification but can be computationally intensive, limiting their scalability.

Bayesian methods have become popular because they naturally handle distributions of model parameters, leading to prediction uncertainty. Exact Bayesian inference is very computationally

expensive (Izmailov et al., 2021), so many lightweight versions are used in practice (Gal & Ghahramani, 2016; Blei et al., 2017; Lakshminarayanan et al., 2017; Tsymbalov et al., 2018; Shelmanov et al., 2021; Fedyanin et al., 2021; Thin et al., 2020; 2021). Early approaches inspired by Bayesian ideas (Gal et al., 2017; Kendall & Gal, 2017; Lakshminarayanan et al., 2017) used information-based measures like Mutual Information (Houlsby et al., 2011; Malinin & Gales, 2021) to quantify epistemic uncertainty and measures like entropy or maximum probability for aleatoric uncertainty. These methods, despite different computational costs, are widely used in the field.

However, the practical expense of Bayesian inference, even in its approximate forms, has led to the introduction of more simplified approaches. Some of these methods leverage hidden neural network representations, considering distances in their hidden space as a proxy for epistemic uncertainty estimation (Van Amersfoort et al., 2020; Liu et al., 2020a; Kotelevskii et al., 2022a; Vazhentsev et al., 2022; Mukhoti et al., 2023). While they offer the advantage of requiring only a single pass over the network, their notion of epistemic uncertainty, often linked to the distance of an object's representation to training data, captures only a part of the full epistemic uncertainty. Despite this limitation, their efficiency and effectiveness in out-of-distribution detection have made them widely used. Another class of models for uncertainty quantification that utilizes generative models (e.g., diffusion models or normalizing flows) has been recently explored (Berry & Meger, 2023; Chan et al., 2024). While these methods offer a promising direction, they require training a generative model, which can be computationally intensive and may limit their applicability in certain settings.

Works by Gruber & Buettner (2023); Adlam et al. (2022) are close to ours. In both papers, decompositions of Bregman divergence loss functions are discussed. While the main focus of these papers is generalized bias-variance decomposition, Gruber & Buettner (2023) also considers an application to uncertainty quantification. In terms of our framework, they arrive to the following decomposition (see derivation in Appendix F):

$$\tilde{R}_{\text{Tot}}^{(1,1)} = \tilde{R}_{\text{Bayes}}^{(2)} + \tilde{R}_{\text{Exc}}^{(3,1)} + \tilde{R}_{\text{Exc}}^{(2,3)}, \tag{8}$$

which is only a partial case of our framework. There, they called $\tilde{R}_{\text{Exc}}^{(3,1)}$ as generalized "variance", and use it as a measure of uncertainty for out-of-distribution detection. However, it is not clear, why exactly this decomposition and approximations were chosen. For example, as we showed in Section 4.1, $\tilde{R}_{\text{Bayes}}^{(1)}$ leads to a tighter approximation of Bayes risk on average, than $\tilde{R}_{\text{Bayes}}^{(2)}$.

Works by Kotelevskii et al. (2022a), Lahlou et al. (2023), and Liu et al. (2019) also consider the same decomposition of risks. However, in all these papers, only specific approximations were explored, and no general connection among existing measures was established.

Another recent work by Hofman et al. (2024) independently explored several of the decompositions presented in our study. However, the analysis of central predictions and the connections to Gruber & Buettner (2023) and the energy-based model were not addressed in their work.

Despite the diversity of these approaches, the arbitrary nature of choosing uncertainty measures has led to ambiguity in understanding uncertainty. This paper aims to address this gap by proposing a unified framework that not only categorizes these diverse methods but also offers a more comprehensive understanding of uncertainty quantification.

## 6 EXPERIMENTS

In this section, we test different uncertainty measures derived from our general framework. Any Bayesian method that produces multiple samples of model weights (or parameters of the first-order distribution) can be used to compute our proposed measures. However, deep ensembles are considered the "gold standard" in uncertainty quantification (Lakshminarayanan et al., 2017). Therefore, for our experiments, we use deep ensembles trained with various strictly proper scoring rules as loss functions. As training (in-distribution) datasets, we consider CIFAR10, CIFAR100 (Krizhevsky, 2009), and TinyImageNet (Le & Yang, 2015).

We evaluate the proposed measures of uncertainty by focusing on two specific problems: *out-of-distribution detection* and *misclassification detection*. For both problems, we trained five deep ensembles independently, each consisting of four members—resulting in a total of 20 models trained completely independently. All ensemble members shared the same architecture but differed due to

Table 2: AUROC for different choices of function $G$ for out-of-distribution detection with CIFAR10 as in-distribution. We used 5 groups of ensembles of size 4. We see, that in most cases, uncertainty measures, based on Log Score, appear to be better. See more detailed results in Appendix J.

| | | CIFAR100 | SVHN | TinyImageNet | CIFAR10C-1 | CIFAR10C-2 | CIFAR10C-3 | CIFAR10C-4 | CIFAR10C-5 |
|---|---|---|---|---|---|---|---|---|---|
| Log | $\tilde{R}_{\text{Bayes}}^{(1)}$ | $91.36 \pm 0.05$ | $96.01 \pm 0.39$ | $90.84 \pm 0.04$ | $60.96 \pm 0.09$ | $67.84 \pm 0.09$ | $72.59 \pm 0.08$ | $77.31 \pm 0.08$ | $83.48 \pm 0.13$ |
| | $\tilde{R}_{\text{Exc}}^{(1,1)}$ | $90.17 \pm 0.06$ | $94.08 \pm 1.02$ | $89.32 \pm 0.07$ | $61.07 \pm 0.11$ | $67.82 \pm 0.09$ | $72.49 \pm 0.08$ | $77.15 \pm 0.13$ | $83.0 \pm 0.2$ |
| | $\tilde{R}_{\text{Exc}}^{(1,2)}$ | $90.38 \pm 0.08$ | $94.35 \pm 0.94$ | $89.57 \pm 0.07$ | $61.1 \pm 0.11$ | $67.87 \pm 0.09$ | $72.55 \pm 0.09$ | $77.23 \pm 0.13$ | $83.14 \pm 0.2$ |
| | $\tilde{R}_{\text{Exc}}^{(3,1)}$ | $90.38 \pm 0.06$ | $94.31 \pm 0.91$ | $89.54 \pm 0.06$ | $61.09 \pm 0.11$ | $67.86 \pm 0.09$ | $72.54 \pm 0.09$ | $77.22 \pm 0.13$ | $83.11 \pm 0.19$ |
| Brier | $\tilde{R}_{\text{Bayes}}^{(1)}$ | $90.44 \pm 0.17$ | $96.09 \pm 0.52$ | $89.89 \pm 0.13$ | $61.03 \pm 0.12$ | $68.04 \pm 0.16$ | $72.46 \pm 0.18$ | $76.83 \pm 0.18$ | $82.47 \pm 0.1$ |
| | $\tilde{R}_{\text{Exc}}^{(1,1)}$ | $89.24 \pm 0.19$ | $94.19 \pm 0.49$ | $88.37 \pm 0.12$ | $60.11 \pm 0.15$ | $66.59 \pm 0.24$ | $71.19 \pm 0.3$ | $75.7 \pm 0.22$ | $81.52 \pm 0.16$ |
| | $\tilde{R}_{\text{Exc}}^{(1,2)}$ | $89.24 \pm 0.19$ | $94.19 \pm 0.49$ | $88.37 \pm 0.12$ | $60.11 \pm 0.15$ | $66.59 \pm 0.24$ | $71.19 \pm 0.3$ | $75.7 \pm 0.22$ | $81.52 \pm 0.16$ |
| | $\tilde{R}_{\text{Exc}}^{(3,1)}$ | $89.24 \pm 0.19$ | $94.19 \pm 0.49$ | $88.37 \pm 0.12$ | $60.11 \pm 0.15$ | $66.59 \pm 0.24$ | $71.19 \pm 0.3$ | $75.7 \pm 0.22$ | $81.52 \pm 0.16$ |
| Spherical | $\tilde{R}_{\text{Bayes}}^{(1)}$ | $90.46 \pm 0.04$ | $96.15 \pm 0.4$ | $89.95 \pm 0.19$ | $61.47 \pm 0.19$ | $68.54 \pm 0.23$ | $72.95 \pm 0.28$ | $77.39 \pm 0.34$ | $82.99 \pm 0.43$ |
| | $\tilde{R}_{\text{Exc}}^{(1,1)}$ | $89.11 \pm 0.15$ | $93.77 \pm 0.66$ | $88.28 \pm 0.2$ | $60.54 \pm 0.18$ | $66.88 \pm 0.26$ | $71.36 \pm 0.29$ | $75.96 \pm 0.29$ | $81.89 \pm 0.33$ |
| | $\tilde{R}_{\text{Exc}}^{(1,2)}$ | $89.17 \pm 0.14$ | $93.86 \pm 0.66$ | $88.35 \pm 0.2$ | $60.54 \pm 0.18$ | $66.88 \pm 0.26$ | $71.37 \pm 0.3$ | $75.97 \pm 0.29$ | $81.91 \pm 0.34$ |
| | $\tilde{R}_{\text{Exc}}^{(3,1)}$ | $89.1 \pm 0.15$ | $93.78 \pm 0.66$ | $88.27 \pm 0.2$ | $60.54 \pm 0.18$ | $66.88 \pm 0.26$ | $71.35 \pm 0.29$ | $75.95 \pm 0.29$ | $81.88 \pm 0.33$ |

randomness in initialization and training. We used ResNet18 (He et al., 2016) as the architecture (additional details can be found in Appendix H). Note that in all the tables below, we report the average performance across different groups of ensembles (AUROC), along with the corresponding standard deviation.

Our experimental evaluation[1] aims to answer the following questions:

1. Is there a best function $G$ that yields better results in the considered downstream tasks?

2. Is Excess risk always better than Bayes risk for out-of-distribution detection?

3. Is Total risk always better than Excess risk for misclassification detection?

4. Is the measure of uncertainty proposed by our framework better than other Bayesian estimates of the energy?

We emphasize that the goal of our experimental evaluation is *not to provide new state-of-the-art measures or compete with other known approaches* for uncertainty quantification. Instead, we aim to *verify whether different uncertainty estimates are indeed related to specific types of uncertainty*. We refer readers to Appendix A, where we discuss why one might expect either type of uncertainty to be good for a specific task. Additionally, we refer the reader to Appendix M for a specific toy example where our measures of epistemic uncertainty can be computed in closed form. In this example, we also discuss cases of prior misspecification and model misspecification.

## 6.1 IS THERE A BEST FUNCTION PLUG-IN CHOICE?

In this section, we address the first question. For this, we consider some specific risks and different plug-in choices for the function $G$. As a loss function, we use strictly proper scoring rules generated by the same function $G$ (hence, we do not include the Zero-One score, as it is not differentiable). We present results in Table 2 (see additional results in Tables 5-9 in Appendix). We see that, in most cases, the Log Score performs better than the others. This result is reassuring because Log Score-based measures are the popular choice in the literature on uncertainty quantification. As for misclassification detection (see Table 4 and Tables 10-13 in Appendix), the results are more comparable, but Log Score is still a good choice.

Hence, the answer to the first question is the following: there is no all-time-best plug-in choice of $G$. However, Log Score-based measures (that are already popular in practice) are typically a good choice in the considered downstream tasks.

---

[1]The source code is publicly available at `https://github.com/stat-ml/uncertainty_from_proper_scoring_rules/`.

Table 3: AUROC (Log Score) for OOD detection with TinyImageNet in-distribution.

| | ImageNet-A | ImageNet-R | ImageNet-O |
|---|---|---|---|
| $\tilde{R}_{Bayes}^{(1)}$ | $83.22 \pm 0.24$ | $82.23 \pm 0.39$ | $72.21 \pm 0.22$ |
| $\tilde{R}_{Bayes}^{(2)}$ | $83.76 \pm 0.24$ | $82.78 \pm 0.37$ | $73.18 \pm 0.2$ |
| $\tilde{R}_{Bayes}^{(3)}$ | $83.61 \pm 0.2$ | $82.67 \pm 0.37$ | $72.86 \pm 0.3$ |
| $\tilde{R}_{Exc}^{(1,1)}$ | $77.11 \pm 0.34$ | $76.52 \pm 0.19$ | $74.46 \pm 0.18$ |
| $\tilde{R}_{Exc}^{(1,2)}$ | $79.09 \pm 0.33$ | $78.38 \pm 0.17$ | $74.79 \pm 0.25$ |
| $\tilde{R}_{Exc}^{(2,1)}$ | $75.94 \pm 0.36$ | $75.4 \pm 0.23$ | $74.07 \pm 0.18$ |
| $\tilde{R}_{Exc}^{(1,3)}$ | $76.21 \pm 0.43$ | $75.55 \pm 0.21$ | $73.84 \pm 0.17$ |
| $\tilde{R}_{Exc}^{(3,1)}$ | $77.56 \pm 0.28$ | $77.02 \pm 0.17$ | $74.46 \pm 0.22$ |
| $\tilde{R}_{Tot}^{(1,1)}$ | $84.26 \pm 0.23$ | $83.32 \pm 0.31$ | $74.93 \pm 0.17$ |
| $\tilde{R}_{Tot}^{(1,2)}$ | $83.76 \pm 0.24$ | $82.78 \pm 0.37$ | $73.18 \pm 0.2$ |
| $\tilde{R}_{Tot}^{(1,3)}$ | $83.93 \pm 0.24$ | $82.95 \pm 0.36$ | $73.72 \pm 0.16$ |
| $\tilde{R}_{Tot}^{(3,1)}$ | $83.88 \pm 0.19$ | $82.99 \pm 0.32$ | $74.09 \pm 0.27$ |
| $E(x; \mathbb{E}_\theta f_\theta)$ | $83.96 \pm 0.23$ | $83.28 \pm 0.41$ | $72.72 \pm 0.33$ |
| $\mathbb{E}_\theta E(x; f_\theta)$ | $82.99 \pm 0.26$ | $82.31 \pm 0.45$ | $71.15 \pm 0.34$ |

Table 4: AUROC (Log Score) for misclassification detection. For loss function and uncertainty plug-in, we use the same $G$, that corresponds to Log Score. CIFAR10-N and CIFAR100-N stand for noisy versions of these datasets (see Appendix I).

| | CIFAR10 | CIFAR100 | CIFAR10-N | CIFAR100-N | TinyImageNet |
|---|---|---|---|---|---|
| $\tilde{R}_{Bayes}^{(1)}$ | $94.39 \pm 0.08$ | $84.61 \pm 0.35$ | $78.26 \pm 4.11$ | $81.9 \pm 0.29$ | $84.99 \pm 0.24$ |
| $\tilde{R}_{Bayes}^{(2)}$ | $94.7 \pm 0.05$ | $85.13 \pm 0.35$ | $78.36 \pm 4.77$ | $81.92 \pm 0.36$ | $85.5 \pm 0.23$ |
| $\tilde{R}_{Bayes}^{(3)}$ | $94.76 \pm 0.09$ | $85.95 \pm 0.31$ | $80.44 \pm 3.69$ | $83.41 \pm 0.23$ | $86.55 \pm 0.26$ |
| $\tilde{R}_{Exc}^{(1,1)}$ | $94.1 \pm 0.14$ | $81.58 \pm 0.34$ | $68.39 \pm 6.07$ | $74.52 \pm 0.83$ | $81.33 \pm 0.29$ |
| $\tilde{R}_{Exc}^{(1,2)}$ | $94.23 \pm 0.14$ | $82.9 \pm 0.31$ | $69.64 \pm 6.1$ | $75.74 \pm 0.78$ | $83.22 \pm 0.28$ |
| $\tilde{R}_{Exc}^{(2,1)}$ | $94.01 \pm 0.14$ | $80.69 \pm 0.35$ | $67.7 \pm 6.09$ | $73.7 \pm 0.86$ | $80.18 \pm 0.31$ |
| $\tilde{R}_{Exc}^{(1,3)}$ | $93.72 \pm 0.16$ | $80.3 \pm 0.33$ | $66.85 \pm 5.94$ | $73.37 \pm 0.87$ | $79.25 \pm 0.25$ |
| $\tilde{R}_{Exc}^{(3,1)}$ | $94.4 \pm 0.13$ | $82.62 \pm 0.34$ | $69.98 \pm 6.08$ | $75.51 \pm 0.8$ | $82.74 \pm 0.34$ |
| $\tilde{R}_{Tot}^{(1,1)}$ | $94.5 \pm 0.06$ | $85.43 \pm 0.35$ | $75.73 \pm 4.68$ | $81.67 \pm 0.41$ | $85.58 \pm 0.22$ |
| $\tilde{R}_{Tot}^{(1,2)}$ | $94.7 \pm 0.05$ | $85.13 \pm 0.35$ | $78.36 \pm 4.77$ | $81.92 \pm 0.36$ | $85.5 \pm 0.23$ |
| $\tilde{R}_{Tot}^{(1,3)}$ | $94.4 \pm 0.06$ | $85.01 \pm 0.37$ | $76.26 \pm 5.04$ | $81.67 \pm 0.39$ | $85.18 \pm 0.23$ |
| $\tilde{R}_{Tot}^{(3,1)}$ | $94.74 \pm 0.07$ | $86.22 \pm 0.31$ | $79.38 \pm 3.85$ | $83.18 \pm 0.29$ | $86.65 \pm 0.26$ |

## 6.2 IS EXCESS RISK BETTER THAN BAYES RISK FOR OUT-OF-DISTRIBUTION DETECTION?

In this section, we evaluate different instances of our framework to identify out-of-distribution samples. Since the uncertainty associated with OOD detection is of epistemic nature (see Appendix A for discussion), we expect that Excess and Total risks will perform well for this task, while Bayes risk will likely fail. In the main part, we consider two datasets as in-distribution, namely CIFAR10 in Table 2 and TinyImageNet in Table 3. For more experiments, we refer to Appendix J.

We distinguish between two types of out-of-distribution data: "soft-OOD" and "hard-OOD". Both are special cases of covariate shift. "Soft-OOD" samples, such as slightly changed versions of CIFAR10 or TinyImageNet, have predicted probability vectors that are still meaningful (see discussion in Appendix B). In our experiments, CIFAR10C can be considered as "soft-OOD", as it is a corrupted version of CIFAR10, while ImageNet-O (see description in Appendix I) is "soft-OOD" for TinyImageNet. "Hard-OOD" samples, however, have completely non-informative predicted probability vectors. For example, when a set of classes is considered during training, but an incoming image does not belong to those classes, the resulting probability distribution over training classes is meaningless.

From the Tables 2 and 3, we see that Excess risk is a good choice for "soft-OOD" (especially in case of Log score). However, as data become more "hard-OOD", the results are unexpected – Bayes risk typically outperforms Excess risk. We provide one possible explanation of this effect in Appendix A.

This highlights a **crucial limitation of Excess risk (which includes ubiquitous BI, RBI, and EPBD) as a measure of epistemic uncertainty.** These measures naturally appear when approximating Excess risk in a Bayesian way, which assumes a specific form of ground-truth distribution approximation. However, this approximation becomes inaccurate for "hard-OOD" samples, making these measures a poor choice in these cases. This criticism aligns with findings from (Wimmer et al., 2023; Schweighofer et al., 2023a; Bengs et al., 2023), which indicate that Bregman Information (in a particular case of Log score) is not an intuitive measure of epistemic uncertainty and does not follow their proposed axioms.

Therefore, the answer to the second question is not absolute. For "soft-OOD" samples, where predicted probability vectors remain meaningful, Excess risk is a good choice. For "hard-OOD" samples, Bayes risk is typically better. Total risk consistently shows decent results, making it a safe choice when the nature of incoming data is unknown.

## 6.3 IS TOTAL RISK ALWAYS BETTER THAN EXCESS RISK FOR MISCLASSIFICATION DETECTION?

We now consider misclassification detection. Misclassification detection is intuitively connected to aleatoric and total uncertainties (see Appendix A). Therefore, we expect Bayes and Total risks to

perform well in this task, while all instances of Excess risk should perform typically worse (given the scenario that there is enough training (in-distribution) data).

It is known that standard versions of CIFAR10 and CIFAR100 lack significant aleatoric uncertainty (Kapoor et al., 2022), making it challenging to demonstrate the usefulness of the appropriate uncertainty measures immediately. To address this, we create special versions of these datasets with introduced label noise (see details in Appendix I).

From the Table 4 we see that Bayes and Total risks indeed outperform Excess risk for misclassification detection (see additional results in Appendix K). Moreover, the difference in performance becomes more significant as more aleatoric noise is introduced into the training dataset. Hence, the answer to this question is mostly positive. Bayes risk and Total risk are better for misclassification detection.

### 6.4 WHICH ENERGY ESTIMATE IS BETTER?

We discovered a connection between our approach and energy-based models that are used for out-of-distribution detection. However, in our approach, energy appears as the difference between two different Bayesian energy estimates for a particular choice of $G$. Which one of these three quantities, based on energy, is better?

Based on the results in Table 3 (see also Tables 5, 9,10 and 13 in Appendix) we can conclude, that $E(x; \mathbb{E}_\theta f_\theta)$ is consistently better, than $\mathbb{E}_\theta E(x; f_\theta)$ in both considered problems. Specifically, $\tilde{R}_{\text{Exc}}^{(3,1)}$ performs better than both energy estimates for misclassification detection on all the datasets in Table 10, while for TinyImageNet the results are close (see Table 13 in Appendix). $\tilde{R}_{\text{Exc}}^{(3,1)}$ is also preferred over energy estimates for Soft-OOD detection for ImageNet-O (see Tables 3 and 9) and CIFAR10C[1,2] (see Table 5).

When the out-of-distribution detection problem is considered and "hard-OOD" is encountered, $\tilde{R}_{\text{Exc}}^{(3,1)}$ is typically worse than both of the energy estimates in terms of AUROC. However, when "soft-OOD" is encountered, their difference $\tilde{R}_{\text{Exc}}^{(3,1)}$, as a particular instance of Excess risk, becomes more effective than both of them (see results on ImageNet-O in Tables 3 and 9). When the misclassification detection problem is considered, $\tilde{R}_{\text{Exc}}^{(3,1)}$ is better than both of the estimates.

Hence, the answer is in line with what we discussed above: Excess risk (and in particular $\tilde{R}_{\text{Exc}}^{(3,1)}$) is a good choice for "soft-OOD." Therefore, when this is the case, $\tilde{R}_{\text{Exc}}^{(3,1)}$ performs well. When "hard-OOD" is the case, $E(x; \mathbb{E}_\theta f_\theta)$ is typically better.

## 7 CONCLUSION

In this paper, we developed a general framework for predictive uncertainty quantification using pointwise risk estimates and strictly proper scoring rules as loss functions. We proposed pointwise risk as a natural measure of predictive uncertainty and derived general results for total, epistemic, and aleatoric uncertainties, demonstrating that epistemic uncertainties can be represented as Bregman divergence within this framework.

We incorporated Bayesian reasoning into our framework, showing that commonly used uncertainty measures, such as Mutual Information and Expected Pairwise KL divergence, are special cases within our general approach. We also discussed the limitations of our framework, elaborating on recent critiques in the literature (Wimmer et al., 2023; Schweighofer et al., 2023a). Moreover, we showed that even energy-based models fall into the framework as a particular approximation of Excess risk.

Finally, in our experiments on image datasets, we evaluated these measures for out-of-distribution and misclassification detection tasks and discussed which measures are most suitable for each scenario.

### ACKNOWLEDGEMENTS

This research was partially supported by RSF grant 20-71-10135. Nikita Kotelevskii would like to thank Sergey Kostrov for deriving the central prediction for the spherical score.

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

## A   WHY IS EPISTEMIC UNCERTAINTY GOOD FOR OUT-OF-DISTRIBUTION DETECTION, AND TOTAL FOR MISCLASSIFICATION DETECTION?

Generally speaking, the out-of-distribution detection problem is not directly related to the conditional distribution $p(y \mid x)$, which is at the core of our paper and generally of supervised learning. The formally correct approach would be to consider some approximation of the covariate distribution $p(x)$ and perform out-of-distribution detection based on it.

However, in the literature, both aleatoric uncertainty (related solely to $p(y \mid x)$) and epistemic uncertainty (associated with the quality of estimation for $p(y \mid x)$) are applied to solve the out-of-distribution problem and demonstrate certain performance in the task.

The success of aleatoric uncertainty that we observe in our experiments for "hard-OOD", might seem mysterious as, conceptually, it is not related to the covariate distribution at all. The empirical success of aleatoric uncertainty for detecting out-of-distribution is related to the fact that the models are usually overconfident for in-distribution objects, leading to very low uncertainty. At the same time, the model confidence for out-of-distribution samples is more arbitrary as nothing pushes the model to be confident in them. Thus, out-of-distribution samples may have higher aleatoric uncertainty than in-distribution objects.

At the same time, the application of epistemic uncertainty to out-of-distribution detection is more grounded, as one can expect that we have less knowledge about the actual dependence for out-of-distribution objects than we do for in-distribution objects. This relation can be formalized for some models, such as kernel methods. For example, the work by Kotelevskii et al. (2022a) shows that $p(x)$ epistemic uncertainty is proportional to the inverse of covariate density for Nadaraya-Watson kernel regression. Moreover, this method falls into the same risk decomposition, but is built on another approximation.

For the Bayesian approximations we considered (except constant approximations), Excess risk can be seen as a "measure of disagreement" between predictions of the members of the ensemble. One may expect ensemble members to have arbitrarily different and diverse predictions for out-of-distribution data, resulting in higher values of Excess risk. Lots of literature on the topic has the same flavor of reasoning. For example, uncertainty papers about Mutual Information, EPKL (Malinin & Gales, 2021; Lakshminarayanan et al., 2017; Gal et al., 2017; Schweighofer et al., 2023a), etc., are the particular case of our framework.

Similarly, total and aleatoric uncertainty measures are effective for misclassification detection because they capture the inherent ambiguity or noise in the data associated with specific inputs. Aleatoric uncertainty reflects the variability in the model's predictions due to the inherent randomness or non-deterministic dependency between covariates and labels, which is a common source of misclassification. Therefore, when a model encounters an input that is difficult to map to a certain class, the aleatoric uncertainty increases.

Measures of epistemic uncertainty effectively capture the disagreement between different members or samples of a Bayesian model. Hence, if there is a non-deterministic dependency between covariates and labels, epistemic uncertainty might increase as well, as different models may produce varying predictions for the same input. Therefore, the sum of both uncertainties – namely, total uncertainty – should be the most effective measure for detecting misclassification events.

## B   LIMITATIONS

We see two limitations to our approach.

**Valid conditional** $\eta(x) = p(y \mid x)$ **for all** $x$**.**   This assumption implies, that regardless of the input $x$, the form of the probability distribution $\eta(x)$ will not change. This means, that even for inputs, that do not belong to $P_{tr}(X)$, the conditional should produce some categorical vector over the same number of classes. Let us consider an example of a binary classification problem, where we want our model to distinguish between cats and dogs (see Figure 1). In this case, the distribution of covariates $P_{tr}(X)$ is the distribution over images of all possible cats and dogs. An image of a pigeon under this distribution should have a negligible probability. Now imagine, that somehow it happened

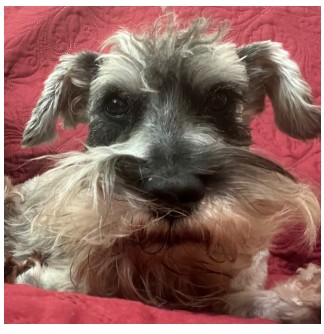 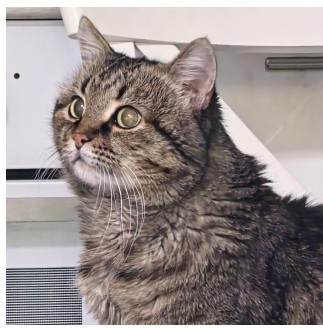 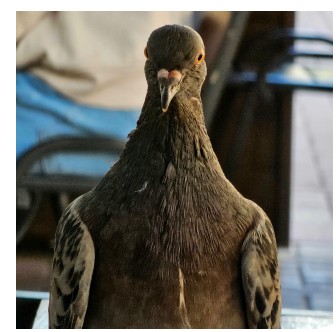

A dog, has sufficient probability under $P_{tr}$. Conditional $\eta(x)$ is **meaningful**.

A cat, has sufficient probability under $P_{tr}$. Conditional $\eta(x)$ is **meaningful**.

A pigeon, has almost no probability mass under $P_{tr}$. Conditional $\eta(x)$ is **vague**.

Figure 1: The figure shows different examples of input objects in binary classification problem (cats vs dogs). The limitation of our approach is that $\eta(x) = P_{tr}(Y \mid X = x)$ should be defined even for objects with tiny mass under $P_{tr}$ (see discussion in Appendix B).

that $x$ is actually an image of a pigeon. Under our assumption, $\eta(x)$ should be valid, so it should produce a vector of probabilities over two classes: Cats and Dogs, despite an input being an apparent out-of-distribution object. There is no good way to define $\eta(x)$ for such input objects, hence we say it is vague. However, for unusual (but still in-distribution) inputs, like rare dog breeds, $\eta(x)$ is meaningful.

**Incorporation of Bayesian reasoning for estimation of $\eta$.** In practice, we do not have access to $\eta(x)$. Hence, we suggested approximating it using the Bayesian approach and proposed two ideas to do it (inner and outer expectations). For Bayes risk the best Bayesian estimate is given by outer expectation (see Appendix E). However, this is not the case for Excess risk. It appears (see discussion in Appendix E) that Excess risk depends on the estimate of the Total risk. But we never know in advance for a particular input $x$, in which regime (overestimated or underestimated Total risk) we are. Thus, we do not know what is the best choice for an approximation to epistemic uncertainty.

## C  DERIVATIONS OF DIFFERENT RISKS WITH PROPER SCORING RULES

We will start with the derivation of Total risk. In what follows, we will omit dependency on $x$ for $\eta(x)$ and $\eta_\theta(x)$.

$$R_{\text{Tot}}(\eta_\theta \mid x) = \int \ell(\eta_\theta, y) dP(y \mid x) = \sum_{k=1}^{K} \Big( \langle G'(\eta_\theta) , \eta_\theta \rangle - G'_k(\eta_\theta) - G(\eta_\theta) \Big) \eta_k =$$
$$\langle G'(\eta_\theta) , \eta_\theta \rangle - G(\eta_\theta) - \langle G'(\eta_\theta) , \eta \rangle.$$

Let us now consider Bayes risk:

$$R_{\text{Bayes}}(x) = \int \ell(\eta, y) dP(y \mid x) = \sum_{k=1}^{K} \Big( \langle G'(\eta) , \eta \rangle - G'_k(\eta) - G(\eta) \Big) \eta_k =$$
$$\langle G'(\eta) , \eta \rangle - \langle G'(\eta) , \eta \rangle - G(\eta) = -G(\eta).$$

Finally, let us consider Excess risk:

$$R_{\text{Exc}}(\eta_\theta \mid x) = \underbrace{\int \ell(\eta_\theta, y) dP(y \mid x)}_{\text{Total risk}} - \underbrace{\int \ell(\eta, y) dP(y \mid x)}_{\text{Bayes risk}} =$$
$$\langle G'(\eta_\theta) , \eta_\theta \rangle - G(\eta_\theta) - \langle G'(\eta_\theta) , \eta \rangle + G(\eta) =$$
$$G(\eta) - G(\eta_\theta) - \langle G'(\eta_\theta) , \eta - \eta_\theta \rangle := D_G\Big(\eta \| \eta_\theta\Big).$$

# D  DERIVATION OF RISKS FOR SPECIFIC CHOICES OF SCORING RULES

In this section, we will derive specific equations for Total, Bayes, and Excess pointwise risks to get the estimates of total, aleatoric, and epistemic uncertainties correspondingly. We will omit subscript $\theta$ in this section, indicating an estimate by using a hat.

Recall equations for proper scoring rule and different risks:

$$\ell(\eta, i) = \langle G'(\eta) , \eta \rangle - G_i'(\eta) - G(\eta),$$
$$R_{\text{Tot}} = \langle G'(\hat{\eta}) , \hat{\eta} \rangle - G(\hat{\eta}) - \langle G'(\hat{\eta}) , \eta \rangle,$$
$$R_{\text{Bayes}} = -G(\eta),$$
$$R_{\text{Exc}} = G(\eta) - G(\hat{\eta}) + \langle G'(\hat{\eta}) , \hat{\eta} - \eta \rangle.$$

## D.1  LOG SCORE (CROSS-ENTROPY)

$$G(\eta) = \sum_{k=1}^{K} \eta_k \log \eta_k,$$
$$G'(\eta)_k = 1 + \log \eta_k,$$

$$\ell(\eta, i) = \langle 1 + \log \eta , \eta \rangle - 1 - \log \eta_i - \sum_{k=1}^{K} \eta_k \log \eta_k =$$

$$\sum_{k=1}^{K} \eta_k \log \eta_k + 1 - 1 - \log \eta_i - \sum_{k=1}^{K} \eta_k \log \eta_k = - \log \eta_i,$$

$$R_{\text{Tot}} = \sum_{k=1}^{K} \Big( (1 + \log \hat{\eta}_k)\hat{\eta}_k - \hat{\eta}_k \log \hat{\eta}_k - (1 + \log \hat{\eta}_k)\eta_k \Big) =$$

$$\sum_{k=1}^{K} \Big( \hat{\eta}_k \log \hat{\eta}_k - \hat{\eta}_k \log \hat{\eta}_k - \eta_k \log \hat{\eta}_k \Big) = \text{CE}\Big[\eta \| \hat{\eta}\Big],$$

$$R_{\text{Bayes}} = - \sum_{k=1}^{K} \eta_k \log \eta_k = \mathbb{H}\eta,$$

$$R_{\text{Exc}} = R_{\text{Tot}} - R_{\text{Bayes}} = \text{CE}\Big[\eta \| \hat{\eta}\Big] - \mathbb{H}\eta = \text{KL}\Big[\eta \| \hat{\eta}\Big].$$

## D.2  QUADRATIC SCORE (BRIER SCORE)

$$G(\eta) = - \sum_{k=1}^{K} \eta_k(1 - \eta_k),$$
$$G'(\eta)_k = 2\eta_k - 1,$$

$$\ell(\eta, i) = \langle 2\eta - 1 , \eta \rangle - 2\eta_i + 1 + \sum_{k=1}^{K} \eta_k(1 - \eta_k) =$$

$$2 \sum_{k=1}^{K} \eta_k^2 - 1 - 2\eta_i + 1 + 1 - \sum_{k=1}^{K} \eta_k^2 = \sum_{k=1}^{K} \eta_k^2 - 2\eta_i + 1,$$

since constant does not affect optimization, we will use the following:

$$\ell(\eta, i) = \sum_{k=1}^{K} \eta_k^2 - 2\eta_i.$$

$$\mathrm{R_{Tot}} = \sum_{k=1}^{K}(2\hat{\eta}_k - 1)\hat{\eta}_k + \sum_{k=1}^{K}\hat{\eta}_k(1 - \hat{\eta}_k) - \sum_{k=1}^{K}(2\hat{\eta}_k - 1)\eta_k =$$

$$\sum_{k=1}^{K}(\hat{\eta}_k^2 - 2\hat{\eta}_k\eta_k + \eta_k^2 - \eta_k^2) + 1 = \|\hat{\eta} - \eta\|_2^2 - \|\eta\|_2^2 + 1,$$

$$\mathrm{R_{Bayes}} = \sum_{k=1}^{K}\eta_k(1 - \eta_k) = 1 - \|\eta_k\|_2^2,$$

$$\mathrm{R_{Exc}} = \mathrm{R_{Tot}} - \mathrm{R_{Bayes}} = \|\hat{\eta} - \eta\|_2^2 - \|\eta\|_2^2 + 1 - 1 + \|\eta_k\|_2^2 = \|\hat{\eta} - \eta\|_2^2.$$

### D.3 ZERO-ONE SCORE

$$G(\eta) = \max_k \eta_k - 1,$$

$$G'(\eta)_k = \mathbb{I}[k = \arg\max_j \eta_j],$$

$$\ell(\eta, i) = \langle \mathbb{I}[k = \arg\max_j \eta_j] , \eta\rangle - \mathbb{I}[i = \arg\max_j \eta_j] - \max_k \eta_k + 1 =$$

$$\max_k \eta_k - \mathbb{I}[i = \arg\max_j \eta_j] - \max_k \eta_k + 1 = 1 - \mathbb{I}[i = \arg\max_j \eta_j] = \mathbb{I}[i \neq \arg\max_j \eta_j],$$

$$\mathrm{R_{Tot}} = \sum_{k=1}^{K}\Big(\hat{\eta}_k\mathbb{I}[k = \arg\max_j \hat{\eta}_k] - \eta_k\mathbb{I}[k = \arg\max_j \hat{\eta}_k]\Big) - \max_k \hat{\eta}_k + 1 =$$

$$\max_k \hat{\eta}_k - \eta_{\arg\max_j \hat{\eta}_k} - \max_k \hat{\eta}_k + 1 = 1 - \eta_{\arg\max_j \hat{\eta}_k},$$

$$\mathrm{R_{Bayes}} = 1 - \max_k \eta_k,$$

$$\mathrm{R_{Exc}} = \mathrm{R_{Tot}} - \mathrm{R_{Bayes}} = 1 - \eta_{\arg\max_j \hat{\eta}_k} - 1 + \max_k \eta_k = \eta_{\arg\max_j \eta_k} - \eta_{\arg\max_j \hat{\eta}_k}.$$

### D.4 SPHERICAL SCORE

$$G(\eta) = \|\eta\|_2 - 1,$$

$$G'(\eta)_k = \frac{\eta_k}{\|\eta\|_2},$$

$$\ell(\eta, i) = \langle \frac{\eta}{\|\eta\|_2} , \eta\rangle - \frac{\eta_i}{\|\eta\|_2} - \|\eta\|_2 + 1 = \|\eta\|_2 - \frac{\eta_i}{\|\eta\|_2} - \|\eta\|_2 + 1 = 1 - \frac{\eta_i}{\|\eta\|_2},$$

$$\mathrm{R_{Tot}} = \sum_{k=1}^{K}\Big(\frac{\hat{\eta}_k\hat{\eta}_k}{\|\hat{\eta}\|_2} - \frac{\eta_k\hat{\eta}_k}{\|\hat{\eta}\|_2}\Big) - \|\hat{\eta}\|_2 + 1 = 1 - \sum_{k=1}^{K}\frac{\eta_k\hat{\eta}_k}{\|\hat{\eta}\|_2} = 1 - \|\eta\|_2\langle\frac{\eta}{\|\eta\|_2} , \frac{\hat{\eta}}{\|\hat{\eta}\|_2}\rangle,$$

$$\mathrm{R_{Bayes}} = 1 - \|\eta\|_2,$$

$$\mathrm{R_{Exc}} = \mathrm{R_{Tot}} - \mathrm{R_{Bayes}} = 1 - \|\eta\|_2\langle\frac{\eta}{\|\eta\|_2} , \frac{\hat{\eta}}{\|\hat{\eta}\|_2}\rangle + \|\eta\|_2 - 1 = \|\eta\|_2\Big(1 - \langle\frac{\eta}{\|\eta\|_2} , \frac{\hat{\eta}}{\|\hat{\eta}\|_2}\rangle\Big).$$

## D.5 Negative log score

$$G(\eta) = -\sum_{k=1}^{K} \log \eta_k,$$

$$G'(\eta)_k = -\frac{1}{\eta_k},$$

$$\ell(\eta, i) = \langle -\frac{1}{\eta}, \eta \rangle + \frac{1}{\eta_i} + \sum_{k=1}^{K} \log \eta_k = -K + \frac{1}{\eta_i} + \sum_{k=1}^{K} \log \eta_k,$$

since constant does not affect optimization, we will have:

$$\ell(\eta, i) = \frac{1}{\eta_k} + \sum_{k=1}^{K} \log \eta_k,$$

$$R_{\text{Tot}} = \sum_{k=1}^{K} \left( -\frac{\hat{\eta}_k}{\hat{\eta}_k} + \frac{\eta_k}{\hat{\eta}_k} + \log \hat{\eta}_k \right) = \sum_{k=1}^{K} \left( \frac{\eta_k}{\hat{\eta}_k} + \log \hat{\eta}_k - 1 \right),$$

$$R_{\text{Bayes}} = \sum_{k=1}^{K} \log \eta_k,$$

$$R_{\text{Exc}} = R_{\text{Tot}} - R_{\text{Bayes}} = \sum_{k=1}^{K} \left( \frac{\eta_k}{\hat{\eta}_k} + \log \hat{\eta}_k - 1 - \log \eta_k \right) = \sum_{k=1}^{K} \left( \frac{\eta_k}{\hat{\eta}_k} - \log \frac{\eta_k}{\hat{\eta}_k} - 1 \right) = D_{\text{IS}}[\eta \| \hat{\eta}].$$

## E Is there the best approximation?

We discussed the choice of the approximation for Bayes risk in the main part of the paper. Here, we discuss whether there is a best choice of Excess risk. If we know it, we can choose the best Total risk.

Recall that for Excess risk there are 9 possible options:

- **Expected Pairwise Bregman Divergence (EPBD):**

$$\tilde{R}_{\text{Exc}}^{(1,1)}(x) = \mathbb{E}_{p(\tilde{\theta}|D_{tr})} \mathbb{E}_{p(\theta|D_{tr})} D_G\left(\eta_{\tilde{\theta}} \| \eta_\theta\right).$$

  Note, that since KL divergence is a special case of Bregman divergence, Expected Pairwise KL (EPKL (Malinin & Gales, 2021; Schweighofer et al., 2023a)) is one of the special cases of this Excess risk estimate.

- **Bregman Information (BI):**

$$\tilde{R}_{\text{Exc}}^{(1,2)}(x) = \mathbb{E}_{p(\tilde{\theta}|D_{tr})} D_G\left(\eta_{\tilde{\theta}} \| \hat{\eta}_{D_{tr}}\right),$$

  which special case is BALD (Gal et al., 2017; Houlsby et al., 2011).

- **Reverse Bregman Information (RBI):**

$$\tilde{R}_{\text{Exc}}^{(2,1)}(x) = \mathbb{E}_{p(\theta|D_{tr})} D_G\left(\hat{\eta}_{D_{tr}} \| \eta_\theta\right).$$

  Its special case for Log score is known as **Reverse Mutual Information** (Malinin & Gales, 2021).

- **Modified Bregman Information (MBI):**

$$\tilde{R}_{\text{Exc}}^{(1,3)}(x) = \mathbb{E}_{p(\theta|D_{tr})} D_G\left(\eta_\theta \| \bar{\eta}\right).$$

  It is similar to Bregman Information, but the deviation is computed from the "central prediction", not central label (BMA).

- **Modified Reverse Bregman Information (MRBI)**:

$$\tilde{R}_{Exc}^{(3,1)}(x) = \mathbb{E}_{p(\theta|D_{tr})} D_G\Big(\bar{\eta} \parallel \eta_\theta\Big).$$

This has similar structure to Reverse Bregman Information (RBI). Again, the deviation here is computed from another "central" prediction.

- **Forward Bias term**:

$$\tilde{R}_{Exc}^{(2,3)}(x) = D_G\Big(\hat{\eta}_{D_{tr}} \parallel \bar{\eta}\Big).$$

It represents a Bregman divergence between two different Bayesian estimates and can be viewed as a bias.

- **Reverse Bias term**:

$$\tilde{R}_{Exc}^{(3,2)}(x) = D_G\Big(\bar{\eta} \parallel \hat{\eta}_{D_{tr}}\Big).$$

This applies similar reasoning to the Forward Bias Term.

- Finally, we obtain two similar constant measures:

$$\tilde{R}_{Exc}^{(3,3)}(x) = D_G\Big(\bar{\eta} \parallel \bar{\eta}\Big) = 0,$$

and

$$\tilde{R}_{Exc}^{(2,2)}(x) = D_G\Big(\hat{\eta}_{D_{tr}} \parallel \hat{\eta}_{D_{tr}}\Big) = 0,$$

which is coherent with the result obtained for the Total risk, when Excess risk (epistemic uncertainty) is equal to 0.

However, it is not clear, what estimate of the Excess risk we should use. Indeed, neither of these estimates are upper nor lower bounds for the true Excess risk. This is because contrary to Bayes risk, we don't have any idea if Excess risk with ground truth $\eta$ reaches any extreme. For another explanation, see Figure 2 and discussion in Appendix B. For simplicity, we will consider only approximations with strategies (1) and (2), as for them we know, at least, which estimate of Bayes risk is better.

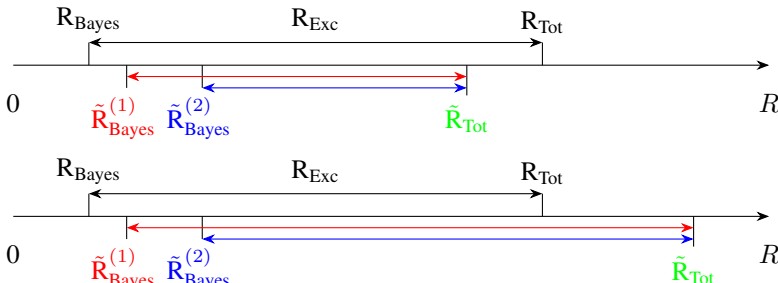

Figure 2: Different situations for risk estimates. Risks typed in black and above the axis are the true ones. Risks, typed in color, and below are estimates. Two-pointed arrows show Excess risks.

**Top.** $\tilde{R}_{Tot}$ underestimates $R_{Tot}$, $\tilde{R}_{Bayes}^{(1)}$ better estimates $R_{Bayes}$, and $\tilde{R}_{Exc}^{(1)}$ better estimates $R_{Exc}$.

**Bottom.** $\tilde{R}_{Tot}$ overestimates $R_{Tot}$, $\tilde{R}_{Bayes}^{(1)}$ better estimates $R_{Bayes}$, and $\tilde{R}_{Exc}^{(2)}$ better estimates $R_{Exc}$. We see, that for different estimates of $R_{Tot}$, we have different best approximations for $R_{Exc}$. See discussion in Appendix B.

In Figure 2, for simplicity, we consider only the Bayesian approximation of the first argument (ground-truth probability). In black, we have actual (real risks), while in color we have different estimates of risks. Also, as two-sided arrows, we show the Excess risk.

If we *underestimate* the Total risk (see top plot in Figure 2), the best choice for Excess risk will be $\tilde{R}_{Exc}^{(1)}$, as despite being a lower bound on Excess risk, it is the best we can do ($\tilde{R}_{Exc}^{(2)}$ in this case will be even worse). However, if we *overestimate* Total risk, then there is no single best choice. In the

bottom plot, when $\tilde{R}_{\text{Tot}}$ significantly overestimates $R_{\text{Tot}}$, the second idea for estimating Excess risk gives a better estimate, despite the first idea for Bayes risk still better.

Hence, the best estimate of Excess risk depends on how well we estimate Total risk. But we never know in advance for a particular input $x$, in which regime (overestimated or underestimated Total risk) we are. Thus, there is no best choice among these risks to approximate epistemic uncertainty.

## F    RELATIONS BETWEEN THE ESTIMATES

In this section, we discuss how the measures of uncertainty are connected. In the main text, we discussed several ways how one can estimate risk given an ensemble of models, posterior, or samples from it. In what follows, we show how one can further decompose these estimates of Excess risk.

Let us start with $\tilde{R}_{\text{Exc}}^{(1,1)}(x)$. Using results of (Pfau, 2013), we have:

$$\tilde{R}_{\text{Exc}}^{(1,1)}(x) = \mathbb{E}_{p(\theta|D_{tr})}\mathbb{E}_{p(\tilde{\theta}|D_{tr})}D_G\Big(\eta_{\tilde{\theta}}\|\eta_\theta\Big) =$$
$$\mathbb{E}_{p(\theta|D_{tr})}D_G\Big(\hat{\eta}_{D_{tr}}\|\eta_\theta\Big) + \mathbb{E}_{p(\tilde{\theta}|D_{tr})}D_G\Big(\eta_{\tilde{\theta}}\|\hat{\eta}_{D_{tr}}\Big) = \tilde{R}_{\text{Exc}}^{(2,1)}(x) + \tilde{R}_{\text{Exc}}^{(1,2)}(x).$$

Since all of these estimates are non-negative, the following holds true:
$$\tilde{R}_{\text{Exc}}^{(1,1)}(x) \geq \tilde{R}_{\text{Exc}}^{(2,1)}(x) \geq \tilde{R}_{\text{Exc}}^{(2,2)}(x) = 0,$$
and
$$\tilde{R}_{\text{Exc}}^{(1,1)}(x) \geq \tilde{R}_{\text{Exc}}^{(1,2)}(x) \geq \tilde{R}_{\text{Exc}}^{(2,2)}(x) = 0,$$
for any $x$.

Moreover, one can show that the following holds:
$$\tilde{R}_{\text{Tot}}^{(1,1)}(x) = \tilde{R}_{\text{Tot}}^{(2,1)}(x) = \tilde{R}_{\text{Bayes}}^{(1)}(x) + \tilde{R}_{\text{Exc}}^{(1,1)}(x) = \tilde{R}_{\text{Bayes}}^{(2)}(x) + \tilde{R}_{\text{Exc}}^{(2,1)}(x),$$
$$\tilde{R}_{\text{Tot}}^{(1,2)}(x) = \tilde{R}_{\text{Tot}}^{(2,2)}(x) = \tilde{R}_{\text{Bayes}}^{(2)}(x) + \tilde{R}_{\text{Exc}}^{(2,2)}(x) = \tilde{R}_{\text{Bayes}}^{(1)}(x) + \tilde{R}_{\text{Exc}}^{(1,2)}(x),$$

$$\tilde{R}_{\text{Tot}}^{(3,3)}(x) = \tilde{R}_{\text{Bayes}}^{(3)}(x) + \tilde{R}_{\text{Exc}}^{(3,3)}(x) = \tilde{R}_{\text{Bayes}}^{(3)}(x) = \tilde{R}_{\text{Tot}}^{(3,1)}(x) - \tilde{R}_{\text{Exc}}^{(3,1)}(x),$$

$$\tilde{R}_{\text{Tot}}^{(1,3)}(x) = \tilde{R}_{\text{Tot}}^{(2,3)}(x) = \tilde{R}_{\text{Tot}}^{(2,1)}(x) - \tilde{R}_{\text{Exc}}^{(3,1)}(x).$$

From Pfau (2013) the following holds:
$$\tilde{R}_{\text{Exc}}^{(2,1)}(x) = \tilde{R}_{\text{Exc}}^{(2,3)}(x) + \tilde{R}_{\text{Exc}}^{(3,1)}. \tag{9}$$

Additionally, Bregman information can be received as follows:
$$BI(x) = \tilde{R}_{\text{Exc}}^{(1,1)}(x) - \tilde{R}_{\text{Exc}}^{(2,1)}(x) = \tilde{R}_{\text{Exc}}^{(1,2)}(x) - \tilde{R}_{\text{Exc}}^{(2,2)}(x) = \tilde{R}_{\text{Bayes}}^{(2)}(x) - \tilde{R}_{\text{Bayes}}^{(1)}(x). \tag{10}$$

Reverse Bregman Information:
$$RBI(x) = \tilde{R}_{\text{Exc}}^{(2,1)}(x) - \tilde{R}_{\text{Exc}}^{(2,2)}(x) = \tilde{R}_{\text{Exc}}^{(1,1)}(x) - \tilde{R}_{\text{Exc}}^{(1,2)}(x) = \tilde{R}_{\text{Tot}}^{(1,1)}(x) - \tilde{R}_{\text{Tot}}^{(1,2)}(x).$$

Interestingly, the EPBD can be written in two equivalent forms:
$$\tilde{R}_{\text{Exc}}^{(1,1)}(x) = \tilde{R}_{\text{Exc}}^{(1,2)}(x) + \tilde{R}_{\text{Exc}}^{(2,1)}(x) = \tilde{R}_{\text{Exc}}^{(1,3)}(x) + \tilde{R}_{\text{Exc}}^{(3,1)}(x). \tag{11}$$

An interesting observation from the (11) is that in general there are two central points: central label and central prediction—where the sum of expected deviations in terms of Bregman divergence lead to the same result, known as EPBD. To the best of our knowledge, this is a novel finding.

Now we are fully equipped to recover the decomposition, used in Gruber & Buettner (2023) (Equation (8)), which is naturally appears, when using $\tilde{R}_{\text{Tot}}^{(1,1)}$ from the above, as well as Equation (9) and Equation (10).

## G  CONNECTION TO ENERGY-BASED MODELS

Here, we will consider a specific case of the framework, instantiated for Logscore. In what follows, we will sometimes omit explicit dependency on $x$ for better presentation. However, this dependency is always assumed.

Recap, that for the special case of Logscore, Bregman divergence is Kullback-Leibler divergence. Hence, $\tilde{R}_{Exc}^{(3,1)} = \mathbb{E}_\theta KL(\bar{\eta} \mid \eta_\theta)$. Using results from Table 1, we can derive that:

$$\log \bar{\eta}_i = \left[\mathbb{E}_\theta \log \eta_\theta\right]_i - \log \sum_j \exp\left[\mathbb{E}_\theta \log \eta_\theta\right]_j.$$

Moreover $\frac{f_\theta(x)}{T} = \text{Softmax}^{-1}(\eta_\theta(x))$, where $T$ is temperature, that scales logits $f_\theta(x)$.

Hence, the logarithm of a probability vector can be further expanded:

$$\left[\log \eta_\theta\right]_i = \log \frac{\left[\exp \frac{f_\theta}{T}\right]_i}{\sum_j \left[\exp \frac{f_\theta}{T}\right]_j} = \left[\frac{f_\theta}{T}\right]_i - \log \sum_j \exp\left[\frac{f_\theta}{T}\right]_j.$$

From these equations, we can derive:

$$\tilde{R}_{Exc}^{(3,1)} = \mathbb{E}_\theta KL(\bar{\eta} \mid \eta_\theta) = \mathbb{E}_\theta \sum_i \bar{\eta}_i \left[\log \frac{\bar{\eta}}{\eta_\theta}\right]_i = \sum_i \bar{\eta}_i \log \bar{\eta}_i - \sum_i \bar{\eta}_i \left[\mathbb{E}_\theta \log \eta_\theta\right]_i =$$

$$\sum_i \bar{\eta}_i \left[\mathbb{E}_\theta \log \eta_\theta\right]_i - \sum_i \bar{\eta}_i \log \sum_j \exp\left[\mathbb{E}_\theta \log \eta_\theta\right]_j - \sum_i \bar{\eta}_i \left[\mathbb{E}_\theta \log \eta_\theta\right]_i =$$

$$- \log \sum_j \exp\left[\mathbb{E}_\theta \log \eta_\theta\right]_j.$$

Furthermore, one may show that:

$$\tilde{R}_{Exc}^{(3,1)} = - \log \sum_i \exp\left[\mathbb{E}_\theta \log \eta_\theta\right]_i = - \log \sum_i \exp\left[\left[\frac{\mathbb{E}_\theta f_\theta}{T}\right]_i - \mathbb{E}_\theta \log \sum_j \exp\left[\frac{f_\theta}{T}\right]_j\right] =$$

$$- \log \frac{\sum_i \exp\left[\frac{\mathbb{E}_\theta f_\theta}{T}\right]_i}{\exp \mathbb{E}_\theta \log \sum_j \exp\left[\frac{f_\theta}{T}\right]_j} = - \log \sum_i \exp\left[\frac{\mathbb{E}_\theta f_\theta}{T}\right]_i + \mathbb{E}_\theta \log \sum_j \exp\left[\frac{f_\theta}{T}\right]_j =$$

$$\frac{1}{T}\left(-T \log \sum_i \exp\left[\frac{\mathbb{E}_\theta f_\theta}{T}\right]_i - \left[-T\mathbb{E}_\theta \log \sum_j \exp\left[\frac{f_\theta}{T}\right]_j\right]\right).$$

Hence:

$$T\tilde{R}_{Exc}^{(3,1)}(x) = \underbrace{-T \log \sum_i \exp\left[\frac{\mathbb{E}_\theta f_\theta}{T}\right]_i}_{E(x;\mathbb{E}_\theta f_\theta)} - \underbrace{\left(-T\mathbb{E}_\theta \log \sum_j \exp\left[\frac{f_\theta}{T}\right]_j\right)}_{\mathbb{E}_\theta E(x;f_\theta)}.$$

Therefore,

$$\tilde{R}_{Exc}^{(3,1)}(x) = \frac{1}{T}\left(E(x; \mathbb{E}_\theta f_\theta) - \mathbb{E}_\theta E(x; f_\theta)\right).$$

Next, we can consider another approximation, namely $\tilde{R}_{\text{Exc}}^{(1,3)}$:

$$\tilde{R}_{\text{Exc}}^{(1,3)} = \mathbb{E}_\theta \text{KL}(\eta_\theta \mid \bar{\eta}) = \sum_i \mathbb{E}_\theta \Big[\eta_\theta \log \eta_\theta\Big]_i - \sum_i \Big[\mathbb{E}_\theta \eta_\theta \log \bar{\eta}\Big]_i =$$

$$\sum_i \mathbb{E}_\theta \Big[\eta_\theta \frac{f_\theta}{T}\Big]_i - \mathbb{E}_\theta \log \sum_j \exp\Big[\frac{f_\theta}{T}\Big]_j - \sum_i \Big[\mathbb{E}_\theta \eta_\theta \mathbb{E}_\theta \log \eta_\theta\Big]_i + \log \sum_j \exp\Big[\mathbb{E}_\theta \log \eta_\theta\Big]_j =$$

$$\sum_i \mathbb{E}_\theta \Big[\eta_\theta \frac{f_\theta}{T}\Big]_i - \mathbb{E}_\theta \log \sum_j \exp\Big[\frac{f_\theta}{T}\Big]_j -$$

$$- \sum_i \Big[\mathbb{E}_\theta \eta_\theta \mathbb{E}_\theta \Big[\big[\frac{f_\theta}{T}\big]_i - \log \sum_j \exp\Big[\frac{f_\theta}{T}\Big]_j\Big]\Big] + \log \sum_j \exp\Big[\mathbb{E}_\theta \log \eta_\theta\Big]_j =$$

$$\sum_i \mathbb{E}_\theta \Big[\eta_\theta \frac{f_\theta}{T}\Big]_i - \mathbb{E}_\theta \log \sum_j \exp\Big[\frac{f_\theta}{T}\Big]_j - \sum_i \mathbb{E}_\theta \eta_\theta \mathbb{E}_\theta \Big[\frac{f_\theta}{T}\Big]_i + \mathbb{E}_\theta \log \sum_j \exp\Big[\frac{f_\theta}{T}\Big]_j +$$

$$+ \log \sum_j \exp\Big[\mathbb{E}_\theta \log \eta_\theta\Big]_j = \sum_i \text{cov}\Big[\big[\eta_\theta\big]_i, \big[\frac{f_\theta}{T}\big]_i\Big] + \log \sum_j \exp\Big[\mathbb{E}_\theta \log \eta_\theta\Big]_j =$$

$$\sum_i \text{cov}\Big[\big[\eta_\theta\big]_i, \big[\frac{f_\theta}{T}\big]_i\Big] - \tilde{R}_{\text{Exc}}^{(3,1)}.$$

Therefore,

$$\tilde{R}_{\text{Exc}}^{(1,3)} + \tilde{R}_{\text{Exc}}^{(3,1)} = \tilde{R}_{\text{Exc}}^{(1,1)} = \sum_i \text{cov}\Big[\big[\eta_\theta\big]_i, \big[\frac{f_\theta}{T}\big]_i\Big].$$

Therefore, Expected Pairwise Kullback-Leibler divergence (EPBD in the partial case of Logscore) is equal to the sum of covariances, between predicted probability of a class, and the corresponding scaled (tempered) logit. To our best knowledge, it is the first interpretation of the result.

## H  TRAINING DETAILS

Training procedures for each dataset were similar. We used ResNet18 architecture. All the networks in the ensemble were trained entirely independently, each starting from a different random initialization of weights. They did not share any parameters.

For CIFAR10-based datasets, we used code from this repository: `https://github.com/kuangliu/pytorch-cifar`. The training procedure consisted of 200 epochs with a cosine annealing learning rate. For an optimizer, we use SGD with momentum and weight decay. For more details see the code.

In Figure 3 (left) we present performance summary statistics of the ensembles. Specifically, we show accuracy, macro averaged precision, recall, and F1-score.

For CIFAR100-based datasets, we used code from this repository: `https://github.com/weiaicunzai/pytorch-cifar100`. The training procedure consisted of 200 epochs with learning rate decay at particular milestones: [60, 120, 160]. For an optimizer, we use SGD with momentum and weight decay. For more details see the code.

Similarly to CIFAR10, in Figure 3 (middle) we present performance summary statistics of the ensembles of ResNet18 architecture.

For TinyImageNet, we used pre-trained models, provided by Torch-Uncertainty team https://github.com/ENSTA-U2IS-AI/torch-uncertainty. They are provided with a single training loss function. Training statistics for this dataset are in Figure 3 (right).

Please note, that for all these datasets, we used different loss functions during training and various instantiations of $G$ for the uncertainty quantification measures. The specific configurations are specified in the captions of the result tables.

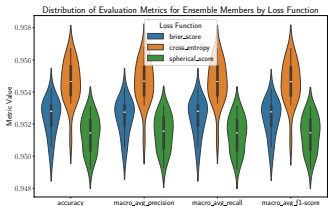 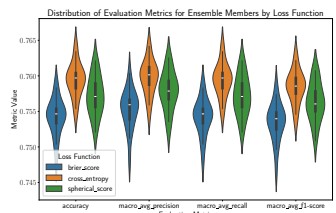 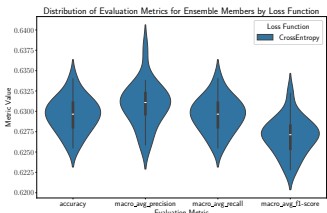

Figure 3: Violin plots for different training loss functions and different metrics for ResNet18 Left: CIFAR10; Middle: CIFAR100; Right: TinyImageNet.

# I DESCRIPTION OF DATASETS

## I.1 NOISY DATASETS

In this section, we describe the noisy versions of CIFAR10 and CIFAR100 datasets created for our experiments, namely CIFAR10-N and CIFAR100-N (N stands for "noisy").

In the datasets, the images are the same as in the original dataset (covariates are not changed). However, some of the labels are randomly swapped. Hence, only labels were changed, while covariates were kept as in the original dataset. The motivation for the creation of this dataset is due to the fact that conventional image classification datasets essentially contain no aleatoric uncertainty (Kapoor et al., 2022). To mitigate the limitation, which is critical for our evaluation, we introduce the label noise manually. By nature, this noise is aleatoric.

**CIFAR10-N.** We decide to do the following pairs of labels that are randomly swapped: 1 to 7, 7 to 1, 3 to 8, 8 to 3, 2 to 5, and 5 to 2.

**CIFAR100-N.** We decided to randomly swap the following pairs of labels: 1 to 7, 7 to 1, 3 to 8, 8 to 3, 2 to 5, 5 to 2, 10 to 20, 20 to 10, 40 to 50, 50 to 40, 90 to 99, 99 to 90, 25 to 75, 75 to 25, 17 to 71, 71 to 17, 13 to 31, 31 to 13, and 24 to 42, 42 to 24.

Both there noisy datasets were used in training. When evaluating misclassification detection, we use original versions of these datasets.

## I.2 IMAGENET DATASETS

Here we present descriptions of variations of ImageNet datasets we have used.

**ImageNet-A.** ImageNet-A (Hendrycks et al., 2021b) is dataset very similar to ImageNet test set, but proven to be more challenging for existing classification models. It contains real-world, unmodified, and naturally occurring examples that are misclassified by ResNet models.

**ImageNet-O.** Closely related to ImageNet-A, ImageNet-O (Hendrycks et al., 2021b) is dataset that contains real-world examples, with classes that are not present in standard ImageNet dataset. It can be used to test out of distribution detection.

**ImageNet-R.** This dataset contains renditions of 200 classes from ImageNet (Hendrycks et al., 2021a). Artistic rendition introduces a significant distribution shift and present a challenging task for classification models. It can be used to test robustness of model and out of distribution detection.

# J ADDITIONAL EXPERIMENTS ON OUT-OF-DISTRIBUTION DETECTION

In this section, we provide additional experiments on out-of-distribution detection. Here, we use different instantiations of $G$ for loss function and for uncertainty quantification, as well as try different in-distribution datasets. Results for CIFAR10 are in Tables 5, 6 and 7. For CIFAR100, additional

Table 5: AUROC for out-of-distribution detection for CIFAR10 (in-distribution). As a loss function for training and for uncertainty quantification we used **Log Score**.

| | CIFAR100 | SVHN | TinyImageNet | CIFAR10C-1 | CIFAR10C-2 | CIFAR10C-3 | CIFAR10C-4 | CIFAR10C-5 |
|---|---|---|---|---|---|---|---|---|
| LogScore $\bar{R}_{Bayes}^{(3)}$ | $90.93 \pm 0.03$ | $95.76 \pm 0.41$ | $90.35 \pm 0.07$ | $60.66 \pm 0.11$ | $67.44 \pm 0.11$ | $72.14 \pm 0.09$ | $76.82 \pm 0.09$ | $82.93 \pm 0.14$ |
| LogScore $\bar{R}_{Bayes}^{(2)}$ | $91.3 \pm 0.05$ | $96.06 \pm 0.49$ | $90.67 \pm 0.04$ | $60.98 \pm 0.1$ | $67.87 \pm 0.09$ | $72.61 \pm 0.08$ | $77.33 \pm 0.09$ | $83.48 \pm 0.15$ |
| LogScore $\bar{R}_{Bayes}^{(1)}$ | $91.36 \pm 0.05$ | $96.01 \pm 0.39$ | $90.84 \pm 0.04$ | $60.96 \pm 0.09$ | $67.84 \pm 0.09$ | $72.59 \pm 0.08$ | $77.31 \pm 0.08$ | $83.48 \pm 0.13$ |
| LogScore $\bar{R}_{Exc}^{(3,3)}$ | $50.0 \pm 0.0$ | $50.0 \pm 0.0$ | $50.0 \pm 0.0$ | $50.0 \pm 0.0$ | $50.0 \pm 0.0$ | $50.0 \pm 0.0$ | $50.0 \pm 0.0$ | $50.0 \pm 0.0$ |
| LogScore $\bar{R}_{Exc}^{(3,2)}$ | $89.08 \pm 0.14$ | $92.57 \pm 1.18$ | $88.15 \pm 0.13$ | $60.87 \pm 0.12$ | $67.46 \pm 0.1$ | $72.04 \pm 0.09$ | $76.58 \pm 0.12$ | $82.24 \pm 0.18$ |
| LogScore $\bar{R}_{Exc}^{(3,1)}$ | $90.38 \pm 0.06$ | $94.31 \pm 0.91$ | $89.54 \pm 0.06$ | $61.09 \pm 0.11$ | $67.86 \pm 0.09$ | $72.54 \pm 0.09$ | $77.22 \pm 0.13$ | $83.11 \pm 0.19$ |
| LogScore $\bar{R}_{Exc}^{(2,3)}$ | $88.64 \pm 0.15$ | $91.95 \pm 1.18$ | $87.68 \pm 0.13$ | $60.82 \pm 0.12$ | $67.36 \pm 0.09$ | $71.89 \pm 0.08$ | $76.38 \pm 0.12$ | $81.94 \pm 0.18$ |
| LogScore $\bar{R}_{Exc}^{(2,2)}$ | $50.0 \pm 0.0$ | $50.0 \pm 0.0$ | $50.0 \pm 0.0$ | $50.0 \pm 0.0$ | $50.0 \pm 0.0$ | $50.0 \pm 0.0$ | $50.0 \pm 0.0$ | $50.0 \pm 0.0$ |
| LogScore $\bar{R}_{Exc}^{(2,1)}$ | $90.06 \pm 0.06$ | $93.92 \pm 1.06$ | $89.19 \pm 0.07$ | $61.06 \pm 0.11$ | $67.8 \pm 0.09$ | $72.45 \pm 0.08$ | $77.1 \pm 0.13$ | $82.93 \pm 0.2$ |
| LogScore $\bar{R}_{Exc}^{(1,3)}$ | $89.99 \pm 0.08$ | $93.84 \pm 1.06$ | $89.12 \pm 0.07$ | $61.06 \pm 0.11$ | $67.79 \pm 0.09$ | $72.43 \pm 0.08$ | $77.07 \pm 0.13$ | $82.9 \pm 0.2$ |
| LogScore $\bar{R}_{Exc}^{(1,2)}$ | $90.38 \pm 0.08$ | $94.35 \pm 0.94$ | $89.57 \pm 0.07$ | $61.1 \pm 0.11$ | $67.87 \pm 0.09$ | $72.55 \pm 0.09$ | $77.23 \pm 0.13$ | $83.14 \pm 0.2$ |
| LogScore $\bar{R}_{Exc}^{(1,1)}$ | $90.17 \pm 0.06$ | $94.08 \pm 1.02$ | $89.32 \pm 0.07$ | $61.07 \pm 0.11$ | $67.82 \pm 0.09$ | $72.49 \pm 0.08$ | $77.15 \pm 0.13$ | $83.0 \pm 0.2$ |
| LogScore $\bar{R}_{Tot}^{(3,3)}$ | $90.93 \pm 0.03$ | $95.76 \pm 0.41$ | $90.35 \pm 0.07$ | $60.66 \pm 0.11$ | $67.44 \pm 0.11$ | $72.14 \pm 0.09$ | $76.82 \pm 0.09$ | $82.93 \pm 0.14$ |
| LogScore $\bar{R}_{Tot}^{(3,2)}$ | $90.95 \pm 0.04$ | $95.8 \pm 0.44$ | $90.33 \pm 0.06$ | $60.7 \pm 0.11$ | $67.49 \pm 0.1$ | $72.19 \pm 0.09$ | $76.88 \pm 0.09$ | $83.0 \pm 0.15$ |
| LogScore $\bar{R}_{Tot}^{(3,1)}$ | $90.72 \pm 0.04$ | $95.44 \pm 0.53$ | $90.03 \pm 0.06$ | $60.71 \pm 0.11$ | $67.49 \pm 0.1$ | $72.17 \pm 0.09$ | $76.83 \pm 0.1$ | $82.88 \pm 0.17$ |
| LogScore $\bar{R}_{Tot}^{(2,3)}$ | $91.15 \pm 0.05$ | $95.91 \pm 0.56$ | $90.49 \pm 0.04$ | $60.98 \pm 0.09$ | $67.85 \pm 0.09$ | $72.58 \pm 0.08$ | $77.29 \pm 0.09$ | $83.41 \pm 0.17$ |
| LogScore $\bar{R}_{Tot}^{(2,2)}$ | $91.3 \pm 0.05$ | $96.06 \pm 0.49$ | $90.67 \pm 0.04$ | $60.98 \pm 0.1$ | $67.87 \pm 0.09$ | $72.61 \pm 0.08$ | $77.33 \pm 0.09$ | $83.48 \pm 0.15$ |
| LogScore $\bar{R}_{Tot}^{(2,1)}$ | $90.81 \pm 0.04$ | $95.39 \pm 0.71$ | $90.06 \pm 0.05$ | $60.96 \pm 0.1$ | $67.8 \pm 0.1$ | $72.49 \pm 0.09$ | $77.16 \pm 0.11$ | $83.18 \pm 0.19$ |
| LogScore $\bar{R}_{Tot}^{(1,3)}$ | $91.15 \pm 0.05$ | $95.91 \pm 0.56$ | $90.49 \pm 0.04$ | $60.98 \pm 0.09$ | $67.85 \pm 0.09$ | $72.58 \pm 0.08$ | $77.29 \pm 0.09$ | $83.41 \pm 0.17$ |
| LogScore $\bar{R}_{Tot}^{(1,2)}$ | $91.3 \pm 0.05$ | $96.06 \pm 0.49$ | $90.67 \pm 0.04$ | $60.98 \pm 0.1$ | $67.87 \pm 0.09$ | $72.61 \pm 0.08$ | $77.33 \pm 0.09$ | $83.48 \pm 0.15$ |
| LogScore $\bar{R}_{Tot}^{(1,1)}$ | $90.81 \pm 0.04$ | $95.39 \pm 0.71$ | $90.06 \pm 0.05$ | $60.96 \pm 0.1$ | $67.8 \pm 0.1$ | $72.49 \pm 0.09$ | $77.16 \pm 0.11$ | $83.18 \pm 0.19$ |
| LogScore $E(x; \mathbb{E}_\theta f_\theta)$ | $91.12 \pm 0.02$ | $96.69 \pm 0.34$ | $90.68 \pm 0.05$ | $60.61 \pm 0.13$ | $67.46 \pm 0.14$ | $72.25 \pm 0.12$ | $76.97 \pm 0.12$ | $83.22 \pm 0.18$ |
| LogScore $\mathbb{E}_\theta E(x; f_\theta)$ | $91.1 \pm 0.01$ | $96.56 \pm 0.44$ | $90.77 \pm 0.04$ | $60.6 \pm 0.13$ | $67.46 \pm 0.13$ | $72.25 \pm 0.11$ | $76.96 \pm 0.11$ | $83.21 \pm 0.16$ |

results are in Table 8. For TinyImageNet, results are in Table 9. In all these experiments, we computed energy scores only for Log score, as these terms naturally appear only for this instantiation.

Note, that for CIFAR10 and CIFAR100, we used matching function $G$ for the loss function and for uncertainty quantification. For TinyImageNet, we had only one loss function, and we apply different instantiations of $G$ only for the evaluation of uncertainty measures.

From all these tables one can observe, that Log Score-based measures typically outperform others, that justifies them as a popular practical choice in uncertainty quantification problems. Also, we see that for "hard-OOD" datasets, Bayes (and Total) risks usually perform better, than Excess risk. In "soft-OOD" cases, such as CIFAR10C for CIFAR10 and ImageNet-O for TinyImageNet, their performance is close, and Excess risk is a good choice.

From the results for TinyImageNet 9, we see, that matching combination of Log Score is usually better, than other instantiations. Note, that since central prediction for Zero One Score is not well defined, we excluded it from the table.

Table 6: AUROC for out-of-distribution detection for CIFAR10 (in-distribution). As a loss function for training and for uncertainty quantification we used **Brier Score**. Note, that due to the symmetrical nature of Brier score, many instances results to the same values.

| | CIFAR100 | SVHN | TinyImageNet | CIFAR10C-1 | CIFAR10C-2 | CIFAR10C-3 | CIFAR10C-4 | CIFAR10C-5 |
|---|---|---|---|---|---|---|---|---|
| BrierScore $\tilde{R}_{Bayes}^{(3)}$ | $90.38 \pm 0.18$ | $96.26 \pm 0.34$ | $89.71 \pm 0.12$ | $61.02 \pm 0.13$ | $68.04 \pm 0.17$ | $72.46 \pm 0.19$ | $76.82 \pm 0.19$ | $82.45 \pm 0.11$ |
| BrierScore $\tilde{R}_{Bayes}^{(2)}$ | $90.38 \pm 0.18$ | $96.26 \pm 0.34$ | $89.71 \pm 0.12$ | $61.02 \pm 0.13$ | $68.04 \pm 0.17$ | $72.46 \pm 0.19$ | $76.82 \pm 0.19$ | $82.45 \pm 0.11$ |
| BrierScore $\tilde{R}_{Bayes}^{(1)}$ | $90.44 \pm 0.17$ | $96.09 \pm 0.52$ | $89.89 \pm 0.13$ | $61.03 \pm 0.12$ | $68.04 \pm 0.16$ | $72.46 \pm 0.18$ | $76.83 \pm 0.18$ | $82.47 \pm 0.1$ |
| BrierScore $\tilde{R}_{Exc}^{(3,3)}$ | $50.0 \pm 0.0$ | $50.0 \pm 0.0$ | $50.0 \pm 0.0$ | $50.0 \pm 0.0$ | $50.0 \pm 0.0$ | $50.0 \pm 0.0$ | $50.0 \pm 0.0$ | $50.0 \pm 0.0$ |
| BrierScore $\tilde{R}_{Exc}^{(3,2)}$ | $50.0 \pm 0.0$ | $50.0 \pm 0.0$ | $50.0 \pm 0.0$ | $50.0 \pm 0.0$ | $50.0 \pm 0.0$ | $50.0 \pm 0.0$ | $50.0 \pm 0.0$ | $50.0 \pm 0.0$ |
| BrierScore $\tilde{R}_{Exc}^{(3,1)}$ | $89.24 \pm 0.19$ | $94.19 \pm 0.49$ | $88.37 \pm 0.12$ | $60.11 \pm 0.15$ | $66.59 \pm 0.24$ | $71.19 \pm 0.3$ | $75.7 \pm 0.22$ | $81.52 \pm 0.16$ |
| BrierScore $\tilde{R}_{Exc}^{(2,3)}$ | $50.0 \pm 0.0$ | $50.0 \pm 0.0$ | $50.0 \pm 0.0$ | $50.0 \pm 0.0$ | $50.0 \pm 0.0$ | $50.0 \pm 0.0$ | $50.0 \pm 0.0$ | $50.0 \pm 0.0$ |
| BrierScore $\tilde{R}_{Exc}^{(2,2)}$ | $50.0 \pm 0.0$ | $50.0 \pm 0.0$ | $50.0 \pm 0.0$ | $50.0 \pm 0.0$ | $50.0 \pm 0.0$ | $50.0 \pm 0.0$ | $50.0 \pm 0.0$ | $50.0 \pm 0.0$ |
| BrierScore $\tilde{R}_{Exc}^{(2,1)}$ | $89.24 \pm 0.19$ | $94.19 \pm 0.49$ | $88.37 \pm 0.12$ | $60.11 \pm 0.15$ | $66.59 \pm 0.24$ | $71.19 \pm 0.3$ | $75.7 \pm 0.22$ | $81.52 \pm 0.16$ |
| BrierScore $\tilde{R}_{Exc}^{(1,3)}$ | $89.24 \pm 0.19$ | $94.19 \pm 0.49$ | $88.37 \pm 0.12$ | $60.11 \pm 0.15$ | $66.59 \pm 0.24$ | $71.19 \pm 0.3$ | $75.7 \pm 0.22$ | $81.52 \pm 0.16$ |
| BrierScore $\tilde{R}_{Exc}^{(1,2)}$ | $89.24 \pm 0.19$ | $94.19 \pm 0.49$ | $88.37 \pm 0.12$ | $60.11 \pm 0.15$ | $66.59 \pm 0.24$ | $71.19 \pm 0.3$ | $75.7 \pm 0.22$ | $81.52 \pm 0.16$ |
| BrierScore $\tilde{R}_{Exc}^{(1,1)}$ | $89.24 \pm 0.19$ | $94.19 \pm 0.49$ | $88.37 \pm 0.12$ | $60.11 \pm 0.15$ | $66.59 \pm 0.24$ | $71.19 \pm 0.3$ | $75.7 \pm 0.22$ | $81.52 \pm 0.16$ |
| BrierScore $\tilde{R}_{Tot}^{(3,3)}$ | $90.38 \pm 0.18$ | $96.26 \pm 0.34$ | $89.71 \pm 0.12$ | $61.02 \pm 0.13$ | $68.04 \pm 0.17$ | $72.46 \pm 0.19$ | $76.82 \pm 0.19$ | $82.45 \pm 0.11$ |
| BrierScore $\tilde{R}_{Tot}^{(3,2)}$ | $90.38 \pm 0.18$ | $96.26 \pm 0.34$ | $89.71 \pm 0.12$ | $61.02 \pm 0.13$ | $68.04 \pm 0.17$ | $72.46 \pm 0.19$ | $76.82 \pm 0.19$ | $82.45 \pm 0.11$ |
| BrierScore $\tilde{R}_{Tot}^{(3,1)}$ | $90.04 \pm 0.18$ | $95.83 \pm 0.34$ | $89.29 \pm 0.12$ | $60.99 \pm 0.13$ | $67.98 \pm 0.17$ | $72.37 \pm 0.19$ | $76.7 \pm 0.2$ | $82.25 \pm 0.13$ |
| BrierScore $\tilde{R}_{Tot}^{(2,3)}$ | $90.38 \pm 0.18$ | $96.26 \pm 0.34$ | $89.71 \pm 0.12$ | $61.02 \pm 0.13$ | $68.04 \pm 0.17$ | $72.46 \pm 0.19$ | $76.82 \pm 0.19$ | $82.45 \pm 0.11$ |
| BrierScore $\tilde{R}_{Tot}^{(2,2)}$ | $90.38 \pm 0.18$ | $96.26 \pm 0.34$ | $89.71 \pm 0.12$ | $61.02 \pm 0.13$ | $68.04 \pm 0.17$ | $72.46 \pm 0.19$ | $76.82 \pm 0.19$ | $82.45 \pm 0.11$ |
| BrierScore $\tilde{R}_{Tot}^{(2,1)}$ | $90.04 \pm 0.18$ | $95.83 \pm 0.34$ | $89.29 \pm 0.12$ | $60.99 \pm 0.13$ | $67.98 \pm 0.17$ | $72.37 \pm 0.19$ | $76.7 \pm 0.2$ | $82.25 \pm 0.13$ |
| BrierScore $\tilde{R}_{Tot}^{(1,3)}$ | $90.38 \pm 0.18$ | $96.26 \pm 0.34$ | $89.71 \pm 0.12$ | $61.02 \pm 0.13$ | $68.04 \pm 0.17$ | $72.46 \pm 0.19$ | $76.82 \pm 0.19$ | $82.45 \pm 0.11$ |
| BrierScore $\tilde{R}_{Tot}^{(1,2)}$ | $90.38 \pm 0.18$ | $96.26 \pm 0.34$ | $89.71 \pm 0.12$ | $61.02 \pm 0.13$ | $68.04 \pm 0.17$ | $72.46 \pm 0.19$ | $76.82 \pm 0.19$ | $82.45 \pm 0.11$ |
| BrierScore $\tilde{R}_{Tot}^{(1,1)}$ | $90.04 \pm 0.18$ | $95.83 \pm 0.34$ | $89.29 \pm 0.12$ | $60.99 \pm 0.13$ | $67.98 \pm 0.17$ | $72.37 \pm 0.19$ | $76.7 \pm 0.2$ | $82.25 \pm 0.13$ |

Table 7: AUROC for out-of-distribution detection for CIFAR10 (in-distribution). As a loss function for training and for uncertainty quantification we used **Spherical Score**. Note, that due to the symmetrical nature of Brier score, many instances results to the same values.

| | CIFAR100 | SVHN | TinyImageNet | CIFAR10C-1 | CIFAR10C-2 | CIFAR10C-3 | CIFAR10C-4 | CIFAR10C-5 |
|---|---|---|---|---|---|---|---|---|
| SphericalScore $\tilde{R}_{Bayes}^{(3)}$ | $90.09 \pm 0.04$ | $95.78 \pm 0.64$ | $89.4 \pm 0.18$ | $61.42 \pm 0.19$ | $68.45 \pm 0.22$ | $72.86 \pm 0.26$ | $77.3 \pm 0.31$ | $82.89 \pm 0.39$ |
| SphericalScore $\tilde{R}_{Bayes}^{(2)}$ | $90.42 \pm 0.03$ | $96.22 \pm 0.54$ | $89.81 \pm 0.18$ | $61.46 \pm 0.2$ | $68.52 \pm 0.23$ | $72.94 \pm 0.28$ | $77.41 \pm 0.33$ | $83.03 \pm 0.42$ |
| SphericalScore $\tilde{R}_{Bayes}^{(1)}$ | $90.46 \pm 0.04$ | $96.15 \pm 0.4$ | $89.95 \pm 0.19$ | $61.47 \pm 0.19$ | $68.54 \pm 0.23$ | $72.95 \pm 0.28$ | $77.39 \pm 0.34$ | $82.99 \pm 0.43$ |
| SphericalScore $\tilde{R}_{Exc}^{(3,3)}$ | $50.0 \pm 0.0$ | $50.0 \pm 0.0$ | $50.0 \pm 0.0$ | $50.0 \pm 0.0$ | $50.0 \pm 0.0$ | $50.0 \pm 0.0$ | $50.0 \pm 0.0$ | $50.0 \pm 0.0$ |
| SphericalScore $\tilde{R}_{Exc}^{(3,2)}$ | $87.97 \pm 0.09$ | $93.18 \pm 0.55$ | $87.2 \pm 0.18$ | $59.2 \pm 0.19$ | $65.18 \pm 0.2$ | $69.61 \pm 0.23$ | $74.24 \pm 0.22$ | $80.43 \pm 0.23$ |
| SphericalScore $\tilde{R}_{Exc}^{(3,1)}$ | $89.1 \pm 0.15$ | $93.78 \pm 0.66$ | $88.27 \pm 0.2$ | $60.54 \pm 0.18$ | $66.88 \pm 0.26$ | $71.35 \pm 0.29$ | $75.95 \pm 0.29$ | $81.88 \pm 0.33$ |
| SphericalScore $\tilde{R}_{Exc}^{(2,3)}$ | $88.0 \pm 0.17$ | $93.06 \pm 0.52$ | $87.1 \pm 0.13$ | $59.29 \pm 0.11$ | $65.18 \pm 0.09$ | $69.61 \pm 0.1$ | $74.24 \pm 0.17$ | $80.39 \pm 0.21$ |
| SphericalScore $\tilde{R}_{Exc}^{(2,2)}$ | $50.0 \pm 0.0$ | $50.0 \pm 0.0$ | $50.0 \pm 0.0$ | $50.0 \pm 0.0$ | $50.0 \pm 0.0$ | $50.0 \pm 0.0$ | $50.0 \pm 0.0$ | $50.0 \pm 0.0$ |
| SphericalScore $\tilde{R}_{Exc}^{(2,1)}$ | $89.03 \pm 0.15$ | $93.66 \pm 0.64$ | $88.2 \pm 0.19$ | $60.53 \pm 0.18$ | $66.87 \pm 0.25$ | $71.34 \pm 0.29$ | $75.93 \pm 0.29$ | $81.85 \pm 0.33$ |
| SphericalScore $\tilde{R}_{Exc}^{(1,3)}$ | $89.12 \pm 0.15$ | $93.77 \pm 0.65$ | $88.29 \pm 0.2$ | $60.54 \pm 0.18$ | $66.88 \pm 0.26$ | $71.36 \pm 0.3$ | $75.96 \pm 0.29$ | $81.89 \pm 0.34$ |
| SphericalScore $\tilde{R}_{Exc}^{(1,2)}$ | $89.17 \pm 0.14$ | $93.86 \pm 0.66$ | $88.35 \pm 0.2$ | $60.54 \pm 0.18$ | $66.88 \pm 0.26$ | $71.37 \pm 0.3$ | $75.97 \pm 0.29$ | $81.91 \pm 0.34$ |
| SphericalScore $\tilde{R}_{Exc}^{(1,1)}$ | $89.11 \pm 0.15$ | $93.77 \pm 0.66$ | $88.28 \pm 0.2$ | $60.54 \pm 0.18$ | $66.88 \pm 0.26$ | $71.36 \pm 0.29$ | $75.96 \pm 0.29$ | $81.89 \pm 0.33$ |
| SphericalScore $\tilde{R}_{Tot}^{(3,3)}$ | $90.09 \pm 0.04$ | $95.78 \pm 0.64$ | $89.4 \pm 0.18$ | $61.42 \pm 0.19$ | $68.45 \pm 0.22$ | $72.86 \pm 0.26$ | $77.3 \pm 0.31$ | $82.89 \pm 0.39$ |
| SphericalScore $\tilde{R}_{Tot}^{(3,2)}$ | $90.03 \pm 0.05$ | $95.7 \pm 0.65$ | $89.33 \pm 0.18$ | $61.42 \pm 0.19$ | $68.44 \pm 0.22$ | $72.84 \pm 0.26$ | $77.28 \pm 0.31$ | $82.86 \pm 0.38$ |
| SphericalScore $\tilde{R}_{Tot}^{(3,1)}$ | $89.82 \pm 0.06$ | $95.36 \pm 0.68$ | $89.08 \pm 0.17$ | $61.39 \pm 0.19$ | $68.4 \pm 0.22$ | $72.78 \pm 0.26$ | $77.21 \pm 0.31$ | $82.75 \pm 0.38$ |
| SphericalScore $\tilde{R}_{Tot}^{(2,3)}$ | $90.39 \pm 0.03$ | $96.2 \pm 0.55$ | $89.77 \pm 0.18$ | $61.46 \pm 0.2$ | $68.52 \pm 0.23$ | $72.94 \pm 0.27$ | $77.41 \pm 0.33$ | $83.03 \pm 0.41$ |
| SphericalScore $\tilde{R}_{Tot}^{(2,2)}$ | $90.42 \pm 0.03$ | $96.22 \pm 0.54$ | $89.81 \pm 0.18$ | $61.46 \pm 0.2$ | $68.52 \pm 0.23$ | $72.94 \pm 0.28$ | $77.41 \pm 0.33$ | $83.03 \pm 0.42$ |
| SphericalScore $\tilde{R}_{Tot}^{(2,1)}$ | $90.24 \pm 0.03$ | $96.0 \pm 0.6$ | $89.59 \pm 0.18$ | $61.44 \pm 0.19$ | $68.49 \pm 0.23$ | $72.9 \pm 0.27$ | $77.36 \pm 0.32$ | $82.96 \pm 0.4$ |
| SphericalScore $\tilde{R}_{Tot}^{(1,3)}$ | $90.39 \pm 0.03$ | $96.2 \pm 0.55$ | $89.77 \pm 0.18$ | $61.46 \pm 0.2$ | $68.52 \pm 0.23$ | $72.94 \pm 0.27$ | $77.41 \pm 0.33$ | $83.03 \pm 0.41$ |
| SphericalScore $\tilde{R}_{Tot}^{(1,2)}$ | $90.42 \pm 0.03$ | $96.22 \pm 0.54$ | $89.81 \pm 0.18$ | $61.46 \pm 0.2$ | $68.52 \pm 0.23$ | $72.94 \pm 0.28$ | $77.41 \pm 0.33$ | $83.03 \pm 0.42$ |
| SphericalScore $\tilde{R}_{Tot}^{(1,1)}$ | $90.24 \pm 0.03$ | $96.0 \pm 0.6$ | $89.59 \pm 0.18$ | $61.44 \pm 0.19$ | $68.49 \pm 0.23$ | $72.9 \pm 0.27$ | $77.36 \pm 0.32$ | $82.96 \pm 0.4$ |

Table 8: AUROC for out-of-distribution detection on CIFAR100 (in-distribution). For training and uncertainty quantification, we used: (Left) **Log Score**; (Middle) **Brier Score**; (Right) **Spherical Score**. Note that due to the symmetric nature of the Brier Score, many instances yield identical values.

| | Log Score | | Brier Score | | Spherical Score | |
|---|---|---|---|---|---|---|
| | CIFAR10 | SVHN | CIFAR10 | SVHN | CIFAR10 | SVHN |
| $\tilde{R}_{Bayes}^{(3)}$ | $77.53 \pm 0.24$ | $86.72 \pm 0.53$ | $77.18 \pm 0.36$ | $83.64 \pm 0.82$ | $77.35 \pm 0.17$ | $84.55 \pm 1.36$ |
| $\tilde{R}_{Bayes}^{(2)}$ | $77.44 \pm 0.23$ | $86.99 \pm 0.59$ | $77.18 \pm 0.36$ | $83.64 \pm 0.82$ | $77.57 \pm 0.16$ | $84.26 \pm 1.56$ |
| $\tilde{R}_{Bayes}^{(1)}$ | $77.30 \pm 0.23$ | $86.96 \pm 0.60$ | $76.93 \pm 0.35$ | $83.65 \pm 0.97$ | $77.39 \pm 0.15$ | $83.74 \pm 1.66$ |
| $\tilde{R}_{Exc}^{(3,3)}$ | $50.0 \pm 0.0$ | $50.0 \pm 0.0$ | $50.0 \pm 0.0$ | $50.0 \pm 0.0$ | $50.0 \pm 0.0$ | $50.0 \pm 0.0$ |
| $\tilde{R}_{Exc}^{(3,2)}$ | $65.33 \pm 0.17$ | $66.83 \pm 1.92$ | $50.0 \pm 0.0$ | $50.0 \pm 0.0$ | $57.90 \pm 0.44$ | $57.16 \pm 2.88$ |
| $\tilde{R}_{Exc}^{(3,1)}$ | $72.68 \pm 0.13$ | $75.98 \pm 0.72$ | $63.26 \pm 0.51$ | $61.34 \pm 1.70$ | $66.32 \pm 0.46$ | $69.07 \pm 2.06$ |
| $\tilde{R}_{Exc}^{(2,3)}$ | $62.97 \pm 0.20$ | $63.81 \pm 2.00$ | $50.0 \pm 0.0$ | $50.0 \pm 0.0$ | $57.11 \pm 0.45$ | $56.03 \pm 2.86$ |
| $\tilde{R}_{Exc}^{(2,2)}$ | $50.0 \pm 0.0$ | $50.0 \pm 0.0$ | $50.0 \pm 0.0$ | $50.0 \pm 0.0$ | $50.0 \pm 0.0$ | $50.0 \pm 0.0$ |
| $\tilde{R}_{Exc}^{(2,1)}$ | $71.35 \pm 0.11$ | $74.17 \pm 0.95$ | $63.26 \pm 0.51$ | $61.34 \pm 1.70$ | $63.94 \pm 0.50$ | $65.66 \pm 2.25$ |
| $\tilde{R}_{Exc}^{(1,3)}$ | $71.42 \pm 0.10$ | $74.61 \pm 1.06$ | $63.26 \pm 0.51$ | $61.34 \pm 1.70$ | $64.77 \pm 0.48$ | $66.78 \pm 2.23$ |
| $\tilde{R}_{Exc}^{(1,2)}$ | $73.20 \pm 0.13$ | $77.06 \pm 0.79$ | $63.26 \pm 0.51$ | $61.34 \pm 1.70$ | $66.97 \pm 0.43$ | $69.85 \pm 2.02$ |
| $\tilde{R}_{Exc}^{(1,1)}$ | $72.08 \pm 0.12$ | $75.30 \pm 0.88$ | $63.26 \pm 0.51$ | $61.34 \pm 1.70$ | $65.30 \pm 0.48$ | $67.55 \pm 2.18$ |
| $\tilde{R}_{Tot}^{(3,3)}$ | $77.53 \pm 0.24$ | $86.72 \pm 0.53$ | $77.18 \pm 0.36$ | $83.64 \pm 0.82$ | $77.35 \pm 0.17$ | $84.55 \pm 1.36$ |
| $\tilde{R}_{Tot}^{(3,2)}$ | $77.55 \pm 0.24$ | $86.74 \pm 0.52$ | $77.18 \pm 0.36$ | $83.64 \pm 0.82$ | $77.08 \pm 0.18$ | $84.31 \pm 1.29$ |
| $\tilde{R}_{Tot}^{(3,1)}$ | $77.50 \pm 0.24$ | $86.49 \pm 0.50$ | $76.46 \pm 0.38$ | $81.88 \pm 0.48$ | $76.78 \pm 0.20$ | $84.20 \pm 1.16$ |
| $\tilde{R}_{Tot}^{(2,3)}$ | $77.41 \pm 0.23$ | $87.00 \pm 0.58$ | $77.18 \pm 0.36$ | $83.64 \pm 0.82$ | $77.62 \pm 0.15$ | $84.40 \pm 1.53$ |
| $\tilde{R}_{Tot}^{(2,2)}$ | $77.44 \pm 0.23$ | $86.99 \pm 0.59$ | $77.18 \pm 0.36$ | $83.64 \pm 0.82$ | $77.57 \pm 0.16$ | $84.26 \pm 1.56$ |
| $\tilde{R}_{Tot}^{(2,1)}$ | $77.39 \pm 0.24$ | $86.77 \pm 0.55$ | $76.46 \pm 0.38$ | $81.88 \pm 0.48$ | $77.65 \pm 0.16$ | $84.75 \pm 1.44$ |
| $\tilde{R}_{Tot}^{(1,3)}$ | $77.41 \pm 0.23$ | $87.00 \pm 0.58$ | $77.18 \pm 0.36$ | $83.64 \pm 0.82$ | $77.62 \pm 0.15$ | $84.40 \pm 1.53$ |
| $\tilde{R}_{Tot}^{(1,2)}$ | $77.44 \pm 0.23$ | $86.99 \pm 0.59$ | $77.18 \pm 0.36$ | $83.64 \pm 0.82$ | $77.57 \pm 0.16$ | $84.26 \pm 1.56$ |
| $\tilde{R}_{Tot}^{(1,1)}$ | $77.39 \pm 0.24$ | $86.77 \pm 0.55$ | $76.46 \pm 0.38$ | $81.88 \pm 0.48$ | $77.65 \pm 0.16$ | $84.75 \pm 1.44$ |
| $E(x; \mathbb{E}_\theta[f_\theta])$ | $77.05 \pm 0.32$ | $87.98 \pm 0.65$ | – | – | – | – |
| $\mathbb{E}_\theta[E(x; f_\theta)]$ | $76.71 \pm 0.32$ | $87.71 \pm 0.69$ | – | – | – | – |

Table 9: AUROC for out-of-distribution detection for TinyImageNet (in-distribution). As a loss function for training, we used **Cross-Entropy**(corresponds to Log Score). For uncertainty estimates, we used different options. From left to right: Log Score, Brier Score, Spherical Score, Zero One Score. Note, that ImageNet-O can be considered as "soft-OOD" for TinyImageNet. Note, that due to the symmetrical nature of Brier score, many instances results to the same values.

| | Log Score | | | Brier Score | | | Spherical Score | | | Zero One Score | | |
|---|---|---|---|---|---|---|---|---|---|---|---|---|
| | ImageNet-A | ImageNet-R | ImageNet-O | ImageNet-A | ImageNet-R | ImageNet-O | ImageNet-A | ImageNet-R | ImageNet-O | ImageNet-A | ImageNet-R | ImageNet-O |
| $\tilde{R}_{Bayes}^{(3)}$ | $83.61 \pm 0.2$ | $82.67 \pm 0.37$ | $72.86 \pm 0.3$ | $83.16 \pm 0.24$ | $82.14 \pm 0.34$ | $73.3 \pm 0.18$ | $81.51 \pm 0.22$ | $80.67 \pm 0.23$ | $74.17 \pm 0.22$ | - | - | - |
| $\tilde{R}_{Bayes}^{(2)}$ | $83.76 \pm 0.24$ | $82.78 \pm 0.37$ | $73.18 \pm 0.2$ | $83.16 \pm 0.24$ | $82.14 \pm 0.34$ | $73.3 \pm 0.18$ | $83.16 \pm 0.24$ | $82.14 \pm 0.34$ | $73.3 \pm 0.18$ | $82.41 \pm 0.25$ | $81.41 \pm 0.31$ | $73.18 \pm 0.16$ |
| $\tilde{R}_{Bayes}^{(1)}$ | $83.22 \pm 0.24$ | $82.23 \pm 0.39$ | $72.21 \pm 0.22$ | $82.4 \pm 0.24$ | $81.31 \pm 0.37$ | $71.89 \pm 0.17$ | $82.68 \pm 0.24$ | $81.6 \pm 0.36$ | $72.06 \pm 0.17$ | $82.27 \pm 0.23$ | $81.2 \pm 0.34$ | $71.9 \pm 0.14$ |
| $\tilde{R}_{Exc}^{(3,3)}$ | $50.0 \pm 0.0$ | $50.0 \pm 0.0$ | $50.0 \pm 0.0$ | $50.0 \pm 0.0$ | $50.0 \pm 0.0$ | $50.0 \pm 0.0$ | $50.0 \pm 0.0$ | $50.0 \pm 0.0$ | $50.0 \pm 0.0$ | - | - | - |
| $\tilde{R}_{Exc}^{(3,2)}$ | $70.4 \pm 0.45$ | $69.75 \pm 0.33$ | $70.31 \pm 0.33$ | $50.0 \pm 0.0$ | $50.0 \pm 0.0$ | $50.0 \pm 0.0$ | $74.49 \pm 0.28$ | $74.43 \pm 0.19$ | $72.75 \pm 0.3$ | - | - | - |
| $\tilde{R}_{Exc}^{(3,1)}$ | $77.56 \pm 0.28$ | $77.02 \pm 0.17$ | $74.46 \pm 0.22$ | $65.46 \pm 0.39$ | $65.99 \pm 0.37$ | $68.81 \pm 0.25$ | $76.55 \pm 0.3$ | $76.16 \pm 0.12$ | $73.16 \pm 0.22$ | - | - | - |
| $\tilde{R}_{Exc}^{(2,3)}$ | $65.94 \pm 0.44$ | $65.34 \pm 0.37$ | $67.49 \pm 0.36$ | $50.0 \pm 0.0$ | $50.0 \pm 0.0$ | $50.0 \pm 0.0$ | $68.85 \pm 0.33$ | $69.15 \pm 0.27$ | $70.18 \pm 0.28$ | - | - | - |
| $\tilde{R}_{Exc}^{(2,2)}$ | $50.0 \pm 0.0$ | $50.0 \pm 0.0$ | $50.0 \pm 0.0$ | $50.0 \pm 0.0$ | $50.0 \pm 0.0$ | $50.0 \pm 0.0$ | $50.0 \pm 0.0$ | $50.0 \pm 0.0$ | $50.0 \pm 0.0$ | $50.0 \pm 0.0$ | $50.0 \pm 0.0$ | $50.0 \pm 0.0$ |
| $\tilde{R}_{Exc}^{(2,1)}$ | $75.94 \pm 0.36$ | $75.4 \pm 0.23$ | $74.07 \pm 0.18$ | $65.46 \pm 0.39$ | $65.99 \pm 0.37$ | $68.81 \pm 0.25$ | $70.64 \pm 0.3$ | $70.78 \pm 0.23$ | $71.04 \pm 0.24$ | $67.15 \pm 0.31$ | $67.28 \pm 0.19$ | $65.96 \pm 0.33$ |
| $\tilde{R}_{Exc}^{(1,3)}$ | $76.21 \pm 0.43$ | $75.55 \pm 0.21$ | $73.84 \pm 0.17$ | $65.46 \pm 0.39$ | $65.99 \pm 0.37$ | $68.81 \pm 0.25$ | $71.41 \pm 0.3$ | $71.46 \pm 0.2$ | $71.36 \pm 0.21$ | - | - | - |
| $\tilde{R}_{Exc}^{(1,2)}$ | $79.09 \pm 0.33$ | $78.38 \pm 0.17$ | $74.79 \pm 0.25$ | $65.46 \pm 0.39$ | $65.99 \pm 0.37$ | $68.81 \pm 0.25$ | $72.78 \pm 0.28$ | $72.67 \pm 0.15$ | $71.76 \pm 0.17$ | $70.83 \pm 0.24$ | $70.66 \pm 0.13$ | $69.13 \pm 0.31$ |
| $\tilde{R}_{Exc}^{(1,1)}$ | $77.11 \pm 0.34$ | $76.52 \pm 0.19$ | $74.46 \pm 0.18$ | $65.46 \pm 0.39$ | $65.99 \pm 0.37$ | $68.81 \pm 0.25$ | $71.9 \pm 0.29$ | $71.91 \pm 0.18$ | $71.56 \pm 0.21$ | $68.05 \pm 0.32$ | $68.21 \pm 0.18$ | $68.22 \pm 0.27$ |
| $\tilde{R}_{Tot}^{(3,3)}$ | $83.61 \pm 0.2$ | $82.67 \pm 0.37$ | $72.86 \pm 0.3$ | $83.16 \pm 0.24$ | $82.14 \pm 0.34$ | $73.3 \pm 0.18$ | $81.51 \pm 0.22$ | $80.67 \pm 0.23$ | $74.17 \pm 0.22$ | - | - | - |
| $\tilde{R}_{Tot}^{(3,2)}$ | $83.77 \pm 0.24$ | $82.84 \pm 0.36$ | $73.3 \pm 0.27$ | $83.16 \pm 0.24$ | $82.14 \pm 0.34$ | $73.3 \pm 0.18$ | $80.62 \pm 0.24$ | $79.92 \pm 0.19$ | $74.21 \pm 0.25$ | - | - | - |
| $\tilde{R}_{Tot}^{(3,1)}$ | $83.88 \pm 0.19$ | $82.99 \pm 0.32$ | $74.09 \pm 0.27$ | $81.75 \pm 0.25$ | $81.08 \pm 0.22$ | $74.68 \pm 0.27$ | $80.47 \pm 0.26$ | $79.74 \pm 0.19$ | $74.11 \pm 0.23$ | - | - | - |
| $\tilde{R}_{Tot}^{(2,3)}$ | $83.93 \pm 0.24$ | $82.95 \pm 0.36$ | $73.72 \pm 0.16$ | $83.16 \pm 0.24$ | $82.14 \pm 0.34$ | $73.3 \pm 0.18$ | $83.27 \pm 0.23$ | $82.28 \pm 0.32$ | $73.76 \pm 0.19$ | - | - | - |
| $\tilde{R}_{Tot}^{(2,2)}$ | $83.76 \pm 0.24$ | $82.78 \pm 0.37$ | $73.18 \pm 0.2$ | $83.16 \pm 0.24$ | $82.14 \pm 0.34$ | $73.3 \pm 0.18$ | $83.16 \pm 0.24$ | $82.14 \pm 0.34$ | $73.3 \pm 0.18$ | $82.41 \pm 0.25$ | $81.41 \pm 0.31$ | $73.18 \pm 0.16$ |
| $\tilde{R}_{Tot}^{(2,1)}$ | $84.26 \pm 0.23$ | $83.32 \pm 0.31$ | $74.93 \pm 0.17$ | $81.75 \pm 0.25$ | $81.08 \pm 0.22$ | $74.68 \pm 0.27$ | $83.25 \pm 0.23$ | $82.27 \pm 0.31$ | $73.86 \pm 0.19$ | $82.52 \pm 0.23$ | $81.56 \pm 0.3$ | $73.35 \pm 0.19$ |
| $\tilde{R}_{Tot}^{(1,3)}$ | $83.93 \pm 0.24$ | $82.95 \pm 0.36$ | $73.72 \pm 0.16$ | $83.16 \pm 0.24$ | $82.14 \pm 0.34$ | $73.3 \pm 0.18$ | $83.27 \pm 0.23$ | $82.28 \pm 0.32$ | $73.76 \pm 0.19$ | - | - | - |
| $\tilde{R}_{Tot}^{(1,2)}$ | $83.76 \pm 0.24$ | $82.78 \pm 0.37$ | $73.18 \pm 0.2$ | $83.16 \pm 0.24$ | $82.14 \pm 0.34$ | $73.3 \pm 0.18$ | $83.16 \pm 0.24$ | $82.14 \pm 0.34$ | $73.3 \pm 0.18$ | $82.41 \pm 0.25$ | $81.41 \pm 0.31$ | $73.18 \pm 0.16$ |
| $\tilde{R}_{Tot}^{(1,1)}$ | $84.26 \pm 0.23$ | $83.32 \pm 0.31$ | $74.93 \pm 0.17$ | $81.75 \pm 0.25$ | $81.08 \pm 0.22$ | $74.68 \pm 0.27$ | $83.25 \pm 0.23$ | $82.27 \pm 0.31$ | $73.86 \pm 0.19$ | $82.52 \pm 0.23$ | $81.56 \pm 0.3$ | $73.35 \pm 0.19$ |
| $E(x; \mathbb{E}_\theta f_\theta)$ | $83.96 \pm 0.23$ | $83.28 \pm 0.41$ | $72.72 \pm 0.33$ | - | - | - | - | - | - | - | - | - |
| $\mathbb{E}_\theta E(x; f_\theta)$ | $82.99 \pm 0.26$ | $82.31 \pm 0.45$ | $71.15 \pm 0.34$ | - | - | - | - | - | - | - | - | - |

Table 10: AUROC for misclassification detection. As a loss function for training and for uncertainty quantification we used **Log Score**.

| | CIFAR10 | CIFAR100 | CIFAR10-N | CIFAR100-N | TinyImageNet |
|---|---|---|---|---|---|
| LogScore $\tilde{R}_{\text{Bayes}}^{(3)}$ | $94.76 \pm 0.09$ | $85.95 \pm 0.31$ | $80.44 \pm 3.69$ | $83.41 \pm 0.23$ | $86.55 \pm 0.26$ |
| LogScore $\tilde{R}_{\text{Bayes}}^{(2)}$ | $94.7 \pm 0.05$ | $85.13 \pm 0.35$ | $78.36 \pm 4.77$ | $81.92 \pm 0.36$ | $85.5 \pm 0.23$ |
| LogScore $\tilde{R}_{\text{Bayes}}^{(1)}$ | $94.39 \pm 0.08$ | $84.61 \pm 0.35$ | $78.26 \pm 4.11$ | $81.9 \pm 0.29$ | $84.99 \pm 0.24$ |
| LogScore $\tilde{R}_{\text{Exc}}^{(3,3)}$ | $50.0 \pm 0.0$ | $50.0 \pm 0.0$ | $50.0 \pm 0.0$ | $50.0 \pm 0.0$ | $50.0 \pm 0.0$ |
| LogScore $\tilde{R}_{\text{Exc}}^{(3,2)}$ | $92.14 \pm 0.19$ | $70.77 \pm 0.38$ | $61.28 \pm 6.53$ | $65.51 \pm 0.99$ | $71.77 \pm 0.13$ |
| LogScore $\tilde{R}_{\text{Exc}}^{(3,1)}$ | $94.4 \pm 0.13$ | $82.62 \pm 0.34$ | $69.98 \pm 6.08$ | $75.51 \pm 0.8$ | $82.74 \pm 0.34$ |
| LogScore $\tilde{R}_{\text{Exc}}^{(2,3)}$ | $91.54 \pm 0.19$ | $67.42 \pm 0.4$ | $59.24 \pm 6.11$ | $62.79 \pm 0.97$ | $66.83 \pm 0.11$ |
| LogScore $\tilde{R}_{\text{Exc}}^{(2,2)}$ | $50.0 \pm 0.0$ | $50.0 \pm 0.0$ | $50.0 \pm 0.0$ | $50.0 \pm 0.0$ | $50.0 \pm 0.0$ |
| LogScore $\tilde{R}_{\text{Exc}}^{(2,1)}$ | $94.01 \pm 0.14$ | $80.69 \pm 0.35$ | $67.7 \pm 6.09$ | $73.7 \pm 0.86$ | $80.18 \pm 0.31$ |
| LogScore $\tilde{R}_{\text{Exc}}^{(1,3)}$ | $93.72 \pm 0.16$ | $80.3 \pm 0.33$ | $66.85 \pm 5.94$ | $73.37 \pm 0.87$ | $79.25 \pm 0.25$ |
| LogScore $\tilde{R}_{\text{Exc}}^{(1,2)}$ | $94.23 \pm 0.14$ | $82.9 \pm 0.31$ | $69.64 \pm 6.1$ | $75.74 \pm 0.78$ | $83.22 \pm 0.28$ |
| LogScore $\tilde{R}_{\text{Exc}}^{(1,1)}$ | $94.1 \pm 0.14$ | $81.58 \pm 0.34$ | $68.39 \pm 6.07$ | $74.52 \pm 0.83$ | $81.33 \pm 0.29$ |
| LogScore $\tilde{R}_{\text{Tot}}^{(3,3)}$ | $94.76 \pm 0.09$ | $85.95 \pm 0.31$ | $80.44 \pm 3.69$ | $83.41 \pm 0.23$ | $86.55 \pm 0.26$ |
| LogScore $\tilde{R}_{\text{Tot}}^{(3,2)}$ | $94.77 \pm 0.05$ | $85.96 \pm 0.32$ | $80.21 \pm 3.96$ | $83.33 \pm 0.26$ | $86.5 \pm 0.25$ |
| LogScore $\tilde{R}_{\text{Tot}}^{(3,1)}$ | $94.74 \pm 0.07$ | $86.22 \pm 0.31$ | $79.38 \pm 3.85$ | $83.18 \pm 0.29$ | $86.65 \pm 0.26$ |
| LogScore $\tilde{R}_{\text{Tot}}^{(2,3)}$ | $94.4 \pm 0.06$ | $85.01 \pm 0.37$ | $76.26 \pm 5.04$ | $81.67 \pm 0.39$ | $85.18 \pm 0.23$ |
| LogScore $\tilde{R}_{\text{Tot}}^{(2,2)}$ | $94.7 \pm 0.05$ | $85.13 \pm 0.35$ | $78.36 \pm 4.77$ | $81.92 \pm 0.36$ | $85.5 \pm 0.23$ |
| LogScore $\tilde{R}_{\text{Tot}}^{(2,1)}$ | $94.5 \pm 0.06$ | $85.43 \pm 0.35$ | $75.73 \pm 4.68$ | $81.67 \pm 0.41$ | $85.58 \pm 0.22$ |
| LogScore $\tilde{R}_{\text{Tot}}^{(1,3)}$ | $94.4 \pm 0.06$ | $85.01 \pm 0.37$ | $76.26 \pm 5.04$ | $81.67 \pm 0.39$ | $85.18 \pm 0.23$ |
| LogScore $\tilde{R}_{\text{Tot}}^{(1,2)}$ | $94.7 \pm 0.05$ | $85.13 \pm 0.35$ | $78.36 \pm 4.77$ | $81.92 \pm 0.36$ | $85.5 \pm 0.23$ |
| LogScore $\tilde{R}_{\text{Tot}}^{(1,1)}$ | $94.5 \pm 0.06$ | $85.43 \pm 0.35$ | $75.73 \pm 4.68$ | $81.67 \pm 0.41$ | $85.58 \pm 0.22$ |
| LogScore $E(x; \mathbb{E}_\theta f_\theta)$ | $93.89 \pm 0.11$ | $82.95 \pm 0.34$ | $74.82 \pm 5.48$ | $76.74 \pm 0.64$ | $83.34 \pm 0.23$ |
| LogScore $\mathbb{E}_\theta E(x; f_\theta)$ | $93.38 \pm 0.15$ | $82.07 \pm 0.34$ | $74.61 \pm 5.23$ | $76.12 \pm 0.63$ | $82.4 \pm 0.24$ |

## K ADDITIONAL EXPERIMENTS ON MISCLASSIFICATION DETECTION

In this section, we present additional results for misclassification detection.

As training datasets, we consider CIFAR10, CIFAR100, their noisy versions as well as TinyImageNet (only with Cross-Entropy loss function).

In Table 10 we present results for all these datasets, when loss function and instantiation are generated by Log Score. In Tables 11 and 12, we present results for CIFAR-like datasets, as for TinyImageNet we have only Cross-Entropy loss function. From all these tables we see that, indeed, Bayes risk and Total risk are the best choices when encounter misclassification detection problem. The gap is even more significant, when more label noise is introduced. Moreover, instantiations for Log Score are typically provide better AUROC, than others. Similarly to the out-of-distribution detection, it justifies typical practical choice for Log Score-based measures.

For TinyImageNet, results are presented in Table 13. Similarly to Table 9, one loss function was used for training, while different instantiations of $G$ were used for the measures of uncertainty. Interestingly, the best results are reached by Zero One score, that is also a popular choice of practitioners.

Table 11: AUROC for misclassification detection. As a loss function for training and for uncertainty quantification we used **Brier Score**. Note, that due to the symmetrical nature of Brier score, many instances results to the same values.

| | CIFAR10 | CIFAR100 | CIFAR10-N | CIFAR100-N |
|---|---|---|---|---|
| BrierScore $\tilde{R}_{Bayes}^{(3)}$ | $94.68 \pm 0.3$ | $86.02 \pm 0.26$ | $80.64 \pm 3.17$ | $84.1 \pm 0.13$ |
| BrierScore $\tilde{R}_{Bayes}^{(2)}$ | $94.68 \pm 0.3$ | $86.02 \pm 0.26$ | $80.64 \pm 3.17$ | $84.1 \pm 0.13$ |
| BrierScore $\tilde{R}_{Bayes}^{(1)}$ | $94.22 \pm 0.27$ | $85.09 \pm 0.27$ | $79.2 \pm 2.35$ | $83.95 \pm 0.14$ |
| BrierScore $\tilde{R}_{Exc}^{(3,3)}$ | $50.0 \pm 0.0$ | $50.0 \pm 0.0$ | $50.0 \pm 0.0$ | $50.0 \pm 0.0$ |
| BrierScore $\tilde{R}_{Exc}^{(3,2)}$ | $50.0 \pm 0.0$ | $50.0 \pm 0.0$ | $50.0 \pm 0.0$ | $50.0 \pm 0.0$ |
| BrierScore $\tilde{R}_{Exc}^{(3,1)}$ | $94.2 \pm 0.34$ | $72.71 \pm 0.34$ | $76.31 \pm 3.39$ | $65.34 \pm 0.43$ |
| BrierScore $\tilde{R}_{Exc}^{(2,3)}$ | $50.0 \pm 0.0$ | $50.0 \pm 0.0$ | $50.0 \pm 0.0$ | $50.0 \pm 0.0$ |
| BrierScore $\tilde{R}_{Exc}^{(2,2)}$ | $50.0 \pm 0.0$ | $50.0 \pm 0.0$ | $50.0 \pm 0.0$ | $50.0 \pm 0.0$ |
| BrierScore $\tilde{R}_{Exc}^{(2,1)}$ | $94.2 \pm 0.34$ | $72.71 \pm 0.34$ | $76.31 \pm 3.39$ | $65.34 \pm 0.43$ |
| BrierScore $\tilde{R}_{Exc}^{(1,3)}$ | $94.2 \pm 0.34$ | $72.71 \pm 0.34$ | $76.31 \pm 3.39$ | $65.34 \pm 0.43$ |
| BrierScore $\tilde{R}_{Exc}^{(1,2)}$ | $94.2 \pm 0.34$ | $72.71 \pm 0.34$ | $76.31 \pm 3.39$ | $65.34 \pm 0.43$ |
| BrierScore $\tilde{R}_{Exc}^{(1,1)}$ | $94.2 \pm 0.34$ | $72.71 \pm 0.34$ | $76.31 \pm 3.39$ | $65.34 \pm 0.43$ |
| BrierScore $\tilde{R}_{Tot}^{(3,3)}$ | $94.68 \pm 0.3$ | $86.02 \pm 0.26$ | $80.64 \pm 3.17$ | $84.1 \pm 0.13$ |
| BrierScore $\tilde{R}_{Tot}^{(3,2)}$ | $94.68 \pm 0.3$ | $86.02 \pm 0.26$ | $80.64 \pm 3.17$ | $84.1 \pm 0.13$ |
| BrierScore $\tilde{R}_{Tot}^{(3,1)}$ | $94.61 \pm 0.31$ | $86.11 \pm 0.23$ | $79.78 \pm 3.23$ | $83.24 \pm 0.19$ |
| BrierScore $\tilde{R}_{Tot}^{(2,3)}$ | $94.68 \pm 0.3$ | $86.02 \pm 0.26$ | $80.64 \pm 3.17$ | $84.1 \pm 0.13$ |
| BrierScore $\tilde{R}_{Tot}^{(2,2)}$ | $94.68 \pm 0.3$ | $86.02 \pm 0.26$ | $80.64 \pm 3.17$ | $84.1 \pm 0.13$ |
| BrierScore $\tilde{R}_{Tot}^{(2,1)}$ | $94.61 \pm 0.31$ | $86.11 \pm 0.23$ | $79.78 \pm 3.23$ | $83.24 \pm 0.19$ |
| BrierScore $\tilde{R}_{Tot}^{(1,3)}$ | $94.68 \pm 0.3$ | $86.02 \pm 0.26$ | $80.64 \pm 3.17$ | $84.1 \pm 0.13$ |
| BrierScore $\tilde{R}_{Tot}^{(1,2)}$ | $94.68 \pm 0.3$ | $86.02 \pm 0.26$ | $80.64 \pm 3.17$ | $84.1 \pm 0.13$ |
| BrierScore $\tilde{R}_{Tot}^{(1,1)}$ | $94.61 \pm 0.31$ | $86.11 \pm 0.23$ | $79.78 \pm 3.23$ | $83.24 \pm 0.19$ |

Table 12: AUROC for misclassification detection. As a loss function for training and for uncertainty quantification we used **Spherical Score**.

| | CIFAR10 | CIFAR100 | CIFAR10-N | CIFAR100-N |
|---|---|---|---|---|
| SphericalScore $\tilde{R}_{Bayes}^{(3)}$ | $94.01 \pm 0.31$ | $85.37 \pm 0.25$ | $79.36 \pm 2.85$ | $80.13 \pm 0.6$ |
| SphericalScore $\tilde{R}_{Bayes}^{(2)}$ | $94.15 \pm 0.29$ | $84.73 \pm 0.24$ | $80.8 \pm 3.0$ | $80.53 \pm 0.54$ |
| SphericalScore $\tilde{R}_{Bayes}^{(1)}$ | $93.64 \pm 0.28$ | $83.97 \pm 0.25$ | $80.35 \pm 3.11$ | $80.03 \pm 0.52$ |
| SphericalScore $\tilde{R}_{Exc}^{(3,3)}$ | $50.0 \pm 0.0$ | $50.0 \pm 0.0$ | $50.0 \pm 0.0$ | $50.0 \pm 0.0$ |
| SphericalScore $\tilde{R}_{Exc}^{(3,2)}$ | $92.81 \pm 0.49$ | $65.29 \pm 0.48$ | $69.95 \pm 5.62$ | $61.62 \pm 0.33$ |
| SphericalScore $\tilde{R}_{Exc}^{(3,1)}$ | $93.46 \pm 0.47$ | $77.3 \pm 0.35$ | $75.18 \pm 3.6$ | $71.81 \pm 0.46$ |
| SphericalScore $\tilde{R}_{Exc}^{(2,3)}$ | $92.76 \pm 0.41$ | $64.33 \pm 0.47$ | $70.03 \pm 5.53$ | $60.8 \pm 0.32$ |
| SphericalScore $\tilde{R}_{Exc}^{(2,2)}$ | $50.0 \pm 0.0$ | $50.0 \pm 0.0$ | $50.0 \pm 0.0$ | $50.0 \pm 0.0$ |
| SphericalScore $\tilde{R}_{Exc}^{(2,1)}$ | $93.39 \pm 0.47$ | $73.76 \pm 0.36$ | $75.06 \pm 3.53$ | $68.84 \pm 0.42$ |
| SphericalScore $\tilde{R}_{Exc}^{(1,3)}$ | $93.51 \pm 0.47$ | $75.08 \pm 0.36$ | $75.22 \pm 3.63$ | $69.97 \pm 0.42$ |
| SphericalScore $\tilde{R}_{Exc}^{(1,2)}$ | $93.57 \pm 0.46$ | $78.25 \pm 0.34$ | $75.32 \pm 3.69$ | $72.69 \pm 0.44$ |
| SphericalScore $\tilde{R}_{Exc}^{(1,1)}$ | $93.49 \pm 0.47$ | $75.83 \pm 0.35$ | $75.2 \pm 3.61$ | $70.59 \pm 0.43$ |
| SphericalScore $\tilde{R}_{Tot}^{(3,3)}$ | $94.01 \pm 0.31$ | $85.37 \pm 0.25$ | $79.36 \pm 2.85$ | $80.13 \pm 0.6$ |
| SphericalScore $\tilde{R}_{Tot}^{(3,2)}$ | $93.98 \pm 0.31$ | $85.16 \pm 0.24$ | $79.29 \pm 2.84$ | $79.95 \pm 0.6$ |
| SphericalScore $\tilde{R}_{Tot}^{(3,1)}$ | $93.92 \pm 0.31$ | $85.42 \pm 0.24$ | $79.21 \pm 2.79$ | $80.13 \pm 0.61$ |
| SphericalScore $\tilde{R}_{Tot}^{(2,3)}$ | $94.15 \pm 0.29$ | $84.88 \pm 0.24$ | $80.66 \pm 3.03$ | $80.6 \pm 0.54$ |
| SphericalScore $\tilde{R}_{Tot}^{(2,2)}$ | $94.15 \pm 0.29$ | $84.73 \pm 0.24$ | $80.8 \pm 3.0$ | $80.53 \pm 0.54$ |
| SphericalScore $\tilde{R}_{Tot}^{(2,1)}$ | $94.14 \pm 0.3$ | $85.23 \pm 0.23$ | $80.15 \pm 2.99$ | $80.82 \pm 0.55$ |
| SphericalScore $\tilde{R}_{Tot}^{(1,3)}$ | $94.15 \pm 0.29$ | $84.88 \pm 0.24$ | $80.66 \pm 3.03$ | $80.6 \pm 0.54$ |
| SphericalScore $\tilde{R}_{Tot}^{(1,2)}$ | $94.15 \pm 0.29$ | $84.73 \pm 0.24$ | $80.8 \pm 3.0$ | $80.53 \pm 0.54$ |
| SphericalScore $\tilde{R}_{Tot}^{(1,1)}$ | $94.14 \pm 0.3$ | $85.23 \pm 0.23$ | $80.15 \pm 2.99$ | $80.82 \pm 0.55$ |

Table 13: AUROC for misclassification detection (TinyImageNet). As a loss function for training we used Cross-Entropy. Here, we try different instantiations of $G$ for uncertainty measures. Note, that due to the symmetrical nature of Brier score, many instances results to the same values.

| | Log Score | Brier Score | Spherical Score | Zero One Score |
|---|---|---|---|---|
| $\tilde{R}_{Bayes}^{(3)}$ | $86.55 \pm 0.26$ | $86.8 \pm 0.28$ | $85.83 \pm 0.36$ | - |
| $\tilde{R}_{Bayes}^{(2)}$ | $85.5 \pm 0.23$ | $86.8 \pm 0.28$ | $86.8 \pm 0.28$ | $87.23 \pm 0.35$ |
| $\tilde{R}_{Bayes}^{(1)}$ | $84.99 \pm 0.24$ | $85.89 \pm 0.27$ | $85.76 \pm 0.26$ | $85.88 \pm 0.27$ |
| $\tilde{R}_{Exc}^{(3,3)}$ | $50.0 \pm 0.0$ | $50.0 \pm 0.0$ | $50.0 \pm 0.0$ | - |
| $\tilde{R}_{Exc}^{(3,2)}$ | $71.77 \pm 0.13$ | $50.0 \pm 0.0$ | $81.82 \pm 0.47$ | - |
| $\tilde{R}_{Exc}^{(3,1)}$ | $82.74 \pm 0.34$ | $75.91 \pm 0.59$ | $83.6 \pm 0.47$ | - |
| $\tilde{R}_{Exc}^{(2,3)}$ | $66.83 \pm 0.11$ | $50.0 \pm 0.0$ | $78.32 \pm 0.58$ | - |
| $\tilde{R}_{Exc}^{(2,2)}$ | $50.0 \pm 0.0$ | $50.0 \pm 0.0$ | $50.0 \pm 0.0$ | $50.0 \pm 0.0$ |
| $\tilde{R}_{Exc}^{(2,1)}$ | $80.18 \pm 0.31$ | $75.91 \pm 0.59$ | $79.62 \pm 0.54$ | $73.59 \pm 0.44$ |
| $\tilde{R}_{Exc}^{(1,3)}$ | $79.25 \pm 0.25$ | $75.91 \pm 0.59$ | $80.73 \pm 0.54$ | - |
| $\tilde{R}_{Exc}^{(1,2)}$ | $83.22 \pm 0.28$ | $75.91 \pm 0.59$ | $81.95 \pm 0.55$ | $79.52 \pm 0.68$ |
| $\tilde{R}_{Exc}^{(1,1)}$ | $81.33 \pm 0.29$ | $75.91 \pm 0.59$ | $81.02 \pm 0.54$ | $76.78 \pm 0.56$ |
| $\tilde{R}_{Tot}^{(3,3)}$ | $86.55 \pm 0.26$ | $86.8 \pm 0.28$ | $85.83 \pm 0.36$ | - |
| $\tilde{R}_{Tot}^{(3,2)}$ | $86.5 \pm 0.25$ | $86.8 \pm 0.28$ | $85.39 \pm 0.37$ | - |
| $\tilde{R}_{Tot}^{(3,1)}$ | $86.65 \pm 0.26$ | $86.4 \pm 0.32$ | $85.43 \pm 0.39$ | - |
| $\tilde{R}_{Tot}^{(2,3)}$ | $85.18 \pm 0.23$ | $86.8 \pm 0.28$ | $86.9 \pm 0.29$ | - |
| $\tilde{R}_{Tot}^{(2,2)}$ | $85.5 \pm 0.23$ | $86.8 \pm 0.28$ | $86.8 \pm 0.28$ | $87.23 \pm 0.35$ |
| $\tilde{R}_{Tot}^{(2,1)}$ | $85.58 \pm 0.22$ | $86.4 \pm 0.32$ | $86.92 \pm 0.3$ | $86.95 \pm 0.31$ |
| $\tilde{R}_{Tot}^{(1,3)}$ | $85.18 \pm 0.23$ | $86.8 \pm 0.28$ | $86.9 \pm 0.29$ | - |
| $\tilde{R}_{Tot}^{(1,2)}$ | $85.5 \pm 0.23$ | $86.8 \pm 0.28$ | $86.8 \pm 0.28$ | $87.23 \pm 0.35$ |
| $\tilde{R}_{Tot}^{(1,1)}$ | $85.58 \pm 0.22$ | $86.4 \pm 0.32$ | $86.92 \pm 0.3$ | $86.95 \pm 0.31$ |
| $E(x; \mathbb{E}_\theta f_\theta)$ | $83.34 \pm 0.23$ | - | - | - |
| $\mathbb{E}_\theta E(x; f_\theta)$ | $82.4 \pm 0.24$ | - | - | - |

## L  DERIVATION OF CENTRAL PREDICTIONS

Central prediction is defined as $\bar{\eta} = \arg\min_z \mathbb{E}_\theta D_G(z\|\eta_\theta)$. For different instantiations of $G$, there will be different central predictions. In what follows, we will abuse the subscription $\theta$, and simply write $\mathbb{E}$ and $\eta$, assuming expectation with respect to $\theta$.

Recall, that Bregman divergence is:
$$D_G(z\|\eta) = G(z) - G(\eta) - \langle G'(\eta)\,,\, z - \eta \rangle.$$

### L.1  LOGSCORE

$$G(\eta) = \sum_{k=1}^{K} \eta_k \log \eta_k,$$
$$G'(\eta)_k = 1 + \log \eta_k,$$

$$\bar{\eta} = \arg\min_z \Big[\sum_k z_k \log z_k - \mathbb{E}\sum_k \eta_k \log \eta_k - \langle 1 + \mathbb{E}\log\eta\,,\, z \rangle + \mathbb{E}\langle 1 + \log\eta\,,\, \eta \rangle \Big] =$$

$$\arg\min_z \Big[\sum_k z_k \log z_k - \mathbb{E}\sum_k \eta_k \log \eta_k - \sum_k z_k \log\big(\exp\mathbb{E}\log\eta\big)_k + \mathbb{E}\sum_k \eta_k \log \eta_k \Big] =$$

$$\arg\min_z \Big[\sum_k z_k \log z_k - \sum_k z_k \log\big(\exp\mathbb{E}\log\eta\big)_k \Big] =$$

$$\arg\min_z \Big[\sum_k z_k \log z_k - \sum_k z_k \log\big(\exp\mathbb{E}\log\eta\big)_k -$$

$$\sum_k z_k \sum_{k'} \log\big(\exp\mathbb{E}\log\eta\big)_{k'} + \sum_k z_k \sum_{k'} \log\big(\exp\mathbb{E}\log\eta\big)_{k'} \Big] =$$

$$\arg\min_z \Big[\sum_k z_k \log z_k - \sum_k z_k \log \frac{\big(\exp\mathbb{E}\log\eta\big)_k}{\sum_{k'}\big(\exp\mathbb{E}\log\eta\big)_{k'}} \Big] = \arg\min_z \Big[\mathrm{KL}(z\|\frac{\big(\exp\mathbb{E}\log\eta\big)_k}{\sum_{k'}\big(\exp\mathbb{E}\log\eta\big)_{k'}})\Big],$$

hence, $\bar{\eta}_k = \dfrac{\big(\exp\mathbb{E}\log\eta\big)_k}{\sum_{k'}\big(\exp\mathbb{E}\log\eta\big)_{k'}}$.

### L.2  BRIER SCORE

$$G(\eta) = -\sum_{k=1}^{K} \eta_k(1 - \eta_k),$$
$$G'(\eta)_k = 2\eta_k - 1,$$

$$\bar{\eta} = \arg\min_z \Big[-\sum_k z_k(1 - z_k) + \mathbb{E}\sum_k \eta_k(1 - \eta_k) - \langle 2\mathbb{E}\eta - 1\,,\, z \rangle + \mathbb{E}\langle 2\eta - 1\,,\, \eta \rangle \Big] =$$

$$\arg\min_z \Big[-\sum_k z_k(1 - z_k) + \mathbb{E}\sum_k \eta_k(1 - \eta_k) - 2\sum_k z_k\mathbb{E}\eta_k + 2\mathbb{E}\sum_k \eta_k^2 \Big] =$$

$$\arg\min_z \Big[\sum_k^K z_k^2 - \mathbb{E}\sum_k \eta_k^2 - 2\sum_k z_k\mathbb{E}\eta_k + 2\mathbb{E}\sum_k \eta_k^2 \Big] =$$

$$= \arg\min_z \Big[\sum_k z_k^2 - 2\sum_k z_k\mathbb{E}\eta_k + \mathbb{E}\sum_k \eta_k^2 \Big] =$$

$$\arg\min_z \Big[\sum_k z_k^2 - 2\sum_k z_k\mathbb{E}\eta_k + \mathbb{E}\sum_k \eta_k^2 + \sum_k (\mathbb{E}\eta_k)^2 - \sum_k (\mathbb{E}\eta_k)^2 \Big] =$$

$$\arg\min_z \Big[\|z - \mathbb{E}\eta\|_2^2 + \mathbb{E}\sum_k \eta_k^2 - \sum_k (\mathbb{E}\eta_k)^2 \Big] = \arg\min_z \Big[\|z - \mathbb{E}\eta\|_2^2 \Big],$$

hence, $\bar{\eta} = \mathbb{E}\eta$.

### L.3 ZERO-ONE SCORE

$$G(\eta) = \max_k \eta_k - 1,$$

$$G'(\eta)_k = \mathbb{I}[k = \arg\max_j \eta_j],$$

$$\bar{\eta} = \arg\min_z \mathbb{E}\Big[\max_k z_k - 1 - \max_k \eta_k + 1 - \langle \mathbb{I}[\arg\max_j \eta_j], \, z - \eta\rangle\Big] =$$

$$\arg\min_z \mathbb{E}\Big[\max_k z_k - \max_k \eta_k - (z - \eta)_{\arg\max_j \eta_j}\Big] = \arg\min_z \mathbb{E}\Big[\max_k z_k - z_{\arg\max_j \eta_j}\Big] =$$

$$\arg\min_z \Big[\max_k z_k - \langle z, \, \mathbb{E}\mathbb{I}[\arg\max_j \eta_j]\rangle\Big] = \arg\min_z \Big[\max_k z_k - \langle z, \, \tilde{p}\rangle\Big] =$$

$$\arg\min_z \langle z, \, \mathbb{I}[\arg\max_j z_j] - \tilde{p}\rangle = 0,$$

where $\tilde{p} = \mathbb{E}\mathbb{I}[\arg\max_j \eta_j]$ are empirical frequencies of class labels, predicted by models, parametrized by samples from $p(\theta \mid D_{tr})$.

We can see, that minimum is always reached with $\bar{\eta} = \text{Uniform}(K)$, where Uniform(K) is a uniform categorical distribution over $K$ classes. Please note, that there might be infinitely many solutions, when at least one of the components of $\tilde{p}$ is 0.

Let us assume that there are no zero components, hence uniform distribution will be the only solution.

However, uniform distribution plug-in violates general decomposition in equation (11).

This might be due to the requirement, that $G$ must be a strictly convex, while $G(\eta) = \max_k \eta_k - 1$ is only convex, and effectively operates with the single (maximal) component of the categorical distribution $\eta$, while undetermine others.

### L.4 SPHERICAL SCORE

$$G(\eta) = \|\eta\|_2 - 1,$$

$$G'(\eta)_k = \frac{\eta_k}{\|\eta\|_2}.$$

$$\bar{\eta} = \arg\min_z \mathbb{E}\Big[\|z\|_2 - \|\eta\|_2 + \langle \frac{\eta}{\|\eta\|_2}, \, \eta - z\rangle\Big]$$

$$= \arg\min_z \mathbb{E}\Big[\|z\|_2 - \langle \frac{\eta}{\|\eta\|_2}, \, z\rangle\Big]$$

$$= \arg\min_z \|z\|_2 - \langle z, \, \mathbb{E}\Big[\frac{\eta}{\|\eta\|_2}\Big]\rangle.$$

Let us introduce following notation:

$$f(z, \eta) = \arg\min_z \|z\|_2 - \langle z, \, \mathbb{E}\Big[\frac{\eta}{\|\eta\|_2}\Big]\rangle,$$

$$\eta_E = \mathbb{E}\Big[\frac{\eta}{\|\eta\|_2}\Big],$$

$$x_0 = \Big(\frac{1}{K}, \dots, \frac{1}{K}\Big) \in \mathbb{R}^K,$$

$$x_\| := x_\| \in \mathbb{R}^K \text{ s.t. } x_\| \| x_0,$$

$$x_\perp := x_\perp \in \mathbb{R}^K \text{ s.t. } x_\perp \perp x_0, \|x_\perp\| = 1,$$

$$a := a \in \mathbb{R}^K \text{ s.t. } a \perp x_0, \, a \perp x_\perp, \|a\|_2 = 1.$$

Then we can say, that there exists $k_\eta, k, k_a$ such that:

$$\eta_E = x_\parallel + k_\eta x_\perp, \; z = x_0 + kx_\perp + k_a a,$$

$$f(k_\eta, k, k_a) = \|x_0 + kx_\perp + k_a a\|_2 - \langle x_0 + kx_\perp + k_a a \,,\, x_\parallel + k_\eta x_\perp \rangle$$

$$= \sqrt{\sum_i (x_0)_i^2 + k^2 \sum_i (x_\perp)_i^2 + k_a^2 \sum_i a_i^2} - (k_\eta k + \langle x_0 \,,\, x_\parallel \rangle)$$

$$= \sqrt{\sum_i (x_0)_i^2 + k^2 + k_a^2} - k_\eta k - \langle x_0 \,,\, x_\parallel \rangle.$$

Let us takes derivatives of $f(\cdot)$ w.r.t $k, k_a$ to find values, that minimize it:

$$\frac{df(k_\eta, k, k_a)}{dk_a} = \frac{k_a}{\sqrt{\sum_i (x_0)_i^2 + k^2 + k_a^2}} = 0,$$

$$\sum_i (x_0)_i^2 + k^2 \neq 0 \implies k_a = 0,$$

$$\frac{df(k_\eta, k, k_a)}{dk} = \frac{k}{\sqrt{\sum_i (x_0)_i^2 + k^2}} - k_\eta = 0,$$

$$k^2 = k_\eta^2 (\sum_i (x_0)_i^2 + k^2),$$

$$k^2(1 - k_\eta^2) = k_\eta^2 \|x_0\|_2^2 \implies k = \frac{k_\eta \|x_0\|_2}{\sqrt{1 - k_\eta^2}},$$

$$x_\parallel = \langle \eta_E \,,\, x_0 \rangle \frac{x_0}{\|x_0\|_2^2},$$

$$k_\eta = k_\eta \|x_\perp\|_2 = \|\eta_E - x_\parallel\|_2^2 = \|\eta_E - \langle \eta_E \,,\, x_0 \rangle \frac{x_0}{\|x_0\|_2^2}\|_2.$$

Finally we have:

$$z = x_0 + kx_\perp + k_a a = x_0 + \frac{k}{k_\eta}(\eta_E - x_\parallel) = x_0 + \frac{\eta_E - x_\parallel}{\sqrt{1 - k_\eta^2}}\|x_0\|_2$$

$$= x_0 + \frac{\eta_E - \langle \eta_E \,,\, x_0 \rangle \frac{x_0}{\|x_0\|_2^2}}{\sqrt{1 - \|\eta_E - \langle \eta_E \,,\, x_0 \rangle \frac{x_0}{\|x_0\|_2^2}\|_2^2}}\|x_0\|_2.$$

Setting $n = \frac{x_0}{\|x_0\|}$ and $m = \eta_E - \langle \eta_E \,,\, x_0 \rangle \frac{x_0}{\|x_0\|_2^2}$ we conclude:

$$z = \|x_0\|_2 \Big[n + \frac{m}{\sqrt{1 - \|m\|_2^2}}\Big].$$

## M  TOY EXAMPLE

In this section, we consider a specific toy example, in which uncertainty measures can be computed in closed form.

Specifically, we consider the problem of binary classification, where the likelihood (predictive model) is simply a Bernoulli distribution, and the prior distribution over model parameter (the probability of success) is Beta distribution. Therefore, prior and likelihood are conjugate, and the posterior is Beta distribution as well.

**Expected Pairwise Kullback–Leibler (EPKL).**  Here, we want to estimate

$$\mathbb{E}_{\theta^*}\mathbb{E}_{\theta}KL(\theta^*\|\theta) =$$
$$\mathbb{E}_{\theta^*}\big[\theta^*\log\theta^*\big] - \mathbb{E}_{\theta^*}\big[\theta^*\big]\mathbb{E}_{\theta}\big[\log\theta\big] + \mathbb{E}_{\theta^*}\big[(1-\theta^*)\log(1-\theta^*)\big] - \mathbb{E}_{\theta^*}\big[(1-\theta^*)\big]\mathbb{E}_{\theta}\big[\log(1-\theta)\big].$$

We assume that $p(\theta^* \mid D) = \text{Beta}(\alpha^*, \beta^*)$, and $p(\theta \mid D) = \text{Beta}(\alpha, \beta)$. Each of the components can be computed analytically:

1. $\mathbb{E}_{\theta^*}\big[\theta^*\log\theta^*\big] = \frac{\alpha^*}{\alpha^*+\beta^*}\big[\psi(\alpha^*+1) - \psi(\alpha^*+\beta^*+1)\big]$;

2. $\mathbb{E}_{\theta^*}\theta^* = \frac{\alpha^*}{\alpha^*+\beta^*}$;

3. $\mathbb{E}_{\theta}\log\theta = \psi(\alpha) - \psi(\alpha+\beta)$;

4. $\mathbb{E}_{\theta^*}\big[(1-\theta^*)\log(1-\theta^*)\big] = \frac{\beta^*}{\alpha^*+\beta^*}\big[\psi(\beta^*+1) - \psi(\alpha^*+\beta^*+1)\big]$;

5. $\mathbb{E}_{\theta^*}(1-\theta^*) = \frac{\beta^*}{\alpha^*+\beta^*}$;

6. $\mathbb{E}_{\theta}\big[\log(1-\theta)\big] = \psi(\beta) - \psi(\alpha+\beta)$.

If we assume, that $\alpha^* = \alpha$ and $\beta^* = \beta$ (the setup we considered in the main part), then one can show, using these six equations that

$$\text{EPKL} = \frac{1}{\alpha+\beta}.$$

Note, that this assumption (on the equal parameters) was done for simplicity. In general case, one can indeed consider different Bayesian models for the approximation of the ground truth distribution and for the approximation of the prediction. But since it is done at the cost of training a separate Bayesian model, we believe it might be not of a big practical value.

**Mutual Information (MI).**  Following the derivations from the main part, MI can be represented as:

$$\mathbb{E}_{\theta}KL(\theta^*\|\mathbb{E}_{\theta}\theta) =$$
$$\mathbb{E}_{\theta^*}\big[\theta^*\log\theta^*\big] - \mathbb{E}_{\theta^*}\big[\theta^*\big]\log\mathbb{E}_{\theta}\theta + \mathbb{E}_{\theta^*}\big[(1-\theta^*)\log(1-\theta^*)\big] - \mathbb{E}_{\theta^*}\big[(1-\theta^*)\big]\log(1-\mathbb{E}_{\theta}\theta).$$

Again, we use the same assumption on the equal parameters of the posterior distributions. With the equations from the above, we obtain:

$$\text{MI} = \frac{1}{\alpha+\beta} + \frac{\alpha}{\alpha+\beta}\big[\psi(\alpha)-\psi(\alpha+\beta)-\log\alpha\big] + \frac{\beta}{\alpha+\beta}\big[\psi(\beta)-\psi(\alpha+\beta)-\log\beta\big] + \log(\alpha+\beta).$$

**Reverse Mutual Information (RMI).**  As we showed in the main part, RMI can be presented as follows:

$$\mathbb{E}_{\theta}KL(\mathbb{E}_{\theta^*}\theta^*\|\theta) =$$
$$\mathbb{E}_{\theta^*}\big[\theta^*\big]\big[\log\mathbb{E}_{\theta^*}\theta^*\big] - \mathbb{E}_{\theta^*}\big[\theta^*\big]\mathbb{E}_{\theta}\big[\log\theta\big] + (1-\mathbb{E}_{\theta^*}\theta^*)\log(1-\mathbb{E}_{\theta^*}\theta^*) - (1-\mathbb{E}_{\theta^*}\theta^*)\mathbb{E}_{\theta}\big[\log(1-\theta)\big].$$

Following the same methodology as for previous cases, we obtain:

$$\text{RMI} = \frac{\alpha}{\alpha+\beta}\big(\log\alpha - \psi(\alpha) + \psi(\alpha+\beta)\big) + \frac{\beta}{\alpha+\beta}\big(\log\beta - \psi(\beta) + \psi(\alpha+\beta)\big) - \log(\alpha+\beta).$$

It is easy to see, that EPKL = MI + RMI (which is consistent with the general result that $\tilde{\text{R}}_{\text{Exc}}^{(1,1)} = \tilde{\text{R}}_{\text{Exc}}^{(1,2)} + \tilde{\text{R}}_{\text{Exc}}^{(2,1)}$).

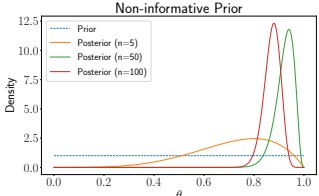 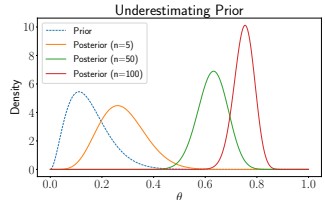 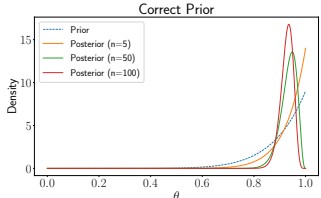

(a) Uniform prior. $\alpha = \beta = 1$. (b) Underestimating prior. $\alpha = 3, \beta = 17$. (c) Correct prior. $\alpha = 9, \beta = 1$.

Figure 4: Different shapes of the posterior distributions.

**Expected Pairwise Brier Score (EPBS).**  In this case, we would like to estimate the following quantity:

$$\mathbb{E}_{\theta^*}\mathbb{E}(\theta^* - \theta)^2 = \mathbb{E}_{\theta^*}\left[\theta^*\right]^2 - 2\mathbb{E}_{\theta^*}\left[\theta^*\right]\mathbb{E}_{\theta}\left[\theta\right] + \mathbb{E}_{\theta}\left[\theta\right]^2.$$

If we assume, that the parameters of the posterior distributions are equal, then the whole equation above reduces to the double variance of $\theta$. In case of Beta distribution, it equals to:

$$\text{EPBS} = 2var\theta = \frac{2\alpha\beta}{(\alpha + \beta)^2(\alpha + \beta + 1)}.$$

Note, that the posterior update rule for this simple Beta-Bernoulli model is:

$$\alpha = \alpha_{prior} + x,$$

and

$$\beta = \beta_{prior} + n - x,$$

where by subscript "prior" we highlight the parameters of the prior distribution, $x$ is the number of successes in observed training dataset, and $n$ is the total number of observations.

Therefore we see, that given larger training dataset, the estimates of epistemic uncertainty will shrink. This is the property we typically want to have from the epistemic uncertainty estimates – the bigger the dataset, the less uncertainty.

However, in case of the prior misspecification (will be discussed below), it leads to certain problems. If the prior parameters $\alpha_{prior}$ and $\beta_{prior}$ are already large (indicating a highly concentrated prior), the posterior will also be highly concentrated, regardless of the data. This can lead to an underestimation of epistemic uncertainty because the model appears more certain than it should be.

### M.1 VARIOUS DISTRIBUTION SHAPES

In this section, we explore different choices of the parameters and demonstrate, how the resulting posterior distribution will look like, given different prior and different sizes of training datasets. We show the resulting plots at the Figure 4. Note, that as a ground-truth we used $\theta^* = 0.9$. We see, that given enough data, the mass of the posterior is concentrated around the correct value of the parameter, regardless of the prior misspecification.

### M.2 PRIOR MISSPECIFICATION

In this section, we demonstrate, how the metrics of epistemic uncertainty behave, given different (possibly misspecified) priors. We present the results in Figure 5. We see, that as we discussed above, these metrics depend on the values if $\alpha$ and $\beta$. Therefore, the more these values are, the less is the estimate of epistemic uncertainty. Since uniform prior contains minimal evidence, it demonstrates maximal uncertainty, and therefor less vulnerable for prior misspecification. Note, that in our experiments on deep ensembles, we considered uniform prior over parameters, therefore the effect of prior misspecification should be negligible.

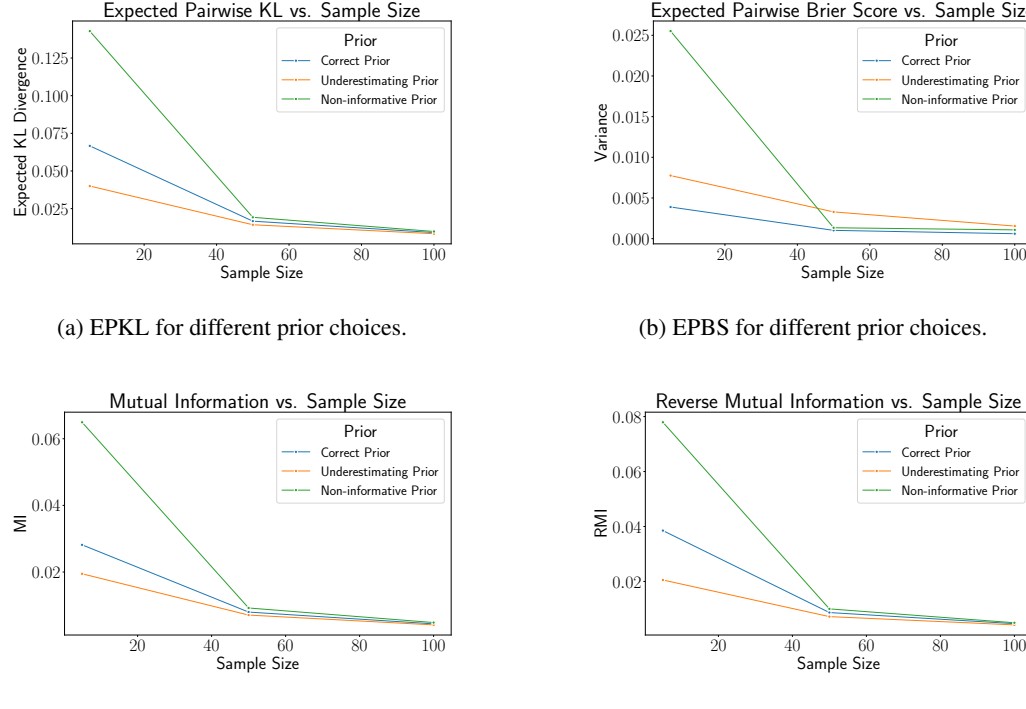

(a) EPKL for different prior choices.

(b) EPBS for different prior choices.

(c) MI for different prior choices.

(d) RMI for different prior choices.

Figure 5: Epistemic uncertainty metrics, given prior misspecification and different samples sizes.

## M.3 ERROR IN POSTERIOR APPROXIMATION

In this section, we consider the case of choosing wrong posterior distribution for inference. We know, that in our toy example, the correct family to consider for inference is Beta. However, we will misspecify the choice and consider Normal distribution instead. We use two approaches for inference. Namely, Laplace Approximation (approximates the posterior around the mode), and Moment Matching (matches the mean and variance of the Beta distribution with a Gaussian).

We present results in Figure 6. We considered uniform distribution as a prior, and we used training datasets of sizes 20 and 100. As expected, we see, that the given less data, errors in approximations are more severe. However, if a bigger dataset is provided, the approximation becomes more accurate.

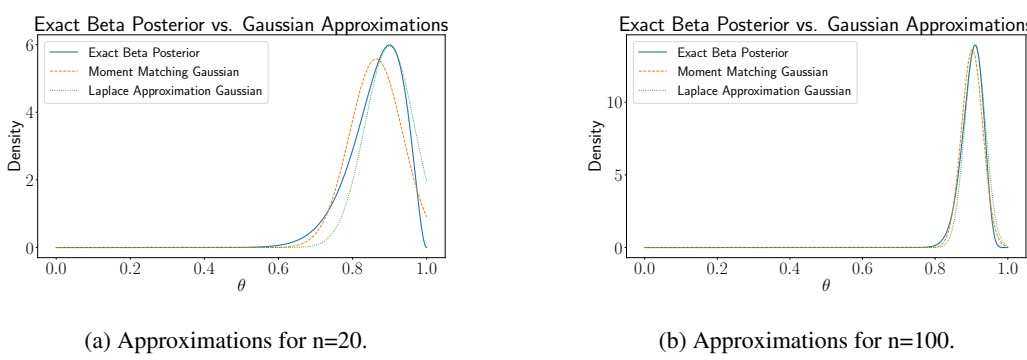

(a) Approximations for n=20.

(b) Approximations for n=100.

Figure 6: Resulting approximations given different sizes of training datasets.

