# OpenReview forum: "From Risk to Uncertainty: Generating Predictive Uncertainty Measures via Bayesian Estimation"
_ICLR.cc/2025/Conference — ICLR 2025 Poster_

### Official Review · Reviewer_Y7My · 2024-10-16

**Soundness:** 3
**Presentation:** 3
**Contribution:** 3
**Rating:** 8
**Confidence:** 4

**Summary:**

The authors use a decomposition of statistical pointwise risk into components, associated with aleatoric and epistemic uncertainties. Together with Bayesian methods, applied as an approximation, they build a framework that allows to generate different predictive uncertainty measures. Experiments are shown that support the theoretical claims.

**Strengths:**

This is a nice and interesting paper that addresses a cogent problem in the literature, that is, the existence of many aleatoric and epistemic uncertainty measures based on the total risk decomposition, and how they're related to one another.

**Weaknesses:**

Taking a Bayesian route has a fundamental drawback, that was highlighted e.g. by Hullermeier and Waegeman (2021, already cited in the paper) and Caprio et al. (2024, Credal Bayesian Deep Learning). Consider the *posterior predictive distribution*,

$$p(\tilde y \mid \tilde x , D)=\int_\Theta p(\tilde y \mid \tilde x, \theta) p(\theta \mid D) \text{d}\theta = \mathbb{E}_{\theta \sim p(\theta \mid D)} [p(\tilde y \mid \tilde x, \theta)],$$

where $p(\tilde y \mid \tilde x, \theta)$ is the model distribution, and $p(\theta \mid D)$ is the parameter posterior. Posterior predictive $p(\tilde y \mid \tilde x , D)$ tells us "how likely" output $\tilde y$ is to be the "correct one" for input $\tilde x$, given the knowledge encapsulated in the data $D$ we collected, which enters the computation via the posterior probability $p(\theta \mid D)$. Oftentimes, scholars claim that, in a Bayesian setting, the distribution on the parameters $\theta$ captures (or is linked to) the epistemic uncertainty (EU) faced by the agent. This is a somehow agreeable premise, akin to a second-order distribution reasoning. If we accept this assertion, though, we see how EU at the predictive level is not quantifiable any more, since it gets washed away by taking the expectation $\mathbb{E}_{\theta \sim p(\theta \mid D)} [\cdot ]$.

This conceptual problem is relevant for this paper, especially because it is related to the ideas of Bayesian averaging of risk and Central label. I'd like the authors to add a discussion on this matter.

**Questions:**

See Weaknesses. The authors should cite https://arxiv.org/abs/2302.09656 and https://link.springer.com/chapter/10.1007/978-3-031-57963-9_1#:~:text=In%20their%20seminal%201990%20paper,bound%20to%20hold%20with%20equality when discussing about the mentioned problem (and possibly in the related work section). They may also cite other approaches that may be considered in the future, such as credal sets ones, studied by Yusuf Sale, Eyke Hülleremeier, Paul Hofman, Michele Caprio, Viktor Bengs, Sebastien Desterke, Fabio Cuzzolin, Thierry Denoeux, Alessio Benavoli, and Cassio de Campos.

Also, I think there's a typo in line 198: shouldn't it be $\eta_{\theta \mid D_{tr}}$ instead of $\eta_{\theta}\mid D_{tr}$?

---

> ### Author Response · Authors · 2024-11-14
> **Author's response**
>
> Thank you for your thoughtful feedback and for taking the time to review our work. We appreciate your comments and overall positive evaluation!
>
> ---
>
> > “we see how the EU at the predictive level is not quantifiable any more, since it gets washed away by taking the expectation E_\theta ..”
>
> This is indeed a very interesting and important remark!
>
> Indeed, providing a point estimate, instead of considering all possible models from the model posterior, will “vanish out” all the epistemic uncertainty.
>
> This vanishing out is indeed present in some of the instances within our framework, namely MI, RMI, etc., and those, with constant predictions (e.g. R_exc(3,2) and R_exc(2, 3)). However, some instances, namely EPBD, do not suffer from this drawback. Therefore, within our framework, there are measures that avoid this integration, potentially capturing epistemic uncertainty more carefully.
> However, in our experiments, it is not that concrete, that it is always better.
>
> But nevertheless, we agree that this is a very important conceptual thing that we need to incorporate, and we will do it in a revision.
>
>
> __Literature on credal sets__
>
> We thank the reviewer for pointing this out. We will add the discussion of credal based sets uncertainty in related work and we will acknowledge the papers you mentioned. Thank you!
>
>
> __On misprint__
>
> Thank you for pointing this out! You are right, we will fix it.
>
> ---
>
> If there are any additional areas where we could provide further clarity, strengthen our results, or improve the paper to better meet your expectations—and possibly enhance your evaluation—we would be most grateful for your suggestions!

---

> ### Author Response · Authors · 2024-11-22
>
> Dear Reviewer Y7My,
>
> We sincerely appreciate your initial positive feedback on our manuscript.
>
> As we approach the end of the author-reviewer discussion period, we wanted to follow up regarding our revisions. In response to your suggestions, we have updated the manuscript by adding a discussion on credal set-based uncertainties in the related work section and have cited the relevant papers. We have also included a note on averaging out uncertainty.
>
> Given the borderline scores and the importance of your input to the final decision, we kindly ask if you have any remaining concerns or additional feedback. Your insights are valuable to us, and we are committed to addressing any issues you might have.
>
> Thank you for your time and consideration.
>
> Best regards,
>
> The Authors

---

> > ### Comment · Reviewer_Y7My · 2024-11-22
> > **Thank you**
> >
> > Dear Authors,
> >
> > Thank you very much for the fruitful discussion. I have increased my score accordingly.

---

### Official Review · Reviewer_cjsf · 2024-10-18

**Soundness:** 3
**Presentation:** 3
**Contribution:** 2
**Rating:** 8
**Confidence:** 4

**Summary:**

The authors propose a family of uncertainty measures based on point-wise risks which allows to decompose the uncertainty into aleatoric (Bayes risk) and epistemic (Excess risk) parts. Plugging-in proper scoring rules into the derived uncertainty measures, they obtain expressions for computing the Bayes and Excess risk for the specific case of proper scoring rules. Finally, the authors compare three different methods for estimating the risks based on the Bayesian formulation, which they allows to derive many of the existing uncertainty measures as special cases.

**Strengths:**

Many parts of the work have appeared in some form before, but combining them is a novel idea. For example, generating the uncertainty measure from the point-wise risks has been explored in [1] and [2] as the authors note, but also in [3]. Proper scoring rules were explored in the context of the uncertainty quantification in [4]. Finally, the various Bayesian risk estimation techniques have existed in the literature for a long time. However, the authors do a good job connecting these pieces and showing how various uncertainty measures naturally arise and correspond to different ways of estimating the risk for a proper scoring rule. The authors supplement this with a deeper investigation of the relationships between the estimates and exploring which estimates are better under specific conditions.

[1] Kotelevskii, Nikita, et al. "Nonparametric uncertainty quantification for single deterministic neural network." Advances in Neural Information Processing Systems 35 (2022): 36308-36323.
[2] Lahlou, Salem, et al. "Deup: Direct epistemic uncertainty prediction." Transactions on Machine Learning (2021).
[3] Liu, Jeremiah, et al. "Accurate uncertainty estimation and decomposition in ensemble learning." Advances in neural information processing systems 32 (2019).
[4] Gruber, Sebastian, and Florian Buettner. "Better uncertainty calibration via proper scores for classification and beyond." Advances in Neural Information Processing Systems 35 (2022): 8618-8632.

**Weaknesses:**

While the work as a whole appears sound, I have a few concerns:
1. How are the proposed risks affected by the model misspecification, e.g., prior misspecification of $p(\theta)$? I think since the method is Bayesian in nature, it's important to outline the assumptions and highlight possible shortcomings.
2. No simulations. Assuming, we know the true $\eta$, it would be interesting to see how the different estimators of the risk fare under (1) various distribution shapes (2) misspecification (of prior) (3) error from approximate posterior (if full posterior is not available).
3. Experimental section is narrowly focused and not well-motivated. The authors devote their experiments to testing whether the aleotoric and epistemic uncertainties are captured well. For this, the authors compute the proposed Total, Bayes, and Excess risks and classify the samples into out-of-distribution and misclassified based on these values. They then test the accuracy of this classification. Since Excess risk is connected to the epistemic uncertainty that is connected to the out-of-distribution classification, the out-of-distribution classification accuracy must be better based on the Excess risk. Similarly for the Bayes risk and the accuracy of identifying misclassification, both of which are connected to the aleatoric uncertainty. First, these experiments hinge on the assumption that indeed the misclassification and out-of-distribution are related to the aleatoric and epistemic uncertainties -- a largely untestable assumption. Second, from the tables, I see very little difference between using the Total, Bayes, and Excess risks across all datasets and tasks: the differences while being statistically significant appear almost negligible, which raises questions whether the suggested measures and/or experiments are meaningful.
4. No comparison with other UQ measures. The authors do not compare their proposed UQ measures with the existing approaches, which leaves the question of why they should be used in practice.
5. (Minor gripe)  Limited practical utility. It appears to compute any of the proposed uncertainty measures, one needs an access to the posterior $p(\theta|X,Y)$. Ultimately, this limits the set of methods to which the proposed approach could be applied.

**Questions:**

1. In Section 4.1 and Appendix E, the authors arrive at the conclusion that choosing the best estimate is often impossible in practice. namely, in Section 4.1, the authors say that exact relationship of the central estimate, $r(\bar\eta)$ with the other ones is not known so it's not clear which choice of the estimate for Bayes risk is best; while in Appendix E, the authors conclude that knowing which estimate to choose for the Excess is impossible to know apriori. Similarly, there is little guidance on the choice of the proper scoring rules. Overall, this leaves me with a 4 (proper scoring rules) * 9 (estimates) possible ways to get the Excess risk. Similarly, for the Bayes risk, we have another 12 possible choices. Are there any general or maybe domain-specific advice on how to choose the best UQ measure from among the proposed ones? Is it possible to have results on the error rate of each estimate for some common distribution families? In Appendix E, the authors note that none of the estimated yield a lower / upper on the Excess risk, which raises a question if it's possible to derive such an estimated bound?
2. I think there are important points missing from the experiments description (Section 6). For example, how are the UQ measures computed given the outputs of the deep ensemble model? And more generally, given a model that only allows access samples from the posterior, are we resorting to a regular Monte Carlo?
3. In Section 6, what is meant by 5 groups of ensembles? Do all of them have the same architecture? I think the wording here is confusing.

---

> ### Author Response · Authors · 2024-11-15
> **Author's response**
>
> We want to thank the reviewer for their thorough review!
>
> __0. Missing reference [3]. (in Strenghts)__
>
> We thank the reviewer for sharing the interesting paper we did not know. We will incorporate appropriate reference to this paper.

---

> ### Author Response · Authors · 2024-11-15
> **Weaknesses 1&2  [1/2]**
>
> __1. How are the proposed risks affected by the model misspecification, e.g., prior misspecification of p(\theta)? I think since the method is Bayesian in nature, it's important to outline the assumptions and highlight possible shortcomings.__
>
> __2. No simulations. Assuming, we know the true \eta, it would be interesting to see how the different estimators of the risk fare under (1) various distribution shapes (2) misspecification (of prior) (3) error from approximate posterior (if full posterior is not available).__
>
>
> _Summary for Weakness 1:_
> We acknowledge that prior misspecification can affect our proposed risks, especially in Bayesian methods with limited data. Our simulations show that using an uninformative prior helps prevent underestimating epistemic uncertainty, and we discuss how model misspecification impacts our uncertainty estimates.
>
> _Summary for Weakness 2:_
> We agree that simulations would provide valuable insights into how our risk estimators perform under various conditions. We have conducted simulations examining different distribution shapes, prior misspecification, and errors from approximate posteriors, and we will include these results in the revised paper.
>
> Below, we will provide detailed answers to both these weaknesses.
>
> ----
>
> Prior selection is indeed crucial in Bayesian methods. The concern about prior misspecification is most significant when we have limited data. With enough data, the likelihood tends to dominate the posterior, reducing the influence of the prior. However, with sparse data, a misspecified prior can lead to inaccurate estimates of uncertainty.
>
> To illustrate this, we conducted a simple simulation where we can compute the uncertainty measures analytically. Specifically, we considered our framework instantiated with $R_{exc}^{(1, 1)}$ and used two scoring functions: the Log Score and the Brier Score. We set up a binary classification problem with a Bernoulli likelihood and a Beta prior. In this setup, the posterior distribution is also Beta, and its parameters can be easily updated based on observed data.
>
> It can be shown (we will add these derivations to the paper), that these epistemic uncertainty (EU) measures will be as follows:
>
> - EPKL (G=Log score) = $\frac{1}{\alpha + \beta}$
> - EPBS (G=Brier score) = $\frac{2\alpha \beta}{(\alpha + \beta)^2 (\alpha + \beta + 1)}$ (basically the double variance of a Beta rv).
>
> Here, alpha and beta are the parameters of the posterior.
>
> From this example, we observe that the EU estimate decreases as $\alpha$ and $\beta$ increases. This means that as we add more data (increasing $\alpha$ and $\beta$), the epistemic uncertainty decreases, which is expected.
>
> The posterior parameters are updated using:
>
> $\alpha = \alpha_{\text{prior}}​ + x$
>
> $\beta = \beta_{\text{prior}} + n - x,$
>
> where x is the number of successes, and n is the total number of observations.
>
> If the prior parameters $\alpha_{\text{prior}}$​ and $\beta_{\text{prior}}$ are already large (indicating a highly concentrated prior), the posterior will also be highly concentrated, regardless of the data. This can lead to an underestimation of epistemic uncertainty because the model appears more certain than it should be.
>
> In our simulations (which we will include in the revised paper), we found that using an uninformative prior (e.g., a uniform prior $\alpha_{\text{prior}} = \beta_{\text{prior}} = 1$) results in EU estimates that reflect the reasonable level of uncertainty, especially when data is limited.
>
> In our experiments with deep ensembles, we effectively used a uniform (improper) prior, which is uninformative. Additionally, our datasets were large (over 10,000 training examples), so the influence of prior misspecification is minimal in our case. However, we agree that in general, careful selection of the prior is important to avoid underestimating epistemic uncertainty.
>
> ---
>
> Now regarding the specific simulations.
>
> __(1) various distribution shapes__
>
> We will add results and plots to the paper to show how different posterior distribution shapes affect the EU estimates.
>
> - _Less Concentrated Posterior (Small $\alpha$ and $\beta$):_ The posterior is wide, indicating high uncertainty about the model parameters. The EU estimate is higher in this case.
> - _More Concentrated Posterior (Large $\alpha$ and $\beta$):_ The posterior is narrow, indicating confidence in the model parameters. The EU estimate is lower.
>
> This relationship makes intuitive sense because the Expected Pairwise Bregman Divergence (EPBD), which $R_{exc}^{(1, 1)}$ represents, measures disagreement among model predictions. A wider posterior implies more disagreement.
>
> In complex models like deep neural networks, the mapping from parameters to predictions is more complicated due to symmetries and non-linearities. While exact behavior is harder to predict, the general intuition remains: a wider posterior over model parameters should lead to higher epistemic uncertainty.

---

> ### Author Response · Authors · 2024-11-15
> **Weaknesses 1&2 [2/2]**
>
> __(2) misspecification (of prior)__
>
> As discussed, a misspecified prior (especially one that is too informative) can lead to underestimating epistemic uncertainty, particularly with limited data.
>
> - _Overly Informative Prior:_ If the prior is too concentrated (large $\alpha_{\text{prior}}$ and $\beta_{\text{prior}}$​), the posterior remains concentrated even if data suggests higher uncertainty.
> - _Uninformative Prior:_ Using a uniform prior allows the data to have more influence on the posterior, leading to more accurate EU estimates.
>
> We will include plots showing how different priors affect the EU estimates across varying data sizes.
>
>
> __(3) error from approximate posterior (if full posterior is not available).__
>
> This is an important practical consideration. Often, we cannot compute the full posterior for complex models and must rely on approximations.
>
> Using our Bernoulli-Beta example, we approximated the Beta posterior with a Normal (therefore not the right one) distribution using:
> - _Laplace Approximation:_ Approximates the posterior around the mode using a Gaussian distribution.
> - _Moment Matching:_ Matches the mean and variance of the Beta distribution with a Gaussian.
>
> Findings:
> - _With Sufficient Data:_ The approximate posteriors closely match the true posterior, and the EU estimates are similar.
> - _With Limited Data:_ The approximations deviate more from the true posterior, leading to differences in EU estimates. This can result in either overestimating or underestimating uncertainty.
>
> We will include comparisons of EU estimates using the true posterior and approximate posteriors to illustrate the impact of approximation errors.

---

> ### Author Response · Authors · 2024-11-15
>
> __3. Experimental section is narrowly focused and not well-motivated.__
> > First, these experiments hinge on the assumption that indeed the misclassification and out-of-distribution are related to the aleatoric and epistemic uncertainties -- a largely untestable assumption.
>
> > Second, from the tables, I see very little difference between using the Total, Bayes, and Excess risks across all datasets and tasks: the differences while being statistically significant appear almost negligible, which raises questions whether the suggested measures and/or experiments are meaningful.
>
> We thank the reviewer for raising this important concern.
>
> The concern consists of two parts, and we will answer them one by one.
>
> In fact, it is a common trend in literature to consider epistemic uncertainty to capture out-of-distribution samples, and aleatoric/total uncertainty to capture misclassification.
>
> Nevertheless, as the reviewer emphasized, the intuition is somewhat not explained in literature and only assumed. For this very reason, to clarify this intuition, we incorporated a dedicated section (see Appendix A), that explains why one should expect this uncertainty decomposition to be useful for certain tasks. Please, let us know, if the discussion in Appendix addresses your concern?
>
> We are more than willing to elaborate on aspects that still might appear not clear.
>
> ----
>
> Regarding the second part of the concern. It is true that overall, the considered measures of uncertainty (aleatoric, epistemic, and total) perform well and close to each other. However (see answer to question 1), there are important differences in certain cases. Note, that the framework generalizes existing measures of uncertainty, including those widely used in practice. Therefore, this close performance is also shared for the metrics widely used in practice.
>
> > “...whether the suggested measures and/or experiments are meaningful”
> this question is very good, but this is not the limitation or drawback of the proposed framework. Rather, it is the general problem in uncertainty quantification.
>
> Both, aleatoric uncertainty (solely related to the ground-truth distribution), and epistemic uncertainty, are ultimately estimated from the model. Therefore, in practice, estimates of these two components eventually are related to each other. Therefore, there is an inherent (practical) correlation between the estimates. This degree of correlation, of course, is very important. But we believe, this is a completely different branch of research, that could be aimed to decorrelate these estimates from each other.
>
> One of the remedies (that, however, limits practical scope) is to train two separate Bayesian models, one for the actual prediction, and another to approximate the ground true distribution. In this situation, the estimates of AU and EU will be more disentangled, apparently at the price of training a separate Bayesian model.

---

> ### Author Response · Authors · 2024-11-15
>
> __4. No comparison with other UQ measures. The authors do not compare their proposed UQ measures with the existing approaches, which leaves the question of why they should be used in practice.__
>
> It is true that we did not consider other measures of UQ in this paper, as our main desire was theoretical/methodological -- that many known measures of uncertainty could be generalized. And then we compared different instances under different instantiations, which already covers many known UQ measures.
>
> Nevertheless, we are willing to consider other approaches, e.g. DDU, DUQ, and we will report on the results as soon as they are ready.

---

> ### Author Response · Authors · 2024-11-15
>
> __5. (Minor gripe) Limited practical utility. It appears to compute any of the proposed uncertainty measures, one needs an access to the posterior p(\theta \mid X, Y). Ultimately, this limits the set of methods to which the proposed approach could be applied.__
>
> This is true, that in general, one needs to have a Bayesian model to approximate these risks in our proposed methodology. One of the interesting directions, which could be a remedy, is to consider approximations that differ from Bayesians. For example the one as in [1], where authors considered a specific model, namely Nadaraya-Watson kernel regression, fitted over the embeddings of a neural net. Interestingly, their approach can be considered as a special case of our framework with proper scoring rules (they used zero-one score), but not with Bayesian approximation. In this case, one does not need to fit the Bayesian model. We believe that more non-Bayesian approaches (other than the one from [1]) can be developed starting from the proposed risk-based framework.
>
> [1] Kotelevskii, Nikita, et al. "Nonparametric uncertainty quantification for single deterministic neural network." Advances in Neural Information Processing Systems 35 (2022): 36308-36323.

---

> ### Author Response · Authors · 2024-11-15
>
> __Questions__
>
>
> > “Are there any general or maybe domain-specific advice on how to choose the best UQ measure from among the proposed ones?”
>
>
> Thank you for your excellent and important question!. We can offer several pieces of advice at different levels:
>
> _High Level (Choice between Bayes, Excess, or Total Risk):_
>
> The best UQ measure depends on the specific problem you are addressing.
>
>
> __Bayes/Total Risk Measures:__
> - _Hard OOD Detection:_ If your goal is to detect objects that are significantly different from the training data (hard-OOD detection), Bayes (which at first is a bit mysterious) or Total risk measures are generally more effective.
> - _Misclassification Detection:_ For tasks where identifying incorrect class assignments is crucial, such as finding misclassifications, Bayes or Total risk measures are preferable.
>
> __Excess Risk Measures:__
> - _Soft-OOD Detection:_ If you need to identify images that are blurry or "fuzzy" (referred to as soft-OOD detection), Excess risk-based measures are typically more suitable.
>
> We respectfully disagree with the notion that there is very little difference between these measures. For example:
> - In __Table 3__ of our paper, which focuses on OOD detection, Bayes and Total risk measures achieve AUROC scores of __83-84__ on hard-OOD datasets, while Excess risk measures score around __77-79__, with standard deviations of approximately 0.1. This shows a meaningful difference in performance.
> - In __Table 4__, addressing misclassification detection on datasets with significant label noise (such as CIFAR10N and CIFAR100N), the difference is even more pronounced:
> - - CIFAR10N: AUROC scores differ by about 10 between Bayes/Total risk and Excess risk estimates.
> - - CIFAR100N: The difference is about 7.
>
> This highlights the importance of selecting the appropriate measure based on the amount of aleatoric noise present. To better illustrate the usefulness of our measures in scenarios with sufficient aleatoric noise, we introduced these crafted datasets (CIFAR10N and CIFAR100N), as classical image classification datasets may not have enough inherent label noise.
>
> _Middle Level (Choice of $G$):_
>
> Our observations suggest that __Log Score-based measures__ are typically (though not always) a better choice for uncertainty quantification. This is particularly evident in:
> - Table 8 (please note: the right column represents the Spherical score, as mentioned in the caption). Here we used different loss functions and different (matching) instantiations of $G$. Please, take a look at the different in performance for Excess risks. Log Score based ones lead with significant margin.
> - Table 9. Here, we trained all the models with Log Score based instantiations (Cross Entropy loss), but when estimated uncertainty, used different choices of $G$. Again, for Excess risks with Log Score, results are way better.
>
> These tables focus on Excess risk-based measures for the CIFAR100 and TinyImageNet datasets. However, for the CIFAR10 dataset, the difference between scoring functions is less significant.
>
> _Low Level (Choice of Specific Approximation):_
>
> At this level, the choice becomes more nuanced.
>
> __Excess Risk Measures Based on Constant Estimates:__
>
> - Measures like $R_{exc}^{(2,3)}$​ and $R_{exc}^{(3,2)}$ are less effective at capturing epistemic uncertainty. This is expected because they "integrate out" model disagreement and rely solely on point estimates.
> - This limitation is evident in Tables 8 and 9, as well as other tables in the appendix covering both misclassification and out-of-distribution detection.
> - We plan to include a discussion of these two measures in our paper to highlight their limitations.
>
> __Other Choices:__
> - It is not straightforward to identify a single measure that consistently outperforms others. Our results indicate that different measures often perform similarly, and their effectiveness can depend on the specific problem and context.
> - For symmetric scores like the Brier score, measures differ only by a multiplicative constant, which does not affect the AUROC, resulting in the same performance.
> - Intuitive Considerations:
> - - We initially thought that $R_{exc}^{(1,1)}$​​ might perform better because it can be broken down into the sum of two other estimates (see Equation 11 in the appendix), potentially capturing more nuances.
> - - However, in the problems we studied, we did not observe a significant difference.
> - - There may be specific problem settings—such as decentralized training with high data heterogeneity between clients—where the order of arguments in the KL divergence (a special case when G is the log score) becomes important (in this case $R_{exc}^{(1,2)}$​ will be reverse KL, and $R_{exc}^{(2,1)}$​ -- forward). But we believe that exploring these scenarios could be a valuable direction for future research.

---

> ### Author Response · Authors · 2024-11-15
>
> > “Is it possible to have results on the error rate of each estimate for some common distribution families?”
>
>
> It is an interesting question! We think that such an analysis is totally possible in an asymptotic regime. In such a regime, in case of parametric models such as logistic regression, Bernstein - von Mises result on asymptotic normality of the posterior start to work. One may directly measure the resulting approximation errors and compare between them. We see it as an exciting next step of the research in this direction. The other approaches, such as finite sample PAC-Bayes analysis, are also possible but might be more involved due to the significant prior impact in finite sample analysis.

---

> ### Author Response · Authors · 2024-11-15
>
> > “For example, how are the UQ measures computed given the outputs of the deep ensemble model? And more generally, given a model that only allows access samples from the posterior, are we resorting to a regular Monte Carlo?”
>
> Yes, thank you for raising this question. Indeed, we resorted to Monte Carlo to approximate these measures with deep ensembles. This is a standard and, most probably, the only general practical way to compute these measures. One may of course consider some specific forms of the posterior distribution, like Normal, and compute these measures analytically, but we believe it may spoil the performance, as the posterior distribution is typically of very complex structure.

---

> ### Author Response · Authors · 2024-11-15
>
> > “In Section 6, what is meant by 5 groups of ensembles? Do all of them have the same architecture? I think the wording here is confusing.”
>
> Thank you again for an important question. We apologize if we missed it in our description and we will write it explicitly. Yes, all the members of the ensemble have the same architecture. They were trained completely independently with different random initializations and random seeds to the same training dataset. Effectively, we have 20 such independent models. Then, we split them into 5 groups (4 models in each group). And this is what we referred to as “5 groups of ensembles”.
> We agree the wording indeed sounds confusing and we will correct it.

---

> ### Comment · Reviewer_cjsf · 2024-11-17
> **Response after rebuttal**
>
> I deeply appreciate the authors' comprehensive response. They have effectively addressed my initial concerns and provided additional details that significantly enhance the quality of the paper. As a result, I am raising my score to reflect these improvements.

---

### Official Review · Reviewer_Ew1e · 2024-10-22

**Soundness:** 3
**Presentation:** 2
**Contribution:** 3
**Rating:** 6
**Confidence:** 3

**Summary:**

This work proposes a generalized framework from which common measures of aleatoric and epistemic uncertainty can be obtained. This is done through Proper Scoring rules and the notion of risk. The framework assumes Bayes (aleatoric) and Excess (epistemic) risk, and explains that the risk may be estimated differently with different ways to approximate the true function and the learned function. Following this they observe which proper scoring rule tends to perform well, and when aleatoric/epistemic uncertainty performs well for a certain task. Lastly, they briefly describe energy based measures.

**Strengths:**

- The paper clearly outlines its contribution. By offering a better theoretical understanding of the commonly used uncertainty measures we may progress the field of disentanglement.
- The paper covers a lot of ground in a limited span, discussing risk, proper scoring rules, different ways for risk estimation, and a substantial amount of experimental results. Overall I think the paper provides a nice overview of different uncertainty measures and how they relate.

**Weaknesses:**

- While the paper unifies various results into a shared formulation, parts of this are not entirely novel and the paper is not clear about this. For example, Schweighofer et al. (2023a)[2] already describe EPKL as a deviation from Mutual Information that does not assume the BMA to be the true model. They also discuss the relation to the reverse Mutual Information. This disagrees with the problem statement on Line 45 which suggests that it is not clear how these measures relate to each other. Similarly, results similar to Table 1 are already given by Hofman et al. [1] (see questions). Consider explicitly showing which parts of the generalization are novel, and which relationships are established (for example by [1] and [2]).
- The current work fails to assert the usefulness of the proposed generalization. I would expect a strong evaluation to create hypotheses that follow from the generalization, and validate those experimentally. Only experiments 6.2 and 6.3 come with a hypothesis, but those hypotheses generally apply to uncertainty quantification and not the proposed generalization. This severely limits the impact of the paper.
- At various locations the authors discuss that aleatoric and/or epistemic uncertainty are vaguely defined, but they do not try to maintain a precise definition following the literature. For example, they argue that epistemic uncertainty is the lack of knowledge of the right model parameters, but this would ignore model misspecification, which is accepted as a source of epistemic uncertainty. Similarly, on Line 36 authors say that aleatoric uncertainty is ambiguity in the label distribution, which ignores uncertainty in the inputs or an otherwise stochastic relationship between the features and the labels. I encourage the authors to attempt to use a more consistent and complete definition of aleatoric and epistemic uncertainty as a starting point. This is particularly relevant because the Pointwise Risk perspective seems to try to give an alternative precise definition. Consider for a source [3], Section 2.4.1.
- It is unclear why the energy-based models are discussed, as they seem to have little relevance. Please clarify the relevance or remove this section.

[1] Hofman, Paul, Yusuf Sale, and Eyke Hüllermeier. "Quantifying Aleatoric and Epistemic Uncertainty with Proper Scoring Rules." arXiv preprint arXiv:2404.12215 (2024).

[2] Schweighofer, Kajetan, et al. "Introducing an improved information-theoretic measure of predictive uncertainty." arXiv preprint arXiv:2311.08309 (2023).

[3] Gawlikowski, Jakob, et al. "A survey of uncertainty in deep neural networks." Artificial Intelligence Review 56.Suppl 1 (2023): 1513-1589.


## Additional Feedback

- In Section 6.4 it would be good to clearly point to where we can find evidence for which claim (table, row and column). This mainly applies to Lines 477-482. I cannot find the results that show R_exc(3,1) is better than both energy-based methods for misclassification detection, nor that R_Exc(3,1) is preferred over energy based measures for Soft-OOD.
- Line 456 seems overstated “all instances of excess risk should perform worse [than Bayes/Total risk on misclassification detection]”. This is true if there is more (separable) aleatoric uncertainty than epistemic uncertainty, but this is not true in general. For example in low data or high dimensional problems, this might not hold. Adding the word “typically” or “usually” could be sufficient.
- Lines 104-107 are repetitive with Lines 36-38.
- The whole paper assumes Deep Ensembles as the BNN, but different behavior may be observed with different models. It could be good to show behavior with MC-Dropout or Flipout.
- Table 5 mentioned on Line 475 is actually in the appendix, but the text presents it as part of the main body.
- In the tables it should be clear whether the plus-minus indicates standard deviation, variance or standard error. If appropriate, it may be helpful to indicate the which parts of the tables relate to conclusions drawn (highlighting in bold may be useful).
- The notation of e.g. R_Exc^{(1, 3)} is hard to keep track of as a reader. Would it be acceptable to rename “Excess” risk to epistemic risk, or Bayes risk to Aleatoric Risk? Perhaps the indices may also be substituted with abbreviations so the connection to the approximations is clear?

**Questions:**

- Previous work by Hofman et al [1] seems to be related and at least partially overlapping, but not cited. For parts of Table 1 it seems Hofman et al. give a similar result also based on proper scoring rules. From this it seems that a general framework for uncertainty measures already exists. Can the authors clarify how their proposed framework extends beyond this?

---

> ### Author Response · Authors · 2024-11-14
> **Author's response**
>
> We thank the reviewer for his/her comprehensive review.
>
>
> __1. Relation to the prior work.__
>
> We appreciate the reviewer's feedback regarding the relation to prior work.
>
> > “parts of this are not entirely novel and the paper is not clear about this”.
>
> Could reviewer please specify which parts he/she finds unclear? We would like to address any confusion and ensure our contributions are clearly presented.
>
> __Discussion on Schweighofer et al. (2023a)__
>
> We would like to emphasize that we have cited Schweighofer et al. (2023a) and properly acknowledged their work in our paper. Their study builds on the critique raised by Wimmer et al. [1] (also cited in our work) and derives the EPKL measure, which, to the best of our knowledge, was first introduced by Malinin and Gales [2] three years earlier.
>
> We believe there are key distinctions between our paper and the one by Schweighofer et al.:
> 1. __Scope of Measures Considered:__ Schweighofer et al. focus specifically on "information-theoretical" measures, which in our framework correspond to a special case when using the Log Score. Our framework is more general and can generate a broader class of uncertainty measures.
> 2. __Starting Point of Analysis:__ Their paper starts from the classical decomposition of predictive uncertainty, which already depends on model estimates (Equation 1 in their paper). In contrast, our work begins with the decomposition of risk, which depends on unknown quantities. This provides a different perspective and foundational approach.
> 3. __Generation of New Uncertainty Measures:__ Their paper does not generalize or provide methods to create new uncertainty measures. Our framework, however, offers ways to derive new measures beyond those previously considered.
>
> We kindly ask the reviewer to let us know if there are specific aspects that are not clear regarding our relation to prior work. We are more than willing to provide additional explanations or clarify any points to improve the paper.
>
> ---
>
> [1] Lisa Wimmer, Yusuf Sale, Paul Hofman, Bernd Bischl, and Eyke Hüllermeier. "Quantifying Aleatoric and Epistemic Uncertainty in Machine Learning: Are Conditional Entropy and Mutual Information Appropriate Measures?" In Uncertainty in Artificial Intelligence, pp. 2282–2292. PMLR, 2023.
>
> [2] Andrey Malinin and Mark Gales. "Uncertainty Estimation in Autoregressive Structured Prediction." In International Conference on Learning Representations, 2021.
>
> __Discussion on Hofman et al. (2024)__
>
> This paper indeed provides a similar derivation of uncertainty measures as ours. However, we can mention that the work that is was made available online 2 months after the first version of our paper was released. For this reason, we believe it would not be appropriate to cite it at this stage. Due to the double-blind review process, we cannot provide specific details about our preprint.
> Please note, that the authors of that paper are informed on the existence of our paper, and we recently contacted them. We definitely will acknowledge their work as concurrent in the camera-ready version of our paper (if accepted).
>
> To ensure fairness and adhere to the review guidelines, we have written a private note to the Area Chair (AC) or Program Chair (PC) regarding this matter.
>
> ---
>
> Reviewer mentioned:
> > “This disagrees with the problem statement on Line 45 which suggests that it is not clear how these measures relate to each other.”
>
> We would like to clarify that there is no contradiction. The paper by Schweighofer et al. (2023a) considers only Log Score-based measures, such as Mutual Information, EPKL, and Entropy. However, other measures like Variation Ratios and Mean Standard Deviations discussed in [3] (cited in Line 45) are not covered in their paper.
>
> Our paper shows that all these measures, including those not addressed by Schweighofer et al., are special cases within our general framework, derived using proper scoring rules and specific Bayesian approximations.
> We hope this explanation clarifies the statement in Line 45. Please let us know if you agree that there is no contradiction.
>
> [3] Yarin Gal, Riashat Islam, and Zoubin Ghahramani. Deep bayesian active learning with image data. In International conference on machine learning, pp. 1183–1192. PMLR, 2017.
>
> ---
>
> > “Consider explicitly showing which parts of the generalization are novel, and which relationships are established (for example by [1] and [2]).”
>
> Thank you for this helpful suggestion.
>
> Regarding Schweighofer et al. (2023a) and Malinin and Gales (2021), we have acknowledged their introduction of EPKL. We will ensure that our paper clearly delineates our original contributions, such as the general framework that unifies a broader class of uncertainty measures.
>
> For the work by Hofman et al., please refer to our earlier comment.

---

> > ### Comment · Reviewer_Ew1e · 2024-11-21
> > **Relation to Prior work**
> >
> > ## 1. Relation to prior work
> > I appreciate the rebuttal from the authors. The more substantial critiques are succesfully refuted, though some minor disagreements remain. I subsequently adjust my rating.
> >
> > ### Schweighofer et al. (2023a)
> > I agree with the authors that this paper is discussed. My specific critique is that the authors state that the relation between measures is not known, but this is not completely accurate. I consider Schweighofer et al. as an example, as they derive EPKL from the "information theoretic" approach.
> >
> > I agree with the authors that their work goes far beyond this by providing a single unification for all measures, but it might be fair to specify which relations are already known.
> >
> > ### Hofman et al. (2024)
> >
> > I trust the authors that their preprint came before Hofman et al., which means Table 1 is considered novel. I believe this does pose a substantial contribution and will therefore adjust my rating.

---

> ### Author Response · Authors · 2024-11-14
>
> __2. “the usefulness of the proposed generalization”__
>
> Thank you for your feedback regarding the usefulness of our proposed generalization. We understand that you have concerns about how our generalization contributes to the field.
>
> In the Strengths section of your review, you mentioned:
> > - "By offering a better theoretical understanding of the commonly used uncertainty measures, we may progress the field of disentanglement."
> > - "I think the paper provides a nice overview of different uncertainty measures and how they relate."
>
> We appreciate these positive remarks and believe they highlight the usefulness of our generalization. Our paper introduces a unified framework that allows for the generation of different uncertainty measures, providing deeper theoretical insights into how these measures are connected.
>
> Could you please clarify what kind of strong evaluation or hypotheses you expect to follow from our generalization? Understanding your expectations would help us address your concerns more effectively.
>
> > “Only experiments 6.2 and 6.3 come with a hypothesis, but those hypotheses generally apply to uncertainty quantification and not the proposed generalization”.
>
> We would like to clarify that all our experimental questions (which can be viewed as hypotheses) arise from our proposed generalization. For example, in Section 6.1, we explore which choice of the function G is better, and we provide an answer to this question. Without our general framework, it would be challenging to formulate and address this question.
>
> Similarly, the questions in Sections 6.2 and 6.3 are closely related to our generalization framework because we associate aleatoric uncertainty with Bayes risk and epistemic uncertainty with Excess risk. While it's true that one could define estimates of aleatoric and epistemic uncertainty in various ways, our framework offers a systematic approach to examine these concepts. We believe the relevance of these questions is important, as they help deepen the understanding of when Excess risk or Bayes risk is more useful, especially in the context of soft-OOD and hard-OOD data.
>
> Furthermore, the question in Section 6.4 is also relevant, as the connection to Energy-Based Models (EBMs) naturally arises from our framework as a special case of approximation.
>
> We hope this explanation clarifies how our experimental questions and hypotheses are connected to our proposed generalization. Please let us know if this addresses your concern or if there are specific aspects you would like us to elaborate on.

---

> > ### Comment · Reviewer_Ew1e · 2024-11-21
> > **Usefulness of the proposed generalization**
> >
> > > Could you please clarify what kind of strong evaluation or hypotheses you expect to follow from our generalization? Understanding your expectations would help us address your concerns more effectively.
> >
> > I am unable to come up with hypotheses that follow from the generalization, which is partly why I'm concerned about the usefulness. If the generalisation allows to generate new measures, or clear hypotheses follow from it that would make it more useful. I think a unification has some inherent usefulness, but I maintain that the usefulness and therefore impact are limited.
> >
> >
> > > We would like to clarify that all our experimental questions (which can be viewed as hypotheses) arise from our proposed generalization.
> >
> > The experimental questions are not a hypothesis, as there's no clear expectation. The experiments (finding the best uncertainty measure, comparing aleatoric, epistemic and total uncertainty) do not rely on the generalisation but generally follow from having multiple uncertainty measures.
> >
> > The relation to energy-based models may be interesting, but it's not clear (to me) what the value of this is.

---

> ### Author Response · Authors · 2024-11-14
>
> __3. Concern on the definition of AU and EU. “Consider for a source [3], Section 2.4.1.”__
>
> Thank you for bringing up this point and referencing the paper [3]. We appreciate your insight and believe this highlights the ongoing discussion about the definitions of uncertainty in the field.
>
> While there is general agreement that aleatoric uncertainty (AU) is irreducible and solely a property of the data, and that epistemic uncertainty (EU) is reducible, the specific details of these definitions can vary. In our work, we follow the definitions provided in [4], which we have cited and acknowledged.
>
> Specifically, Section 2.3 of [4] states:
> > "Aleatoric uncertainty refers to the irreducible part of the uncertainty, which is due to the nondeterministic nature of the sought input/output dependency, that is, to the stochastic dependency between instances x and outcomes y, as expressed by the conditional probability."
>
> According to this definition, AU focuses on the inherent randomness in the output given the input and does not explicitly include "uncertainty in the inputs," as you mentioned.
>
> Furthermore, the same section defines EU as:
> > "Model uncertainty and approximation uncertainty, on the other hand, are subsumed under the notion of epistemic uncertainty, that is, uncertainty due to a lack of knowledge about the perfect predictor, for example caused by uncertainty about the parameters of a model."
>
> Based on these definitions, we believe our approach aligns with established literature. We would appreciate it if you could clarify what you mean by using a "more consistent and complete definition of aleatoric and epistemic uncertainty as a starting point," so we can address your concern more effectively.
>
> Regarding "model misspecification" as a source of epistemic uncertainty, we totally agree, and we want to emphasize that it is included in our consideration. In our framework, this is reflected in the definition Excess Risk. If the model is misspecified, the Excess Risk cannot be fully eliminated, thereby accounting for this source of uncertainty.
>
> Regarding aleatoric uncertainty, you noted that it includes "uncertainty in the inputs or an otherwise stochastic relationship between the features and the labels." We agree with this perspective. In line 34 of our paper, we intended to convey the latter part of your statement, and we will revise the wording to make this clearer.
>
> As for the "uncertainty in the inputs," this is an insightful remark. In our current framework, we assume the common setup where we have access to the true inputs $x$ without noise. Considering scenarios where only noisy instances $\tilde{x}$ are available is indeed interesting and could extend our work to new problem settings. While this is less typical in the literature, it presents a valuable direction for future research. We will include a corresponding remark in the text.
>
> Thank you again for your thoughtful feedback. We hope this addresses your concern, and we are open to further discussion to clarify any remaining issues.
>
> ---
>
> [4] Eyke Hüllermeier and Willem Waegeman. Aleatoric and epistemic uncertainty in machine learning: An introduction to concepts and methods. Machine Learning, 110:457–506, 2021.

---

> ### Author Response · Authors · 2024-11-14
>
> __4. “It is unclear why the energy-based models are discussed, as they seem to have little relevance. Please clarify the relevance or remove this section”__
>
> Thank you for expressing your concern about the discussion of energy-based models (EBMs) in our paper. We appreciate the opportunity to clarify their relevance.
>
> In our framework, EBMs naturally appear as a special case of Bayesian approximation. Specifically, the Excess Risk we analyze corresponds to the difference between energies in a Bayesian model induced by different potentials—that is, the energy induced by the averaged potential versus the average energy generated by individual potentials. This connection is significant because EBMs have been previously used in out-of-distribution detection, although not within a Bayesian context.
>
> We find it interesting that EBMs appear organically in our framework, and we believe this is a valuable finding to share. EBMs are a popular class of models for OOD detection. We have cited a key paper in this area [5], and we think that including another influential work [6] would strengthen this section.
>
> We hope this explanation clarifies why we discuss EBMs in our paper. Please let us know if you have any further questions or if anything remains unclear.
>
>
> [5] Weitang Liu, Xiaoyun Wang, John Owens, and Yixuan Li. "Energy-based Out-of-Distribution Detection." Advances in Neural Information Processing Systems, 33:21464–21475, 2020.
>
> [6] Grathwohl, W., Wang, K. C., Jacobsen, J. H., Duvenaud, D., Norouzi, M., & Swersky, K. "Your classifier is secretly an energy based model and you should treat it like one." International Conference on Learning Representations, 2020.

---

> ### Author Response · Authors · 2024-11-14
> **Response to Additional Feedback [1/2]**
>
> > In Section 6.4 it would be good to clearly point to where we can find evidence for which claim (table, row and column). This mainly applies to Lines 477-482. I cannot find the results that show R_exc(3,1) is better than both energy-based methods for misclassification detection, nor that R_Exc(3,1) is preferred over energy based measures for Soft-OOD.
>
> Thank you for pointing out the need to clearly indicate where the evidence for our claims can be found. We appreciate your feedback and apologize for any confusion. Here are the specific locations in the paper where you can find the supporting results.
>
> *R_exc(3,1) is better than both energy-based methods for misclassification detection:*
>
> - Table 10, line 1366 (R_exc(3,1)) vs lines 1371-1372 (energies). All datasets.
> - Table 13, line 1480 (R_exc(3,1)) vs lines 1501-1502 (energies). TinyImageNet. Here the results are close (R_exc(3,1) is in the middle of two energy estimates).
>
> *R_Exc(3,1) is preferred over energy based measures for Soft-OOD:*
>
> - Table 3, lines 442 (R_exc(3,1)) vs lines 446-447 (energies). ImageNet-O.
> - Table 5, lines 1203-1204 (R_exc(3,1)) vs lines 1209-1210 (energies). CIFAR10C-[1, 2]
> - Table 9, lines 1339-1340 (R_exc(3,1)) vs line 1349 (energies). ImageNet-O.
>
> > Line 456 seems overstated “all instances of excess risk should perform worse [than Bayes/Total risk on misclassification detection]”. This is true if there is more (separable) aleatoric uncertainty than epistemic uncertainty, but this is not true in general. For example in low data or high dimensional problems, this might not hold. Adding the word “typically” or “usually” could be sufficient.
>
> You are correct; our statement on Line 456 may be overstated. We considered the scenario where there is sufficient data, but as you pointed out, in situations with limited data or high-dimensional problems, this might not hold true. Excess risk does not always perform worse than Bayes or total risk in misclassification detection under all circumstances. We agree with your suggestion and will adjust the wording by adding "typically" or "usually" to reflect this nuance.
>
> > Lines 104-107 are repetitive with Lines 36-38.
>
> We agree that these lines convey similar information. Our intention was to recap the key points to reinforce understanding for the reader. However, we understand your concern about redundancy. If you feel that this repetition is unnecessary, we are happy to remove one of these sections to improve the clarity of the paper.
>
> > The whole paper assumes Deep Ensembles as the BNN, but different behavior may be observed with different models. It could be good to show behavior with MC-Dropout or Flipout.
>
> You are absolutely correct that different models may exhibit different behaviors, and exploring them could provide additional depth to our study.
> We chose to focus on Deep Ensembles because they typically offer strong performance and are straightforward to train. Deep Ensembles are well-regarded for their effectiveness in uncertainty quantification, which aligns with the goals of our paper.
>
> However, we agree that evaluating other methods like MC-Dropout or Flipout could enhance our work. We will make our best effort to incorporate additional results using these approaches before the end of the rebuttal period.
>
>
> > Table 5 mentioned on Line 475 is actually in the appendix, but the text presents it as part of the main body.
>
> We agree with the reviewer. We will fix it, thank you!
>
> > In the tables it should be clear whether the plus-minus indicates standard deviation, variance or standard error. If appropriate, it may be helpful to indicate the which parts of the tables relate to conclusions drawn (highlighting in bold may be useful).
>
> Again, we agree with the reviewer and we will incorporate proposed changes. Thank you.

---

> ### Author Response · Authors · 2024-11-14
> **Response to Additional Feedback [2/2]**
>
> > The notation of e.g. R_Exc^{(1, 3)} is hard to keep track of as a reader. Would it be acceptable to rename “Excess” risk to epistemic risk, or Bayes risk to Aleatoric Risk? Perhaps the indices may also be substituted with abbreviations so the connection to the approximations is clear?
>
> We agree that using more intuitive names could help make the paper easier to read. However, we chose to use the conventional terms "Excess Risk" and "Bayes Risk" from statistics to stay consistent with established terminology.
>
> Regarding your suggestion to rename "Excess Risk" to "Epistemic Risk" and "Bayes Risk" to "Aleatoric Risk," we considered this option. We felt that keeping the traditional names would avoid confusion for readers familiar with the standard definitions in statistics.
>
> As for substituting the indices with abbreviations to make the connection to the approximations clearer, we appreciate the idea. We did think about using abbreviations, but with so many possible combinations, it became challenging to create unique and meaningful names for all of them. We were concerned that this might make the notation even more complicated.
>
> We appreciate your suggestion and are committed to improving the clarity of our work. Thank you for helping us make our paper better.

---

> ### Author Response · Authors · 2024-11-14
>
> __Questions__
>
> > Previous work by Hofman et al [1] seems to be related and at least partially overlapping, but not cited. For parts of Table 1 it seems Hofman et al. give a similar result also based on proper scoring rules. From this it seems that a general framework for uncertainty measures already exists. Can the authors clarify how their proposed framework extends beyond this?
>
> As mentioned above, due to the double-blind review policy, we are limited in how much detail we can provide about this specific paper without risking a violation of anonymity.
>
> The paper you referenced is not the first introducing generalization of uncertainty measures based on proper scoring rules. Please note, that the authors of that paper are informed on the existence of our paper, and we recently contacted them.
>
> We hope you understand our constraints due to the review process, and we are happy to provide further clarification within those limits.

---

### Official Review · Reviewer_fVqT · 2024-10-23

**Soundness:** 3
**Presentation:** 3
**Contribution:** 2
**Rating:** 6
**Confidence:** 4

**Summary:**

The authors introduce a framework for quantifying uncertainty by decomposing it into aleatoric and epistemic. The authors use Bayesian methods to approximate and derive a unified framework for generating uncertainty measures. The framework is validated through experiments on classification tasks for image datasets, specifically on out-of-distribution detection.

**Strengths:**

1. The paper is well written and clear.
2. The paper presents a solid theoretical contribution by unifying different predictive uncertainty measures under a Bayesian risk decomposition framework. It connects well-known uncertainty quantification methods (e.g., Mutual Information, Expected Pairwise KL Divergence) with a common theoretical foundation, which adds clarity and depth to the topic.

**Weaknesses:**

### 1. Limited Scope of Experimental Section

The paper's experimental section only considers image classification tasks, which have been extensively researched in the uncertainty community. It could be strengthened by:

   - **Broadening the task domain**: The paper could provide results on regression tasks to bolster the experimental validation.
   - **Exploring state-of-the-art methods**: Consideration of more cutting-edge methods, such as generative models for uncertainty quantification, could enhance the experimental insights.

### 2. Redundant Conclusions

Given the lack of novelty in experimental design (see 1), the conclusions drawn in the experimental section are similar to those in previous works [1, 2], which limits the novelty of the paper’s contributions to the literature.

### 3. Assumption of Well-Approximated Data Distributions

The framework relies on the decomposition of total risk into Bayes risk and excess risk, which assumes that the underlying true data distribution can be well approximated. In practice, this assumption may not hold in many real-world scenarios, especially with complex, high-dimensional data.

### 4. Lack of Baseline Comparisons

The paper appears to lack comparison to strong baselines. While the authors state:

> "We emphasize that the goal of our experimental evaluation is not to provide new state-of-the-art measures or compete with other known approaches for uncertainty quantification."

They also mention:

> "Instead, we aim to verify whether different uncertainty estimates are indeed related to specific types of uncertainty."

However, the paper fails to provide comparisons to modern estimates of uncertainty such as [3, 4], which could help contextualize their findings and strengthen the evaluation.

### 5. Lack of training details

The authors appear to have omitted important training details in Appendix H. Specifically, it is unclear:

   - What loss function was used to train the models?
   - How were the ensembles trained? Were they trained as completely independent networks, or did they share components?

Providing these details is crucial for reproducibility and understanding how the proposed framework was implemented.

[1] Lisa Wimmer, Yusuf Sale, Paul Hofman, Bernd Bischl, and Eyke Hüllermeier. Quantifying aleatoric and epistemic uncertainty in machine learning: Are conditional entropy and mutual information appropriate measures? In Uncertainty in Artificial Intelligence, pp. 2282–2292. PMLR, 2023.

[2] Kajetan Schweighofer, Lukas Aichberger, Mykyta Ielanskyi, and Sepp Hochreiter. Introducing an improved information-theoretic measure of predictive uncertainty. In NeurIPS 2023 Workshop on Mathematics of Modern Machine Learning, 2023a.

[3] Berry, Lucas, and David Meger. "Escaping the sample trap: Fast and accurate epistemic uncertainty estimation with pairwise-distance estimators." arXiv preprint arXiv:2308.13498 (2023).

[4] Chan, Matthew A., Maria J. Molina, and Christopher A. Metzler. "Hyper-Diffusion: Estimating Epistemic and Aleatoric Uncertainty with a Single Model." arXiv preprint arXiv:2402.03478 (2024).

**Questions:**

Does the loss function to quantify uncertainty and the training loss function need to be the same?

---

> ### Author Response · Authors · 2024-11-14
> **Author's response**
>
> We would like to thank the reviewer for their thoughtful feedback.
>
> We would like to kindly clarify one point in your summary: while we conducted experiments on out-of-distribution detection, we also considered misclassification detection as a downstream task in our experimental evaluation.
>
> __1. Limited Scope of Experimental Section__
>
> > __“experimental section only considers image classification tasks, which have been extensively researched in the uncertainty community”__
>
> We appreciate your concern about the scope of our experimental evaluation. It's true that image classification is a well-explored area in uncertainty quantification research. However, despite extensive studies, there is still no "silver bullet" that perfectly addresses uncertainty quantification in this domain. This is why many different uncertainty measures have been proposed, each with varying degrees of success. What has been missing is a clear understanding of how these measures relate to each other.
>
> As you kindly noted in your "Strength" section:
> > “The paper presents a solid theoretical contribution by unifying different predictive uncertainty measures”...”It connects well-known uncertainty quantification methods (e.g., Mutual Information, Expected Pairwise KL Divergence) with a common theoretical foundation, which adds clarity and depth to the topic.”
>
> We believe our primary contribution is this unifying theoretical framework. By showing that various existing measures are special cases of a general approach, we provide new insights that add clarity to the field.
>
> In the experimental part, we aim to showcase some of the important properties of the derived uncertainty measures. To achieve this, we focused on the image domain to ensure consistency and clarity of the results. Many recognized papers in machine learning research, particularly in uncertainty quantification, validate their ideas using a single domain like image classification. The papers you cited in your review [1, 2] also focus solely on this domain. While we understand the importance of broader experimental validation, we believe our focus is appropriate given the theoretical nature of our contribution.
>
> Given that you acknowledged our "solid theoretical contribution," we are a bit confused about why our contribution was evaluated as poor (1). Could you please clarify this point so we can better understand your concerns and improve our work?
>
> ---
>
> [1] Lisa Wimmer, Yusuf Sale, Paul Hofman, Bernd Bischl, and Eyke Hüllermeier. Quantifying aleatoric and epistemic uncertainty in machine learning: Are conditional entropy and mutual information appropriate measures? In Uncertainty in Artificial Intelligence, pp. 2282–2292. PMLR, 2023.
>
> [2] Kajetan Schweighofer, Lukas Aichberger, Mykyta Ielanskyi, and Sepp Hochreiter. Introducing an improved information-theoretic measure of predictive uncertainty. In NeurIPS 2023 Workshop on Mathematics of Modern Machine Learning, 2023a.

---

> ### Author Response · Authors · 2024-11-14
>
> __2. Redundant Conclusions__
>
> > __“Given the lack of novelty in experimental design (see 1)...”__
>
> Thank you for your feedback. We would like to clarify our understanding of your concern. In your first point, you mentioned a lack of experimental evaluation rather than a lack of novelty in experimental design. Regarding our experimental design, we believe we addressed an interesting new problem by considering both soft-OOD (out-of-distribution) and hard-OOD data types. This approach allowed us to show that Excess risk is more suitable for soft-OOD data, while challenges may arise in hard-OOD scenarios. To the best of our knowledge, it was not explicitly evaluated in the previous work.
>
> > __“the conclusions drawn in the experimental section are similar to those in previous works [1, 2]”__
>
> We are a bit unclear about which specific conclusions you are referring to. The two papers you mentioned seem to address different issues:
> - __Wimmer et al.__ critique Mutual Information with respect to a set of introduced axioms.
> - __Schweighofer et al.__ introduce a new uncertainty quantification measure (EPKL) that satisfies more axioms.
>
> It appears there are no shared conclusions between these papers and ours. Could you please specify which conclusions you believe are similar?
>
> Moreover, if there are shared conclusions, we view this as a positive sign. Consistency with previous work can reinforce the validity of our findings. Since our paper aims to generalize previous studies, it's reassuring that our conclusions align rather than contradict earlier research.
>
> Could you please clarify your concern? Why are consistent conclusions in a work that generalizes prior studies considered a drawback? Are there specific conclusions you believe overlap significantly?

---

> ### Author Response · Authors · 2024-11-14
>
> __3. Assumption of Well-Approximated Data Distributions__
>
> Thank you for raising this important concern.
>
> You mentioned:
> > “The framework relies on the decomposition of total risk into Bayes risk and excess risk, which assumes that the underlying true data distribution can be well approximated. In practice, this assumption may not hold in many real-world scenarios, especially with complex, high-dimensional data.”
>
> We would like to clarify that the decomposition into Bayes risk and Excess risk is a general concept that does not depend on specific assumptions, except for the existence of the true conditional distribution $p(y \mid x)$. This distribution can be arbitrarily complex, depending on the problem at hand.
>
> In supervised learning tasks like classification and regression, our goal is to approximate $p(y \mid x)$ with an estimated distribution $\hat{p}(y \mid x)$, which is often modeled using neural networks in modern deep learning. Despite the complexity of the true distribution $p(y \mid x)$, neural networks have shown practical success in approximating it effectively.
>
> We are not entirely sure whether your concern about "complex, high-dimensional data" refers to the input features $x$ or the labels $y$. In either case, neural networks have been effective for both scenarios:
> - __High-Dimensional Inputs (x):__ Neural networks handle high-resolution images and other complex data types well.
> - __Complex Labels (y):__ They also perform successfully in tasks with many class labels, such as ImageNet classification with 1,000 classes.
>
> Therefore:
>
> - __Neural Networks as Approximators:__ Neural networks are suitable tools for approximating the true distribution $p(y \mid x)$, even in complex, high-dimensional settings.
> - __General Decomposition:__ The decomposition into Bayes risk and Excess risk is general and does not rely on the assumption that p(y \mid x) can be well approximated.
>
> We hope this addresses your concern. If we have misunderstood your point or if there are specific aspects you would like us to elaborate on, we would appreciate further clarification.

---

> ### Author Response · Authors · 2024-11-14
>
> __4. Lack of Baseline Comparisons__
>
> The reviewer says:
>
> > ‘They [authors] also mention:
> "Instead, we aim to verify whether different uncertainty estimates are indeed related to specific types of uncertainty."
> However, the paper fails to provide comparisons to modern estimates of uncertainty such as [3, 4], which could help contextualize their findings and strengthen the evaluation.’
>
>
> We appreciate your feedback and would like to clarify our approach. Our main goal was to check if different uncertainty estimates are related to specific types of uncertainty, namely aleatoric and epistemic uncertainty. To do this, we conducted two experiments in different downstream tasks, each designed to present a specific type of uncertainty. We then examined whether our estimates could capture these uncertainties.
>
> Since our focus was on exploring the uncertainty measures within our framework, rather than benchmarking all known uncertainty measures, we did not see the need to implement or compare with other methods. We believe this approach is appropriate for the scope of our theoretical contribution.
>
> We are grateful for the references you provided to the recent preprints [3] and [4], which we were not aware of. We are happy to include them in the related work section of our paper.
>
> However, we believe that including these methods in our experimental comparison may not be directly relevant. For example, [4] proposes a computationally intensive approach where a diffusion model generates the parameters of a neural network. This method may be impractical for many real-world applications due to its high computational cost. Additionally, they used unconventional baselines in their experiments, making it difficult to assess how their approach scales to the datasets we used.
> Regarding [3], their work focuses on reinforcement learning tasks, not on computer vision classification, which is the domain of our experiments. Therefore, a direct comparison may not be appropriate.
>
> We would like to emphasize that the main contribution of our paper is theoretical, as you kindly acknowledged in your review under the Strengths section. Our aim is to provide a unifying theoretical framework for uncertainty measures, and we believe our experiments support this goal.

---

> ### Author Response · Authors · 2024-11-14
>
> __5. Lack of training details__
>
> >“What loss function was used to train the models?”
>
> Thank you for pointing this out. We agree that the details about the loss functions could be clearer. We used different loss functions generated by different instances of G. The specific loss function used for training is mentioned in the captions of the relevant tables.
> For example:
> - In Tables starting from Table 5, we explicitly state the loss function in the caption.
> - In Table 2, we mention in the text (lines 392-393) that we used the same G for both the loss function and the evaluation. This means that, for instance, in Table 2 (Log), we used Cross-Entropy as the loss function and employed log-score-based uncertainty measures.
> - For Tables 3 and 4, we followed the same approach, as noted in their captions.
>
> We are open to suggestions on how we can improve clarity in this regard. Would it help if we explicitly emphasized in Appendix H that the loss functions are specified in the table captions? Please let us know if this would address your concern.
>
> > "How were the ensembles trained? Were they trained as completely independent networks, or did they share components?"
>
> The ensembles were trained as completely independent networks without sharing any components. We will explicitly mention this in both the experimental description section and in Appendix H.
>
> Would this clarification resolve your concern?

---

> ### Author Response · Authors · 2024-11-14
>
> __Questions__
>
> > Does the loss function to quantify uncertainty and the training loss function need to be the same?
>
> Thank you for this very insightful question; we appreciate the reviewer's attention to this detail.
>
> Initially, we believed that matching the loss function for training and uncertainty quantification might be beneficial. We observed that this approach tends to perform slightly better on average. However, a more detailed analysis showed that this is not always the case. Sometimes using the same loss function for both training and uncertainty estimation leads to improvements, but other times it may not perform as well.
>
> We also noticed that in very specific situations—such as a particular combination of dataset, model architecture, and choice of G (for example, with the CIFAR100 dataset, training with the Brier score, and using the Spherical loss for uncertainty quantification)—there were occasional drops in performance for a few random seeds. However, these instances were rare, and we did not observe consistent issues across our experiments.
>
> In summary, while using the same loss function for both training and uncertainty quantification can sometimes offer advantages, it is not universally better. The effectiveness may vary depending on the specific conditions, and we recommend considering the particular context when choosing the loss functions.

---

> ### Author Response · Authors · 2024-11-22
>
> Dear Reviewer fVqT,
>
> As we approach the end of the author-reviewer discussion period, we wanted to kindly follow up regarding our revised manuscript and responses to your comments. We have addressed all the concerns you raised and incorporated the changes you suggested into the updated paper.
>
> If you have any remaining questions or additional feedback, please let us know so we can respond before the discussion period concludes. Your input is valuable to us, and we are committed to ensuring that our paper meets your expectations.
>
> We greatly appreciate your time and consideration.
>
> Best regards,
>
> The Authors

---

> ### Author Response · Authors · 2024-11-22
>
> Dear Reviewer fVqT,
>
> In response to your concern about the lack of baselines, we have conducted an additional experiment on the out-of-distribution detection problem using DDU [1] and DUQ [2]. In this experiment, we used a ResNet18 network trained on the CIFAR10 dataset and computed the AUROC for detecting out-of-distribution samples on the following datasets: CIFAR100, SVHN, and TinyImageNet.
>
> The results are presented below:
>
> **DUQ:**
>
> * CIFAR10 vs CIFAR100: 85.33
> * CIFAR10 vs SVHN: 84.21
> * CIFAR10 vs TinyImageNet: 85.04
>
> **DDU:**
>
> * CIFAR10 vs CIFAR100: 86.79
> * CIFAR10 vs SVHN: 96.89
> * CIFAR10 vs TinyImageNet: 86.75
>
> The corresponding best mean results for our measures with deep ensembles (based on the LogScore instantiation) are:
>
> * CIFAR10 vs CIFAR100: 91.36
> * CIFAR10 vs SVHN: 96.01
> * CIFAR10 vs TinyImageNet: 90.84
>
> These results show that our framework works well and even outperforms baseline methods specifically designed for out-of-distribution detection that explicitly utilize the fitted density of a trained model's representations.
>
> [1] Mukhoti, J., Kirsch, A., van Amersfoort, J., Torr, P. H., & Gal, Y. (2021). Deep deterministic uncertainty: A simple baseline. arXiv preprint arXiv:2102.11582.
>
> [2] Van Amersfoort, J., Smith, L., Teh, Y. W., & Gal, Y. (2020, November). Uncertainty estimation using a single deep deterministic neural network. In International Conference on Machine Learning (pp. 9690-9700). PMLR.

---

> ### Comment · Reviewer_fVqT · 2024-11-25
>
> 1.
> Thank you for addressing this comment. While I acknowledge the strong theoretical contributions of the paper, I evaluate papers based on both theoretical and experimental contributions. That said, my initial evaluation was too harsh, and I have adjusted my score accordingly. I also believe it was an oversight that previous works, such as [1, 2], did not consider a 1D toy regression problem—especially given that these papers discuss uncertainty more broadly, not just within the classification domain.
>
> 2.
> Thank you for further emphasizing the novelty of your work. However, reaching conclusions similar to previous studies suggests less novelty and limits the paper's added value to the research community. In bold, the paper states:
>
> > This highlights a crucial limitation of Excess risk (which includes ubiquitous BI, RBI, and EPBD) as a measure of epistemic uncertainty.
>
> This seems to indicate a significant finding. However, three sentences later, the authors acknowledge that this aligns with previous studies' results [1, 2], thus implying that the result may not be as novel as initially suggested.
>
> 3.
> Thank you for your explanation. I believe my initial understanding of this aspect was incorrect.
>
> 4.
> Thank you for your feedback. Just for your reference, [3] addresses supervised learning problems in robot dynamics, not reinforcement learning tasks. I understand this may seem like a minor point, but I wanted to ensure accuracy.
>
> 5.
> Thank you for adding the training details. I believe these additions make the results more reproducible, and I appreciate the authors’ efforts to address this.
>
> I have adjusted my score to reflect the quality of the improved draft. I also want to thank the authors for addressing my comments and clarifying some misunderstandings on my part. However, I remain concerned about the lack of regression problems in uncertainty-focused papers, especially since many of these papers discuss uncertainty in a broader context.
>
>
> [1] Lisa Wimmer, Yusuf Sale, Paul Hofman, Bernd Bischl, and Eyke Hüllermeier. Quantifying aleatoric and epistemic uncertainty in machine learning: Are conditional entropy and mutual information appropriate measures? In Uncertainty in Artificial Intelligence, pp. 2282–2292. PMLR, 2023.
>
> [2] Kajetan Schweighofer, Lukas Aichberger, Mykyta Ielanskyi, and Sepp Hochreiter. Introducing an improved information-theoretic measure of predictive uncertainty. In NeurIPS 2023 Workshop on Mathematics of Modern Machine Learning, 2023a.
>
> [3] Berry, Lucas, and David Meger. "Escaping the sample trap: Fast and accurate epistemic uncertainty estimation with pairwise-distance estimators." arXiv preprint arXiv:2308.13498 (2023).

---

### Author Response · Authors · 2024-11-21
**New revision**

Dear Reviewers,

We sincerely appreciate your thoughtful reviews and the valuable feedback you provided. We have carefully considered all your comments and have made the corresponding changes to the paper. Specifically, we have:

- Included the missing references on credal sets and recent methods for uncertainty quantification.
- Expanded the discussion of related work, particularly on information-theoretical measures.
- Added a simulation toy experiment to illustrate our concepts.
- Introduced other corrections and suggestions you proposed.

We hope that these revisions address your concerns and enhance the quality of the paper. Please let us know if there are any remaining issues or suggestions.

We would also appreciate if you could consider updating your scores based on the revised manuscript.

Thank you once again for your time and consideration.

Additional experiments (DDU, DUQ, as well as Dropout and Flipout approximations) will be added as soon as they ready, we hope today or tomorrow.

---

### Meta-Review · Area_Chair_vq3o · 2024-12-16

**Metareview:**

This study provides a novel and unified perspective on evaluating predictive uncertainty in classification problems, specifically addressing aleatoric and epistemic uncertainty. It proposes a systematic decomposition method for total loss based on proper scoring rules, enabling a unified re-derivation of various estimators for aleatoric and epistemic uncertainty found in existing studies. One weakness of this study is that, while it unifies various estimators, it offers limited insights into which estimators are most effective. Despite this limitation, the study makes significant contributions by advancing a unified and Bayesian perspective on uncertainty evaluation. It provides new insights into aleatoric and epistemic uncertainty through numerical experiments, offering important implications for addressing real-world problems in uncertainty evaluation. For these reasons, the study is deemed suitable for acceptance.

**Additional Comments On Reviewer Discussion:**

Although all reviewers initially pointed out that the relationship between this study and existing research was inadequate, the authors successfully addressed these concerns in their rebuttal and the revised manuscript now includes sufficient discussion to clarify these points. Additionally, reviewers fVqT and cjsf raised concerns about the lack of numerical experiments, particularly regarding alternative methods and uncertainty evaluation criteria. These issues were appropriately addressed through the inclusion of additional experiments, such as those using toy data.

As a result, all reviewers now hold a positive evaluation of this study.

---

### Decision · Program_Chairs · 2025-01-22

Accept (Poster)